

# Quality assessment of the Ozone_cci Climate Research Data Package (release 2017): 2. Ground-based validation of nadir ozone profile data products

Arno Keppens[1], Jean-Christopher Lambert[1], José Granville[1], Daan Hubert[1], Tijl Verhoelst[1], Steven Compernolle[1], Barry Latter[2], Brian Kerridge[2], Richard Siddans[2], Anne Boynard[3,4], Juliette Hadji-Lazaro[3], Cathy Clerbaux[3,5], Catherine Wespes[5], Daniel R. Hurtmans[5], Pierre-François Coheur[5], Jacob C. A. van Peet[6], Ronald J. van der A[6], Katerina Garane[7], Maria Elissavet Koukouli[7], Dimitris S. Balis[7], Andy Delcloo[8], Rigel Kivi[9], Réné Stübi[10], Sophie Godin-Beekmann[3], Michel Van Roozendael[1], Claus Zehner[11]

[1]Royal Belgian Institute for Space Aeronomy (BIRA-IASB), Brussels, 1180, Belgium
[2]Rutherford Appleton Laboratory (RAL) & National Centre for Earth Observation (NCEO), Chilton, Didcot, OX11, UK
[3]LATMOS/IPSL, UPMC Univ. Paris 06 Sorbonne Universités, UVSQ, CNRS, Paris, 78280, France
[4]SPASCIA, Ramonville-Saint-Agne, 31520, France
[5]Université Libre de Bruxelles (ULB), Brussels, 1050, Belgium
[6]Royal Netherlands Meteorological Institute (KNMI), De Bilt, 3731, the Netherlands
[7]Laboratory of Atmospheric Physics, Aristotle University of Thessaloniki (AUTH), Thessaloniki, 54124, Greece
[8]Royal Meteorological Institute of Belgium (RMIB), Brussels, 1180, Belgium
[9]Finnish Meteorological Institute (FMI-ARC), Sodankylä, 99601, Finland
[10]Federal Office of Meteorology and Climatology, Payerne, 1530, Switzerland
[11]European Space Agency (ESA/ESRIN), Frascati, 00044, Italy

*Correspondence to*: Arno Keppens (arno.keppens@aeronomie.be)

**Abstract.** Atmospheric ozone plays a key role in air quality and the radiation budget of the Earth, both directly and through its chemical influence on other trace gases. Assessments of the atmospheric ozone distribution and associated climate change therefore demand accurate vertically-resolved ozone observations with both stratospheric and tropospheric sensitivity, both on the global and regional scales, and both in the long term and at shorter timescales. Such observations have been acquired by two series of European nadir-viewing ozone profilers, namely the scattered-light UV-visible spectrometers of the GOME family, launched regularly since 1995 (GOME, SCIAMACHY, OMI, GOME-2A/B, TROPOMI, and the upcoming Sentinel-5 series), and the thermal infrared emission sounders of the IASI type, launched regularly since 2006 (IASI on Metop platforms and the upcoming IASI-NG on Metop-SG). In particular, several Level-2 retrieved, Level-3 monthly gridded, and Level-4 assimilated nadir ozone profile data products have been improved and harmonised in the context of the ozone project of the European Space Agency's Climate Change Initiative (ESA Ozone_cci). To verify their fitness-for-purpose, these ozone datasets must undergo a comprehensive quality assessment (QA), including (a) detailed identification of their geographical, vertical and temporal domains of validity, (b) quantification of their potential bias, noise and drift and their dependences on major influence quantities, and (c) assessment of the mutual consistency of data from different sounders. For this purpose we have applied to the Ozone_cci Climate Research Data Package (CRDP) released in 2017 the versatile



QA/validation system Multi-TASTE which has been developed in the context of several heritage projects (ESA's Multi-TASTE, EUMETSAT's O3M-SAF, and the European Commission's FP6 GEOmon and FP7 QA4ECV). This work, as the second in a series of four Ozone_cci validation papers, reports for the first time on data content studies, information content studies and ground-based validation for both the GOME- and IASI-type climate data records combined. The ground-based
reference measurements have been provided by the Network for the Detection of Atmospheric Composition Change (NDACC), NASA's Southern Hemisphere Additional Ozonesonde programme (SHADOZ), and other ozonesonde and lidar stations contributing to the World Meteorological Organisation's Global Atmosphere Watch (WMO GAW). Dependence of the Ozone_cci data quality on major influence quantities – resulting in data screening suggestions to users – and perspectives for the Copernicus Sentinel missions are discussed.

## 1 Introduction

Climate studies related to atmospheric composition and the Earth's radiation budget require accurate monitoring of the horizontal and vertical distribution of ozone on the global scale and in the long term (WMO, 2010). Atmospheric ozone concentration profiles have been retrieved from solar backscatter ultraviolet radiation measurements by nadir viewing satellite spectrometers since the 1960s, starting with the USSR Kosmos missions in 1964-1965 (Iozenas et al., 1969) and
NASA's Orbiting Geophysical Observatory in 1967-1969 (Anderson et al., 1969) and BUV on Nimbus 4 in 1970-1975 (Heath et al., 1973), and continuing with the SBUV(/2) series after 1978, the GOME family of sensors since 1995, and the OMPS-nadir series started in 2011. Thermal infrared emission measurements of the ozone profile by nadir viewing satellite spectrometers were introduced more recently with Aura TES in 2004 and the series of Metop IASI since 2006. Over the past decades these retrievals have been frequently quality-checked and often improved in order to meet climate research user
requirements like the Global Climate Observing System (GCOS) targets (WMO, 2010). Yet both the verification of retrieval algorithm updates and the validation of their outputs against fiducial reference measurements (FRM) are still essential parts of the climate monitoring process, to be performed by specialised independent groups (Donlon, 2014; Loew et al., 2017).

The data quality assessment presented in this work (as part of a series of four papers addressing total ozone columns, nadir ozone profiles, limb ozone profiles, and tropical tropospheric ozone columns, respectively) has been performed in the
context of the European Space Agency's Climate Change Initiative (ESA CCI), aiming at better using satellite data records for the monitoring of essential climate variables (ECV) (http://www.esa-ozone-cci.org/). A major goal of the Ozone_cci subproject is to produce time series of tropospheric and stratospheric ozone distributions from current and historical missions that meet the requirements for reducing the uncertainty in estimates of global radiative forcing. Yet Keppens et al. (2015), based on analysis principles discussed by Rodgers (2000), have illustrated that the comparison of nadir (ozone) profiles with
FRM, although very informative on a specific data product, usually is insufficient to fully appreciate the relative quality of different retrieval products and to verify their compliance with user requirements. The present work therefore adopts the more exhaustive seven-step evaluation approach established in Keppens et al., 2015, including (1) satellite data collection



and post-processing, (2) dataset content study, (3) information content study, (4) FRM data selection, (5) co-located datasets study, (6) data harmonisation, and (7) comparative analyses and their dependences on physical influence quantities of relevance.

Section 2 first introduces the vertical profile retrieval schemes that have been used to generate the ESA Ozone_cci nadir
profile (NP) Climate Research Data Package (CRDP). These are namely the RAL version 2.14 for the backscatter UV-visible instruments and the FORLI (Fast Optimal Retrievals on Layers for IASI) version 20151001 for the thermal infrared mission instruments, developed at the Rutherford Appleton Laboratory (RAL, United Kingdom) and by the French-Belgian ULB/LATMOS cooperation, respectively. The RAL processor has been applied to retrieve L2 NP from the ERS2 GOME, Envisat SCIAMACHY, Metop-A GOME-2, Metop-B GOME-2, and AURA OMI instruments, while the FORLI algorithm
has retrieved Metop-A and Metop-B IASI ozone profiles. Sections 3 to 5 then describe the validation approach and the FRM data selection, data and information content studies, and report on the comparative validation analyses, respectively. Section 6 concludes with general discussions of the results and with an assessment of the compliance with GCOS requirements for vertically-resolved ozone climate modelling, e.g. in view of CCI contributions to the Tropospheric Ozone Assessment Report (TOAR).

## 2 Ozone_cci nadir ozone profile CRDP

### 2.1 CRDP overview

The 2017 release of the ESA Ozone_cci Climate Research Data Package contains thirteen nadir ozone profile products in total, as listed in Table 1, and a description of their associated uncertainties. The latter are included in the comparison results discussion presented in Section 5. The time span of the products is indicated in Table 2. All five level-2 (L2) backscatter
UV-VIS instrument retrievals are performed by the Rutherford Appleton Laboratory (RAL, UK) algorithm, while the infrared thermal emission measurements of the IASI instruments are processed by a collaboration between the Belgian ULB (Université Libre de Bruxelles, Belgium) and the French LATMOS (Laboratoire Atmosphères, Milieux, Observations Spatiales, Paris, France), using their FORLI (Fast Optimal Retrievals on Layers for IASI) algorithm.

Monthly-averaged level-3 (L3) products and assimilated level-4 (L4) atmospheric fields of the ozone profile are produced
from the L2 UV-VIS data by the Royal Meteorological Institute of the Netherlands (KNMI). The L4 product is generated by assimilation of the L2 GOME and GOME-2A products (NP_GOME and NP_GOME2A). Version 0004 of the L3 and L4 products has been considered in this work (see Table 1). For the thermal infrared IASI instrument on Metop-A, only a tropospheric L3 product (prefix TTC instead of NP in Table 1) has been generated by the ULB/LATMOS team, of which the first release (version 0001) is under study in this work.



| Satellite/instrument | Level | CCI CRDP product ID | Processor |
|---|---|---|---|
| ERS-2 GOME | | NP_GOME | RAL v2.14 |
| Envisat SCIAMACHY | | NP_SCIAMACHY | RAL v2.14 |
| Metop-A GOME-2 | | NP_GOME2A | RAL v2.14 |
| Metop-B GOME-2 | L2 | NP_GOME2B | RAL v2.14 |
| AURA OMI | | NP_OMI | RAL v2.14 |
| Metop-A IASI | | NP_IASIA | FORLI v20151001 |
| Metop-B IASI | | NP_IASIB | FORLI v20151001 |
| ERS2 GOME | | NP_L3_GOME | KNMI v0004 |
| Envisat SCIAMACHY | | NP_L3_SCIAMACHY | KNMI v0004 |
| Metop-A GOME2 | L3 | NP_L3_GOME2A | KNMI v0004 |
| AURA OMI | | NP_L3_OMI | KNMI v0004 |
| Metop-A IASI | | TTC_IASI | ULB/LATMOS v0001 |
| merged (GOME, GOME-2A) | L4 | NP_L4_ASSIM | KNMI v0004 |

**Table 1: Overview of the nadir ozone profile data products generated and delivered in the Ozone_cci CRDP.**

| CCI CRDP product ID | 95 | 96 | 97 | 98 | 99 | 00 | 01 | 02 | 03 | 04 | 05 | 06 | 07 | 08 | 09 | 10 | 11 | 12 | 13 | 14 | 15 |
|---|---|---|---|---|---|---|---|---|---|---|---|---|---|---|---|---|---|---|---|---|---|
| NP_GOME | | 26 | | | | | | | | | | | | | | | 25 | | | | |
| NP_SCIAMACHY | | | | | | | | 31 | | | | | | | | | 09 | | | | |
| NP_GOME2A | | | | | | | | | | | | | 14 | | | | | | 28 | | |
| NP_GOME2B | | | | | | | | | | | | | | | | | | | 18 | | 23 |
| NP_OMI | | | | | | | | | | 40 | | | | | | | | | | | 52 |
| NP_IASIA | | | | | | | | | | | | | 01 | | | | | | | | 52 |
| NP_IASIB | | | | | | | | | | | | | | | | | | | 10 | | 14 |
| NP_L3_GOME | | | | | | | | | | | | | | | | | | | | | |
| NP_L3_SCIAMACHY | | | | | | | | | | | | | | | | | | | | | |
| NP_L3_GOME2A | | | | | | | | | | | | | | | | | | | | | |
| NP_L3_OMI | | | | | | | | | | | | | | | | | | | | | |
| TTC_IASI | | | | | | | | | | | | | | | | | | | | | |
| NP_L4_ASSIM | | | | | | | | | | | | | | | | | | | | | |

**Table 2: Time coverage (up to 2015) of the nadir ozone profile data products generated and delivered in the Ozone_cci CRDP (numbers indicate start and end weeks for L2 data).**

## 2.2 L2 UV-VIS retrieval algorithm

Full timeseries of the ERS-2 GOME (1996-2011), Envisat SCIAMACHY (2002-2011), Metop-A GOME-2 (2007-2013), Metop-B GOME-2 (2013-2015), and AURA OMI (2004-2015) nadir ozone profile data were retrieved at the Rutherford Appleton Laboratory using version 2.14 of its RAL retrieval system. Each ozone profile is provided in volume-mixing ratio (VMR) and number density (ND) units on a fixed vertical grid with 20 levels ranging between 0 and 80 km, while the values of the 19 intermediate partial ozone column layers are provided as well. The RAL retrieval is a three-step process (Munro et al., 1998; Siddans, 2003; Miles et al., 2015).

In the first step, the vertical profile of ozone is retrieved from Sun-normalized radiances at selected wavelengths of the ozone Hartley band, in the range 265-307 nm, which primarily contains information on stratospheric ozone. Prior ozone profiles come from the McPeters-Labow-Logan (McPeters et al., 2007) climatology, except in the troposphere where a fixed value of



$10^{12}$ ozone molecules per cubic meter is assumed. A prior correlation length of 6 km is applied to construct the covariance matrix. The surface albedo, a scaling factor for the Ring effect, and the dark signal are retrieved jointly. In the second step, the surface albedo for each of the ground pixels is retrieved from the Sun-normalized radiance spectrum between 335 and 336 nm. Then, in step three, information on lower stratospheric and tropospheric ozone is added by exploiting the

temperature dependence of the spectral structure in the ozone Huggins bands. The wavelength range from 323 to 334 nm is used in conjunction with ECMWF ERA-Interim (ERA-I) meteorological fields (Dee et al., 2011). Each direct Sun spectrum is thereby fitted to a high-resolution (0.01 nm) solar reference spectrum to improve knowledge of wavelength registration and slit function width. In this step the a-priori ozone profile and its error are the output of step one, except that a prior correlation length of 8 km is imposed.

RAL's radiative transfer model (RTM) is derived from GOMETRAN (Rozanov et al., 1997), but the original code has been modified substantially in order to increase its efficiency without losing accuracy. Within the RTM there is no explicit representation of clouds, but their effects are incorporated as part of the Lambertian surface albedo (from step 2 of the retrieval). Therefore a negative bias in retrieved ozone is to be expected where high or thick cloud is extensive and there is limited photon penetration (no 'ghost column' is added). The linear error analysis of the RAL retrieval is additionally

complicated by the three-step retrieval approach. Particularly as the ozone prior covariance used in step three is not identical to the solution covariance output from step one. This is handled by linearizing each step and propagating the impact of perturbations in parameters affecting the measurements through to the final solution. The estimated standard deviation of the final retrieval is taken to be the square-root of the step-three solution covariance.

In this work, all nadir ozone profile screening of RAL retrievals follows the recommendations as outlined in the latest

version of RAL's Ozone Profile Algorithm Product User Guide (PUG). As summarised in Table 3, the filtering requires that the normalised cost function is less than two, the convergence flag equals one, all ozone profile values are positive, the solar zenith angle is below 80°, and the effective cloud fraction (ECF) below 20 %. Additionally, for GOME-2A and B the Band 1 slant column density has to stay below 500 DU, and the OMI outer two pixels from each swath are rejected (see product-specific criteria in Table 3). Back-scan measurements are never considered.





| Filtering criterion | UV-VIS RAL algorithm v2.14 | TIR FORLI algorithm v20151001 |
|---|---|---|
| Averaging kernel matrix | / | - DFS > 1<br>- All elements < 2<br>- First derivative < 0.5<br>- Second derivative < 1 |
| Chi-square test | 1 | 1 |
| Convergence | 1 | 1 |
| Cost function (normalised) | < 120 (< 2) | / |
| Effective cloud fraction | < 0.20 | < 0.13 |
| Negative ozone values | Rejected | Rejected |
| Product-specific | - GOME-2A/B: January-to-May band 1 SCD < 500 DU<br>- GOME-2B from June 2015<br>- OMI: outer two pixels from each swath rejected | - Ozone rejected if incomplete H2O retrieval<br>- IASI-B: March 8 to April 24, 2013 rejected (erroneous setting) and from April, 2015 |
| Solar zenith angle | < 80° | < 83° (day-time) or > 91° (night-time) |
| Surface pressure | Rejected if unrealistic | Rejected if unrealistic |
| Surface temperature | / | Rejected if unrealistic |
| Tropospheric ozone | / | Ratio of 6 km integrated column to total integrated column > 0.085 |

**Table 3: L2 nadir ozone profile filtering criteria applied in this work (first column) and their settings for the RAL UV-VIS retrieval algorithm (second column) and the FORLI TIR retrieval algorithm (third column). Values that do not comply with the settings are rejected as suggested by the respective data providers.**

### 2.3 L2 TIR retrieval algorithm

The Ozone_cci Metop-A and Metop-B IASI nadir ozone profile data for 2008-2015 and 2013-2015, respectively, were generated in a near real time mode using the FORLI-O3 (Fast Optimal Retrievals on Layers for IASI Ozone) latest version 20151001 (see Hurtmans et al. (2012) for a full description of the retrieval parameters and performances). FORLI-O3 relies on a fast radiative transfer and a retrieval algorithm based on the optimal estimation method (Rodgers, 2000). In the current version of FORLI-O3, look-up tables (LUTs) were precomputed to cover a larger spectral range (960-1105 cm$^{-1}$) using the

HITRAN 2012 spectroscopic database (Rothman et al., 2013) and correcting numerical implementation, especially with regard to the LUTs at higher altitude compared to the previous version. Ozone is retrieved using the 1025-1075 cm$^{-1}$ spectral range, which is dominated by ozone absorption with only few overlapping water vapour lines and a weak absorption contribution of methanol. The a priori information used in the FORLI algorithm consists of a single global ozone prior profile and a variance-covariance matrix built from the McPeters-Labow-Logan climatology (McPeters et al., 2007), as for

RAL. A purely diagonal wavenumber-dependent effective noise at a value around 2 10$^{-8}$ W/cm/sr is considered in the retrievals (Hurtmans et al., 2012).

The FORLI-O3 product consists of a vertical profile retrieved on a uniform and fixed 1 km vertical grid on 40 layers from the surface up to 40 km, with an extra residual layer from 40 km to the top of the atmosphere (60 km in practice). Associated



averaging kernels and relative total error profiles are provided on the same vertical grid. A posteriori filtering of the data – performed by ULB/LATMOS before data distribution – is applied to keep only the more reliable data, by removing those corresponding to poor spectral fits (root mean square of the spectral fit residual higher than 3.5 $10^{-8}$ W/cm/sr) or incomplete water vapour retrievals. Additionally, quality flags rejecting biased or sloped residuals, suspect averaging kernels, and

violations of the maximum number of iterations are applied (see Table 3). Cloud contaminated IASI scenes characterized by a fractional cloud cover above 13 % are also filtered out, as identified using cloud information from the EUMETCast operational processing (August et al., 2012). Upon discussion within the Ozone_cci community, it has been decided to in this work also reject FORLI ozone profiles whose ratio of the 0-6 km integrated column to the fully integrated column exceed 0.085. These provisional fixes however are corrected for in the online Ozone_cci nadir ozone profile product release.

**2.4 L3 monthly gridded data**

For the thermal infrared IASI instrument on Metop-A, a tropospheric level-3 (L3) product (prefix TTC instead of NP in Table 1) has been generated by the ULB/LATMOS team from their quality-screened L2 nadir ozone profile retrievals directly. This product consists of horizontally gridded (1° latitude by 1° longitude) monthly averages of the zero to six kilometre vertically integrated IASI-A ozone observations.

Monthly-averaged L3 profile products are produced from the filtered RAL v2.14 GOME, GOME-2A, SCIAMACHY, and OMI data by the Royal Meteorological Institute of the Netherlands (KNMI). Version 0004 of the KNMI L3 products has been used in this work (see Table 1). The KNMI level-3 data consist of monthly ozone profile averages, also on a one-by-one degree latitude-longitude grid, containing 19 layers between 20 fixed pressure levels at each grid-point. The algorithm that calculates the monthly-averaged ozone fields assumes that the L2 satellite ground pixel vertices are ordered as indicated

in Figure 1. Each pixel's across-track direction is defined by the lines the lines AD and BC, while the along-track direction is defined by the lines AB and DC (note that corners C and D are reversed with respect to the GOME/GOME-2 convention). The along-track pixel edges and cross-track pixel edges are divided into a number of points. The first point on AB and the first on DC determine a line which is divided into the same number of points as AD. Each of these points is then assigned to a grid cell. Suppose that the horizontal line of diamonds in Figure 1 represents the ground subpixels (numbered 1 to 7) and

the two dashed lines denote the grid cell boundaries which are numbered the same way as the ground pixel corners (i.e. grid cell A is the lower right cell). In that case, subpixels 1 to 3 are added to grid cell A, and the counter for grid cell A is increased by 3. Subpixels 4 to 7 are added to grid cell D and the counter for grid cell D is increased by 4. The pixel values $x_i$ are weighted by the square inverse of their uncertainties ($\sigma_i^{-2}$) before adding, so the weighted mean grid cell value $x_c$ and the corresponding standard deviation $\sigma_c$ are given by

$$x_c = \sigma_c^2 \sum_i x_i \sigma_i^{-2} \qquad (1)$$

and

$$\sigma_c = \left( \sum_i \sigma_i^{-2} \right)^{-\frac{1}{2}} \qquad (2)$$



respectively.

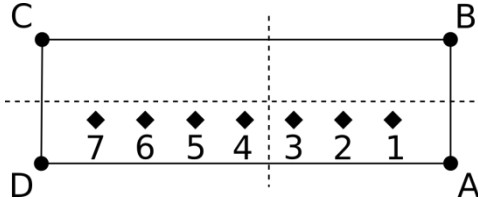

**Figure 1: A L2 ground pixel ABCD is divided into ground subpixels (diamonds 1 to 7). Each subpixel is assigned to a TM5 assimilation grid cell (dashed) and the average and standard deviation are calculated (see text).**

### 2.5 L4 data assimilation

Assimilated level-4 (L4) ozone fields are produced from the screened Ozone_cci UV-VIS nadir ozone profile data by the Royal Meteorological Institute of the Netherlands (KNMI) by use of its chemical transport model TM5. The resulting L4 assimilated fields consist of 44 ozone layers on a two-by-three degree latitude-longitude grid for four times a day (0, 6, 12, 18 h). Version 0004 of the L4 products has been used in this work, meaning that the assimilation input is limited to the L2 GOME (Jan. 1, 1996 to May 31, 2011) and GOME-2A (May 1, 2007 to June 30, 2013) products (NP_GOME and NP_GOME2A in Table 1).

A complete description of KNMI's assimilation algorithm can be found in Van Peet et al. (2017). The covariance matrices and the averaging kernel matrices from the L2 optimal estimation retrievals are thereby used. For the atmospheric model, the covariance matrix must be specified as well. The observations and the model data are combined using a Kalman filter technique. The averaging kernel matrix is incorporated into the observation operator and the observation and model covariance matrices are used in the Kalman equations to calculate the analysis fields. In order to reduce biases between multiple instruments, an ozonesonde-based bias correction has been developed. For this correction, only sondes collocated with cloud free retrievals (i.e. cloud fraction < 0.2) have been used. This correction is applied to the L2 data before the assimilation, meaning that the ozonesonde measurements involved (from 64 stations) cannot be used for the Ozone_cci L4 comparative validation exercise (see Section 5.6) as FRM used for comparisons have to be independent of the validated product.

### 3 Validation approach and reference data

### 3.1 Quality assessment of atmospheric satellite data

This work adopts the exhaustive seven-step satellite data quality assessment approach presented in Keppens et al., 2015, as schematised in its Appendix A. This approach includes (1) satellite data collection and post-processing, (2) dataset content study, (3) information content study, (4) FRM data selection, (5) co-located datasets study, (6) data harmonisation, and (7) comparative analyses including dependences on physical influence quantities of relevance. The satellite data collection and




post-processing (mainly L2 profile screening) is described by the previous section. The resulting satellite data content studies and information content studies are discussed in the next Section 4. These include statistics on the L2 data screening and spatiotemporal coverage, and averaging kernel-based information content measures, respectively. The FRM data selection, co-located datasets study, and data harmonisation on the other hand are included as the successive subsections within this section. The comparative analysis follows later in Section 5.

**3.2 Ground-based reference data selection**

Ground-based data records from the well-established Network for the Detection of Atmospheric Composition Change (NDACC), Southern Hemisphere Additional Ozonesonde programme (SHADOZ), and other ozonesonde and lidar stations contributing to the World Meteorological Organisation's Global Atmosphere Watch (WMO GAW) ozonesonde and lidar networks are used as a transfer standard against which the nadir ozone profile retrievals are compared. Like for the satellite data, and prior to searching for co-locations with satellite ECV data, data screening has been applied to the FRM. The recommendations of the ground-based data providers to discard unreliable measurements are thereby followed, both on entire profiles and on individual vertical levels. Measurements with unrealistic pressure, temperature, or ozone readings are rejected automatically. Ozonesonde measurements at pressures below 5 hPa (beyond 30-33 km) and lidar measurements outside of the 15-47 km vertical range are rejected as well. The raw ozonesonde profiles retrieved from the public NDACC, SHADOZ data archives and World Ozone and UV Data Centre (WOUDC) are moreover quality-screened according to the criteria outlined in Hubert et al. (2016) for a similar analysis on space-borne limb observations of atmospheric ozone. The resulting spatio-temporal distribution of ground-based observations is summarised in Figure 2. Despite the higher concentration of FRM in the northern mid-latitudes (20-60°) and before 2014, the distribution is sufficiently homogeneous to consider global comparison statistics and to enable drift assessments.

The uncertainties related to the sonde and lidar FRM used in this work are discussed in Keppens et al. (2015) and Hubert et al. (2016). Essentially, ozonesondes measure the vertical profile of ozone partial pressure with order of 10m vertical sampling (100–150m actual vertical resolution) from the ground up to the burst point of the balloon, usually between 30 and 33 km. Their estimated bias is smaller than 5 %, and the precision remains within the order of 3%. Above 28 km the bias increases for all sonde types. Below the tropopause, due to lower ozone concentrations, the precision decreases slightly to 3-5 %, depending on the sonde type. The tropospheric bias also becomes larger, between 5 and 7 %. Stratospheric ozone lidar systems are sensitive from the tropopause up to about 45-50 km altitude with a vertical resolution that declines with altitude from 0.3 to 3-5 km. The estimated bias and precision are about 2% between 20 and 35 km and increase to 10% outside this altitude range where the signal-to-noise ratio is smaller.





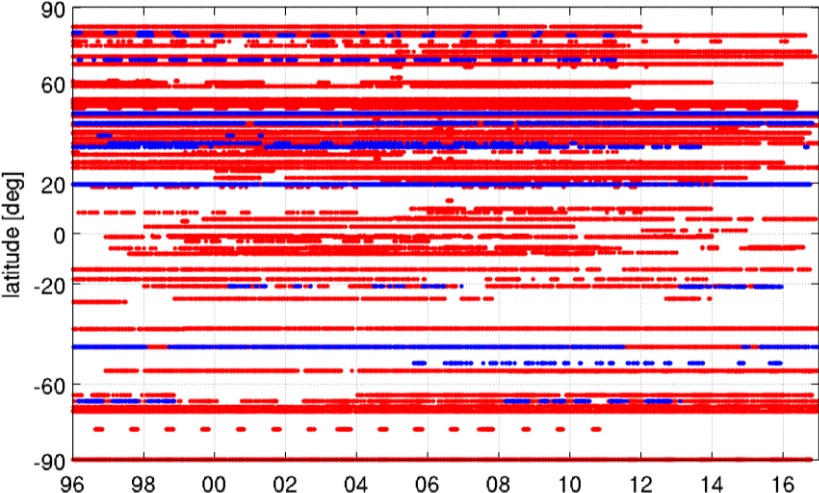

**Figure 2: Latitude-time sampling (1996-2016) of the ground-based ozonesonde (red dots) and stratospheric ozone lidar (blue dots) measurements, obtained from the NDACC, SHADOZ, and WOUDC reference network databases.**

### 3.3 Co-location and harmonisation of satellite and reference data

From all quality-approved L2 nadir ozone profile data, only those that are located within a certain radius of an NDACC, SHADOZ, or GAW ozonesonde or stratospheric lidar station location are retained for further analysis. This radius is adapted to the ground pixel size of each spaceborne instrument, in such a way that the ground-based station is roughly located within the satellite pixel (see Table 4). Additionally, only co-locations with a maximal time difference of 6 hours for ozonesondes and 12 hours for lidars are allowed. These time windows are chosen to generally have at least one satellite co-location with

each FRM, given the satellite's local solar time (LST) and the fact that ozonesondes are typically launched around local noon, while lidar measurements are taken during the night. When multiple L2 satellite pixel co-locations with one unique ground-based measurement occur, only the closest satellite measurement is kept. For the L3 and L4 nadir ozone profile data, only the grid cell that overlaps with the ground-based station location is considered. All FRM within this grid cell within one month are included in the analyses for the L3 comparisons. For the six-hourly assimilated L4 data, the unique temporally

closest ground-based reference measurement is always less than 3 hours away.

Calculating difference profiles also requires harmonisation of the satellite and reference ozone profiles in terms of at least their unit representation and vertical sampling (Keppens et al., 2015). While ozonesondes report measurements in partial pressure, easily converted into volume-mixing-ratio (VMR) units and also in number-density (ND) using the on-board PTU measurements, the lidar data are given in number density and in general the files do not provide associated temperature

profiles for a beforehand ND to VMR conversion. The latter has therefore been accomplished by consistently applying pressure and temperature fields that were extracted from the latest ERA-Interim reanalysis. Moreover, if there is no GPS altitude data in the ozonesonde data files, the altitude scale is reconstructed via the hydrostatic equation from the pressure and temperature recordings by the radiosonde attached to the ozonesonde. The number density profiles are integrated to





partial column profiles by use of these corresponding altitude grids. The partial column profiles are then converted to the fixed satellite vertical grids by use of mass-conserved regridding, meaning that the integrated ozone column between the outer vertical edges is conserved (Langerock et al., 2015).

The optimal estimation method used in the RAL and FORLI retrieval systems consists in minimizing the difference between

the measured atmospheric spectra and spectra simulated by a radiative transfer code (forward model). Since the retrieval is performed at higher vertical sampling than the actual amount of independent pieces of profile information available from the measurement, the retrieval is in general under-constrained and consequently unstable. Retrieval schemes therefore include additional constraints, e.g. in the form of a-priori information on the profile, its shape and its allowed covariance. As a result, the retrieved quantity is a mix of information contributed by the measurement and of a-priori information, as represented in

its vertically correlated averaging kernels. The satellite Level-2 and ground-based profiles' vertical smoothing is in this work by default harmonised (i.e. reducing the vertical smoothing difference error) by averaging kernel smoothing of the FRM (Keppens et al., 2015). The mass-conservation regridded ground-based profile $x_g$ is thereby converted into its vertically smoothed form $x_g'$ by multiplication with the satellite profile's averaging kernel matrix $A$ (in partial column units), yet taking into account the kernel's sensitivity to the prior profile $x_p$ of the optimal estimation retrieval:

$$x_g' = Ax_g + (I - A)x_p \qquad\qquad\qquad\qquad\qquad\qquad (3)$$

The reference profile hence becomes a vertically smoothed combination of the ground-based measurement (by multiplication with $A$) and the prior profile (by multiplication with $I - A$, with $I$ being the unit matrix of dimensions $A$) (Rodgers, 2000).

| CCI CRDP product ID | LST | SPI (#/scan) | pixel size | Co-loc. | Period |
|---|---|---|---|---|---|
| NP_GOME | 10:30A | 0:1:2 (3) | 320 x 40 km² | 100 km | 1996-2010 |
| NP_SCIAMACHY | 10:00A | 1:1:4 (4) | 240 x 32 km² | 100 km | 2003-2010 |
| NP_GOME2A | 09:30A | 0:2:22 (12) | 160 x 160 km² | 100 km | 2008-2012 |
| NP_GOME2B | 09:30A | 0:2:22 (12) | 160 x 160 km² | 100 km | 2013-2015* |
| NP_OMI | 01:30P | 9:4:49 (11/15) | 52 x 48 km² | 50 km | 2005-2015 |
| NP_IASIA | 09:30A(+P) | 0:2:118 (60) | 12 km (diam.) | 10 km | 2008-2015 |
| NP_IASIB | 09:30A(+P) | 0:1:119 (120) | 12 km (diam.) | 10 km | 2013-2015* |
| NP_L3_GOME | / | / | 1° x 1° | Overlap | 1996-2010 |
| NP_L3_SCIAMACHY | / | / | 1° x 1° | Overlap | 2003-2010 |
| NP_L3_GOME2A | / | / | 1° x 1° | Overlap | 2008-2012 |
| NP_L3_OMI | / | / | 1° x 1° | Overlap | 2005-2015 |
| TTC_IASI | / | / | 1° x 1° | Overlap | 2008-2012 |
| NP_L4_ASSIM | / | / | 2° x 3° | Overlap | 1996-2012 |

**Table 4: Local solar time (LST), possible scan pixel indices (SPI, with number of pixels per scan between brackets), ground pixel size, co-location distance, and temporal range of the comparative analysis. The asterisk with the Metop-B instruments indicates that the corresponding time series are not sufficiently long for drift studies. Next to the spatial co-location, a selection of the closest satellite measurement in time within 6 h for ozonesondes and 12 h for lidars takes place.**



## 4 Nadir ozone profile retrieval content

### 4.1 Data content

The nadir ozone profile CRDP L2 data content study focuses on the spatiotemporal distribution of the retrieved satellite profiles in the first place, next to the regular file structure, file content, and value checks for the quantities of highest

relevance (also see Table 3). Figure 3 displays the latitude-time distribution per 10° latitude band and per month of the relative amount of screened profiles for all nadir profile L2 datasets (except for IASI on Metop-B). The data that are screened fail the filtering criteria suggested to data users as described in Table 3 and are therefore omitted from further analysis. Where the screening goes from 0 % (all data passes, in blue) to 100 % (no data passes, in red), one could equally insightfully interpret the plots as showing the spatiotemporal coverage of the satellite data ranging between 100 % (full

coverage, in blue) and 0 % (no coverage left, in red), respectively.

The screening for the GOME and SCIAMACHY instrument retrievals is quite high (60-80 % on average), mainly due to the cloud screening that rejects all effective cloud fractions above 20 %. The ECF has less impact on the GOME-2 and OMI instruments, but the solar-zenith angle screening (if higher than 80°) still causes meridian and seasonal coverage variations. Moreover a latitudinal striping can be observed for all UV-VIS instrument distributions, although this is partially due to the

satellite pixel co-adding before retrieval and the station overpass data selection afterwards. The decreased GOME-2B availability from June 2015 onwards points at a retrieval issue and justifies additional screening, as shown in Table 3. The IASI screening on the other hand appears very low, yet this is due to the pre-screening by the product providers before data delivery. E.g. the IASI cloud screening (if the fraction is higher than 13 %) cannot be observed from the plots. Only the seasonality of the tropospheric ozone screening (ratio of the 6 km integrated column to total integrated column > 0.085)

becomes clear near the Antarctic. The IASI-B availability is fully similar to IASI-A (and overlapping in time) and therefore not shown.



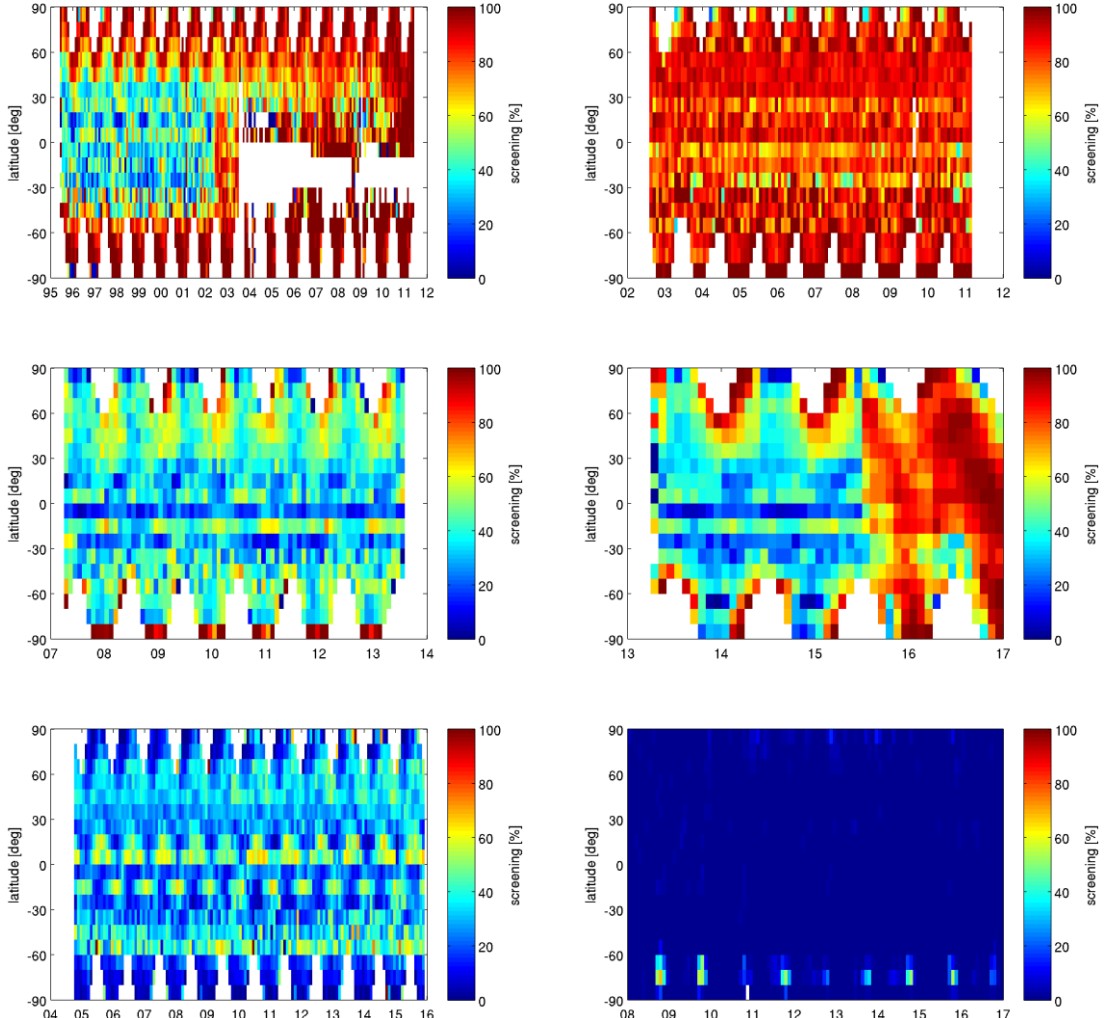

**Figure 3: GOME, SCIAMACHY, GOME-2A, GOME-2B, OMI, and IASI-A (left to right and top to bottom) latitude-time distribution of relative data screening, taking into account the quality flags presented in Table 3. The decreased GOME-2B availability from June 2015 onwards justifies additional screening. IASI-B is fully similar to IASI-A.**

## 4.2 Information content

### 4.2.1 Information quantities

Each quantity that is retrieved using the optimal estimation technique contains information both from the satellite measurement and from the a-priori profile and covariance matrix. The contribution of prior information can be significant where the measurement is weakly or even not sensitive to the atmospheric ozone profile, e.g. in case of fine scale structures of the profile, below optically thick tropospheric clouds, and at the lower altitudes. The information distribution is captured



by the retrieval's ex-ante vertical averaging kernel matrix $A$ (sometimes also AKM hereafter), which represents the sensitivity of the retrieved state $\hat{x}$ to changes in the true profile $x_t$ at a given altitude:

$$A(m,n) = \partial\hat{x}(m)/\partial x_t(n) \tag{4}$$

A study of the algebraic properties of this averaging kernel matrix, denoted information content study, can help understanding how the system captures actual atmospheric signals. Through straightforward analysis however, it can be easily demonstrated that typical information content measures as discussed in this section usually depend on the units of the averaging kernel matrices they are calculated from (Keppens et al., 2015). As these measures however should be unit-independent, fractional AKMs $A_F$ must be considered.

From Eq. (4), the fractional AKM is calculated by dividing the nominator and denominator by the corresponding retrieved and true ozone profile value, respectively. As the true profile however is not known, it is replaced by its best available estimate $\langle x_t \rangle$ being again the retrieved profile:

$$A_F^{RAL}(m,n) = A(m,n)x_t(n)\hat{x}^{-1}(m) \approx A(m,n)\langle x_t(n)\rangle\hat{x}^{-1}(m) = A(m,n)\hat{x}(n)\hat{x}^{-1}(m) \tag{5}$$

This approach is directly used for determining the fractional averaging kernel matrices in the UV-VIS RAL v2.14 retrieval products, wherefrom the RAL superscript. The FORLI v20151001 algorithm that performs the thermal infrared retrievals however, performs a unit-independent optimal estimation that immediately yields fractional AKMs. These fractional matrices are made unit-dependent by use of the prior profile before saving into the data files, allowing for more straightforward application (e.g. for vertical smoothing operations) by data users. For the information content studies presented here, this defractionalisation operation therefore has to be inverted:

$$A_F^{FORLI}(m,n) = A(m,n)x_p(n)x_p^{-1}(m) \tag{6}$$

Hereafter, starting from the averaging kernels provided as part of the Ozone_cci CRDP L2 nadir ozone profile products, the degree of freedom in the signal (DFS) and the vertical sensitivity are studied. These quantities are given by the AKM trace and row sum profile, respectively. The DFS of a retrieved atmospheric profile is a non-linear measure for the number of independent quantities that can be determined and as such loosely related to the Shannon information content (Rodgers, 2000). The vertical sensitivity to the measurement is a unit-normalised measure for how sensitive the retrieved ozone value at a certain height is to ozone values at all heights. According to Rodgers (2000) p. 47, measurement sensitivity "can be thought of as a rough measure of the fraction of the retrieval that comes from the data, rather than from the a-priori".

Besides the more common DFS and sensitivity information content quantities, in this work the vertical averaging kernels' offset and width are considered as well. The offset is an estimate of the uncertainty on the retrieval height registration, given either by the direct vertical distance (in km) between an averaging kernel's peak sensitivity altitude $z_{peak}$ and its nominal retrieval altitude $z_{nom}$ as $d(m) = z_{peak}(m) - z_{nom}(m)$ or as the so-called centroid offset $d_c(m) = c(m) - z_{nom}(m)$ (Rodgers, 2000) with

$$c(m) = \left(\sum_n z(n)A_F^2(m,n)\Delta z(n)\right)\left(\sum_n A_F^2(m,n)\Delta z(n)\right)^{-1} \tag{7}$$

Ideally, within each kernel, this distance equals zero.



Ozone_cci user requirements also specify an upper limit of the vertical resolution of the nadir ozone profile retrievals. In the literature different methods have been proposed to estimate the vertical resolution from the width of the vertical averaging kernels (see overview in Keppens et al., 2015), but usually it is determined either as a full width at half-maximum (FWHM) value around the kernel's peak altitude or as the Backus-Gilbert spread (BG) or resolving length around its centroid:

$$w_{BG}(m) = 12(\sum_n [c(m) - z(n)]^2 A_F^2(m,n)\Delta z(n))(\sum_n A_F^2(m,n)\Delta z(n))^{-2} \qquad (8)$$

Whereas an averaging kernel's direct offset and FWHM width only take into account its central sensitivity peak, Eqs. (7) and (8) point out that the centroid offset and BG-spread include all vertical kernel information. As a result, the centroid at a given altitude can be considered a measure of the overall retrieval barycentre for that altitude, with the Backus-Gilbert spread showing the retrieval's full extent, also taking into account sensitivity fluctuations. Other information content diagnostics, such as the measurement quality quantifier (MQQ) and the AKMs' eigenvectors and eigenvalues, have previously been studied but are not reported here (Keppens et al., 2015).

### 4.2.2 Degrees of freedom in the signal

Figure 4 displays the latitude-time distribution per 10° latitude band and per month of the median DFS for all nadir profile L2 datasets (except for IASI on Metop-B). RAL's UV-VIS DFS is typically around 5, with the lowest values for SCIAMACHY (4 to 5) and the highest for OMI (5 to 5.5), and quite stable in time, reflecting the signal degradation correction that is incorporated within the RAL v2.14 retrieval algorithm. Seasonal DFS variations amount to about 0.5, which is approximately the same as the DFS decrease per decade, except for the more stable OMI retrieval. The temporal DFS behaviour is also reflected in the AKMs' eigenvalues and eigenvectors (not included). More exceptional are the 2 to 3 DFS outliers for SCIAMACHY, which typically occur in the South-Atlantic Anomaly (SAA). Such SAA outliers also occur in other instrument retrievals, but to a lesser extent (also see next sections). Also note that the decreased retrieval performance for GOME-2B from June 2015 (eventually resulting in its total screening) actually has little effect on its DFS behaviour. Due to its stronger meridian and seasonal dependence, the FORLI TIR median DFS for IASI-A ranges between 2 towards the poles and 4 towards the equator. The overall degradation however is negligible as for OMI. The IASI-B spatiotemporal DFS behaviour is fully similar to IASI-A (and overlapping in time) and therefore not shown.



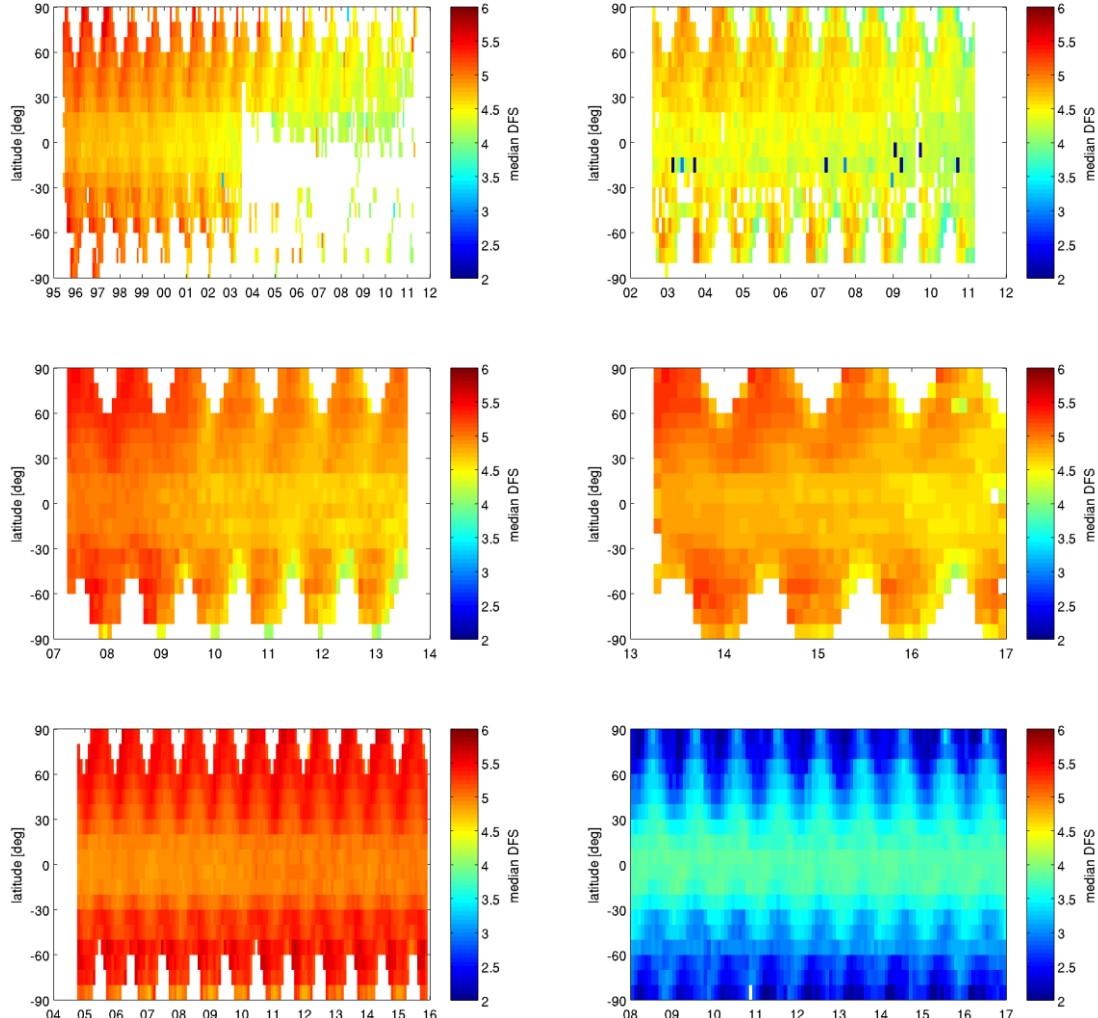

**Figure 4: GOME, SCIAMACHY, GOME-2A, GOME-2B, OMI, and IASI-A (left to right and top to bottom) latitude-time distribution of degrees of freedom in the signal (DFS). IASI-B is fully similar to IASI-A.**

### 4.2.3 Height-resolved information content

Exemplary plots containing the global GOME-2A (left column) and IASI-A (right column) information content in terms of vertical sensitivity, retrieval offset, and averaging kernel width are displayed in Figure 5. Their dependence on DFS, solar zenith angle (SZA), or thermal contrast (TC) is introduced by the plot colour, whereby profiles corresponding to out-of-range (OOR) influence quantity values are plotted in magenta. The other RAL v2.14 UV-VIS and FORLI v20151001 TIR retrieval products show similar statistics, respectively.

The vertical sensitivity profiles, which are the same in all three plots for each product, are close to unity around the ozone peak and above (25 to 45 km) for all retrieval products under consideration. Typically the sensitivity decreases above and





below due to the smaller ozone concentrations (therefore the vertical range is limited to 50 km), but the actual behaviour strongly depends on the retrieval algorithm. The RAL retrieval usually results in a very strong over-sensitivity around the upper troposphere and lower stratosphere (UTLS), with a median value of 3. This peak partially compensates for the under-sensitivity right above and below, with the sensitivity dropping down to about 0.5 in the lowest 0-6 km column. The peak
value moreover heavily correlates with the SZA, as one can expect for an UV-VIS retrieval algorithm. On the other hand, some RAL sensitivity profiles quickly decrease to zero when going from 25 to 40 km altitude. These are connected to very low DFS values (around two or below), as identified to occur around the SAA. Most of the retrieval information in these profiles is therefore located around the UTLS and in the troposphere.

The IASI instrument retrievals do not show this stratospheric decline for excessively low DFS values, but instead show
sensitivity outliers around the UTLS, ranging from below -1 to above 2. Although the overall IASI sensitivity variability is strongest around the equator, these outliers typically occur in the polar regions, as can be expected from Figure 4, and go together with excessively high retrieved ozone peaks. The strong sensitivity variability in general hampers the averaging kernel smoothing of the reference profiles before comparison (see Eq. (3)), as this procedure then introduces a bias instead of reducing the vertical smoothing difference error. Usually however, except for decreased surface-level sensitivity (0.5) and a
median 1.5 peak around the UTLS with slight compensation above and below, the FORLI v20151001 sensitivity is more vertically consistent.

Also according to Figure 5, little difference can be observed between the median UV-VIS retrieval offset in terms of its direct and centroid measures. The height registration uncertainty remains below 10 km (except again for the low DFS values), being negative in the upper stratosphere and positive towards the Earth's surface, as can be expected for any nadir
ozone profile retrieval. Note however that the direct offset is more discrete than the Backus-Gilbert spread due to its one-to-one connection with the vertical retrieval grid steps. This discreteness of the direct offset is even clearer for the FORLI IASI retrievals that are performed on a fixed 1 km vertical grid. The direct offset here is lower than the centroid offset on average, but amplifies some of the latter's features, like the peak and jump around 5 and 25 km altitude, respectively. The FORLI IASI height registration uncertainty in terms of the centroid offset steadily increases from zero at 40 km to about 30 km near
the surface, meaning that the retrieval barycentre altitude is decreasing slower than the nominal retrieval altitude. The dependence on DFS and thermal contrast however is rather small.

The behaviour of an averaging kernel's sensitivity and offset is typically also reflected in its width. Figure 5 demonstrates that the RAL retrieval's sensitivity peak in the UTLS goes together with a strongly increasing Backus-Gilbert spread, exceeding 60 km towards the Earth's surface. The median FWHM width staying below 15 km indicates that the high BG-
spread values are due to fluctuations in the averaging kernels of the retrieval, showing several highs and lows next to the peak value. At higher altitudes, the median kernel width decreases first to about 20 km, and further to 10 km in the upper stratosphere, although individual results strongly depend on the SZA. From the low up to the middle latitudes the resolving length shows little seasonal variation, but from the mid-latitudes to the polar areas an annual variation indeed appears clearly from the ground up to the lower stratosphere, with maxima in winter and minima in summer (not shown). This conduct



correlates directly with the annual variation of the slant column density (highest in winter and lowest in summer) and thus of the sensitivity.

The connection between averaging kernel offset and width is even stronger for FORLI's v20151001 TIR retrieval scheme. At 25 km and below, where the offset shows fluctuations, the Backus-Gilbert spread is strongly variable and its median explodes, although acceptable values of the order of 15 km are found above 25 km altitude. As for the RAL retrieval scheme, the median FWHM width staying around 10 km overall indicates that the high BG-spread values are not due to the presence of a single broad sensitivity peak, but rather to strong fluctuations in the averaging kernels that are again little dependent on DFS or thermal contrast. Like already observed for the IASI vertical sensitivity, the strongest averaging kernel width variability occurs in the tropics.



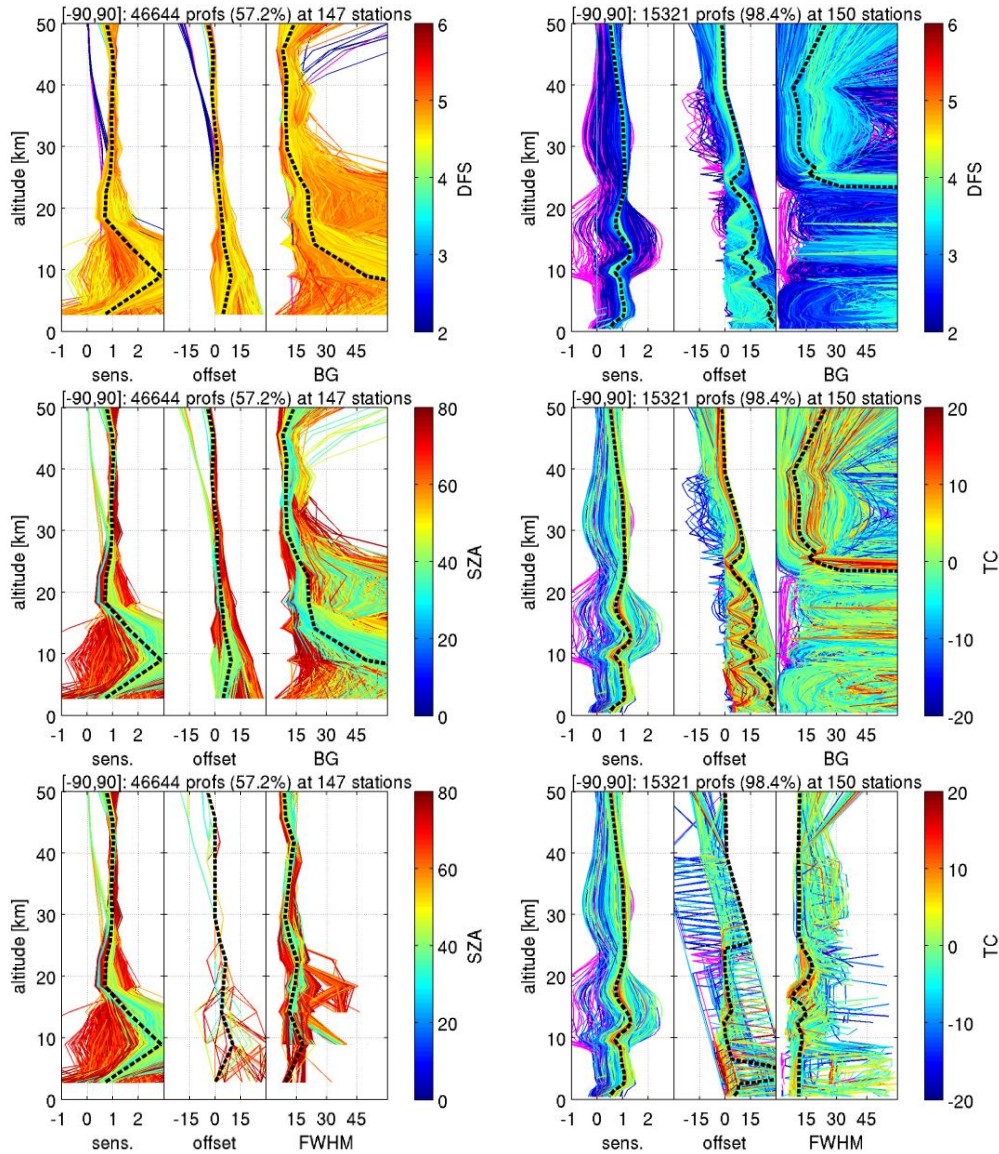

**Figure 5: Global GOME-2A (left) and IASI-A (right) information content in terms of vertical sensitivity, retrieval offset (in km),**
**and averaging kernel width (in km) and their dependence on DFS, SZA, or thermal contrast (TC). Out-of-range profiles are**
**plotted in magenta. Different measures are used for the offset and kernel width in the second and third rows. Plot titles provide the**
**absolute and relative amounts of profiles after screening, and the number of ground-based overpass stations.**

## 5. Ground-based comparisons

### 5.1 Comparison statistics

10  The baseline output of the L2 validation exercises consists of median absolute and relative nadir ozone profile differences at

individual stations or within latitude bands for the entire time series. This median difference is a robust (against outliers)



estimator of the vertically dependent systematic error, i.e. the bias, of the satellite data product. The bias profiles for the entire list of stations are then combined and visualized as a function of several influence quantities in order to reveal any dependences of the systematic error. The influence quantities considered in this work are latitude (meridian dependence), quarter, being December-January-February (DJF), March-April-May (MAM), June-July-August (JJA), and September-

October-November (SON), total ozone column (TOC), DFS, SZA, scan pixel index, (effective) cloud fraction (for the UV-VIS products), thermal contrast (for the TIR products), and time. The latter actually results in drift studies, i.e. the annual or decadal bias change of the satellite product with respect to the ground-based reference time series.

Besides the median difference, also the Q84-Q16 interpercentile spread (IP68) on the differences is calculated as a robust estimator of the random errors in the satellite data product, i.e. the precision profile. However, this spread on the differences

will also include contributions from ground-based random uncertainties (limited to a few percent, as indicated in Section 3.2) and representativeness (sampling and smoothing) differences between the satellite and reference measurements, and therefore in fact provides and upper limit on the actual satellite uncertainty. In case of a normal distribution of the ozone differences, median and IP68 are equivalent to mean and standard deviation, but they offer the advantage to be much less sensitive to occasional outliers.

The long-term stability of the systematic errors in the ozone data products is a key user requirement. Robust linear regressions including an uncertainty estimate based on a bootstrapping approach (Hubert et al., 2016) are performed on the satellite-ground difference profiles for all stations within the predefined latitude bands or on the global scale. The uncertainty on the global drift that is as such introduced by inhomogeneities across the ground-based network is of the order of about 5 % per decade, but in fact partially covered by the confidence interval obtained by the bootstrapping. This value was

estimated from the standard deviation on the ensemble of single-station drift estimates in ground-based comparisons with limb sounding instruments by Hubert et al. (2016), who use the same quality-checked selection of FRM stations. To avoid spurious effects due to a seasonal cycle in the differences, only time series of fiver years or longer are used for this drift assessment. Moreover, only fully available years of the satellite datasets have been considered for comparative analysis in order not to introduce seasonal effects at the beginning and the end of each time series. This is with the exception of the

Metop-B GOME-2 and IASI instruments however, that have not been used for drift studies (indicated with an asterisk in Table 4).

Due to its six-hourly assimilated content, the L4 comparative validation approach is fully similar to the L2 statistics described above. The strongly reduced amount of parameters in the L4 data product files however, reduces the number of influence quantity dependences that can be studied. These have therefore been limited to the latitude and quarter. Next to

that, as vertical averaging kernel matrixes are only available for the L2 retrieved data, no averaging kernel smoothing can be applied before comparison. Yet as mentioned in Section 2.5, the L2 averaging kernel matrices are incorporated into the equations to calculate the analysis fields. Also remember that the satellite instrument bias correction by use of ozonesonde measurements, the 64 stations involved are not used for the L4 comparative validation exercise.


The situation is quite different for the validation statistics of the L3 monthly gridded averages. No L2 averaging kernels are used for the data generation and no merging or bias correction are implemented. The satellite-based and one-by-one degree gridded nadir profile level-3 data $x_s^{L3}$ can be compared with spatially co-located ground-based reference profiles $x_r$ directly, or with monthly (gridded) averages $\langle x_r \rangle$ of the latter (i.e. a ground-based level-3-type dataset). Yet both approaches introduce spatial and temporal representativeness errors into the difference statistics that upon taking (monthly) averages as a bias estimator $\langle \Delta x \rangle$ yield comparable outcomes:

$$\langle \Delta x \rangle = N_m^{-1}\left[\left(x_s^{L3} - x_{r,1}\right) + \left(x_s^{L3} - x_{r,2}\right) + \cdots + \left(x_s^{L3} - x_{r,m}\right)\right] = x_s^{L3} - \langle x_r \rangle \tag{9}$$

For sufficiently fine-gridded L3 data, the comparisons can therefore be limited to direct differences with ground-based reference measurements, if one additionally only considers ground-based stations with a sufficient number $N_m$ of valid measurements per month. This number has been set to six (per month, or about at least one measurement each five days) in the L3 validation presented in this work. As such, an implicit averaging of at least six ozonesonde or lidar measurements per month is introduced in the comparison statistics. The one-by-one degree box that overlaps with the ground measurements is thereby taken as the co-located measurement. Thanks to this high horizontal resolution of the Ozone_cci L3 satellite nadir ozone profile products and the constraint on the temporal representativeness of the ground-based data, representativeness errors are thus kept to a minimum.

## 5.2 L2 UV-VIS nadir ozone profiles

In this section comparison results between L2 RAL v2.14 nadir ozone profiles and ground-based ozonesonde and lidar measurements are reported in the form of statistics on the median relative difference (bias) and 68 % interpercentile spread of ozone differences as a function of several influence quantities. Figure 6 to Figure 10 contain the results for GOME, SCIAMACHY, GOME-2A, GOME-2B, and OMI, respectively, as a function of latitude, quarter, total ozone column, DFS, SZA, scan pixel index, and effective cloud fraction. Note that the number of comparisons (shown in each plot title) is higher for the latter as the ECF filter has been switched off. Estimates of the relative satellite errors provided with the RAL v2.14 products have been added to the graphs (grey lines), in order to discuss them with respect to the ozone differences and spreads. In each plot the third subgraph displays the median sensitivity of the retrieved ozone profile as a function of altitude (and the relevant influence quantity), as calculated from the fractional RAL v2.14 vertical averaging kernels.

Before discussing the comparison results in terms of influence quantities, it is interesting to note that the vertical averaging smoothing of the ground-based reference data mostly yields qualitatively similar bias and spread estimates as when merely the regridded data are considered (not included). The comparisons from regridded reference data however show a vertically oscillating structure (as smoothing difference error) that largely disappears for the kernel smoothed comparisons. This structure is strongest around the Tropics, yielding significant differences between the regridded and smoothed data, mostly due to a positive bias peak just below 20 km for the regridded data. The corresponding comparison spreads indicate that the random uncertainty on the bias is reduced by about 10 % on average by applying the averaging kernel smoothing. This value



provides a rough estimate of the vertical smoothing difference error between the ground-based reference data on the one hand and the satellite data on the other hand.

Focussing on the comparisons involving averaging kernel smoothed partial column profiles, one observes that generally the five RAL v2.14 UV-VIS retrieval products agree similarly with the ground-based data, showing a rather typical Z-curve with zero biases approximately at 5 and 25 km altitude (the third around 55 km is not on the plots because of the sparseness of the FRM data availability above 50 km). The negative bias peak in the UTLS and above (5 to 25 km) and the positive bias peak in the upper stratosphere (between 25 and 55 km) both amount to about 20 to 40 %. Comparison results for the 0-6 km subcolumn show that the bias again shifts towards 40 % positive values in that layer, with the exception of the OMI instrument that keeps its median tropospheric bias within 10 %. The sensitivity for this lowest layer however is reduced to about 0.5, meaning that generally about 50 % of the retrieval information comes from the prior profile rather than from the measurement. In the 0 to 45 km altitude range, the UV-VIS nadir ozone profile comparison uncertainties in terms of the 68 % interpercentile spread display a U-shaped curve with a minimum of about 10 % around 25 km. The uncertainty increases to roughly 40 % at 45 km, to slightly decrease again above, but rises even more strongly where the sensitivity profile peaks and towards the ground.

The individual L2 UV-VIS comparison graphs also contain information on the validity of ex-ante uncertainties provided for the satellite nadir ozone profile retrievals (thin grey lines). The relative error reported in the RAL v2.14 data files amounts to about 5 % at the altitude of the ozone maximum, up to about 10 % at higher altitudes, and up to 40 % in the lower troposphere. In theory the IP68 spread should be close to the combined uncertainty of the satellite data, the ground-based data, and metrology errors due to remaining differences in vertical and horizontal smoothing of atmospheric variability (including co-location mismatch errors). The latter is difficult to assess, but one can expect that the bias and spread estimates resulting from the comparisons are close to the combined uncertainty of satellite and ground-based data, or at least the ex-ante satellite uncertainty in practice. The plots in Figure 6 to Figure 10 show that this is hardly the case (also see the discussion in the previous paragraph). The satellite measurement uncertainties provided in the product files do not cover the systematic and random uncertainties obtained by FRM comparisons (subtraction of the FRM uncertainties discussed in Section 3.2 does not make a difference). This means that the total satellite measurement and retrieval uncertainty is typically underestimated in the RAL v2.14 nadir ozone profile products. The only exception is given by the OMI tropospheric ozone data that have a bias below their ex-ante uncertainty. The total ex-post satellite uncertainty is an unknown number because of precision ignorance, but can be estimated to range in between the combined (quadratic sum) bias and satellite random uncertainty and the combined bias and comparison spread (although the latter contains error contributions that are not part of the satellite observation, like co-location mismatch).

Looking at the dependence of the L2 UV-VIS product comparison results on the eight influence quantities shown in Figure 6 to Figure 10, one can observe that the latitude band and total ozone column have the biggest impact on the RAL v2.14 retrieval performance. Especially in the UTLS and the troposphere the comparison variability is very high, which is also reflected in the strong differences in spread between different influence quantity ranges. Smaller biases are typically obtained





in the northern hemisphere and for larger total ozone columns. The latter is indeed expected to result in an improved satellite measurement and retrieval sensitivity, and thus more stable averaging kernel behaviour with smaller vertical dependences. On the other hand, the DFS and SZA behaviour is somewhat smaller and, as one can again expect for UV-VIS observations, rather similar, with the higher solar zenith angles typically corresponding to the larger DFS values and the smallest biases.

This effect is most clear for the GOME and SCIAMCHY instruments though, while the overall DFS dependence for the other instruments is less obvious. For all UV-VIS instrument except GOME-2B however, some satellite profiles with very low DFS, nearly-zero stratospheric sensitivity and high bias occur (mainly in the SAA, see previous sections). These profiles result from retrievals without stratospheric measurement information (hence the low DFS) and should appropriately be screened by users accordingly, e.g. using a DFS < 3 flag. Nadir ozone profiles flagged as such should then only be

considered for tropospheric ozone monitoring or fully rejected because of the increased bias.

Again more or less in line with nadir ozone profile retrieval expectations, the comparison results depend little on the surface albedo and effective cloud fraction, except for the lowermost 0 to 6 km retrieval layer. Higher ozone concentrations logically correspond with lower cloud fractions and higher albedos. Note however that the ECF and surface albedo dependence is also reflected, yet inversely, in the UTLS, due to the typically high sensitivity peak in this region and the low compensation

above. This effect is most clearly visible for the GOME-2B and OMI instruments. Instead of the full-profile effective cloud screening suggested by the RAL team now, one could thus apply layer screening up to the UTLS instead. Finally, for the UV-VIS retrievals under consideration the quarter and scan pixel index have hardly any effect on the comparison results, meaning that the RAL v2.14 retrieval algorithm copes with ozone seasonality and instrument viewing angle effects very appropriately.





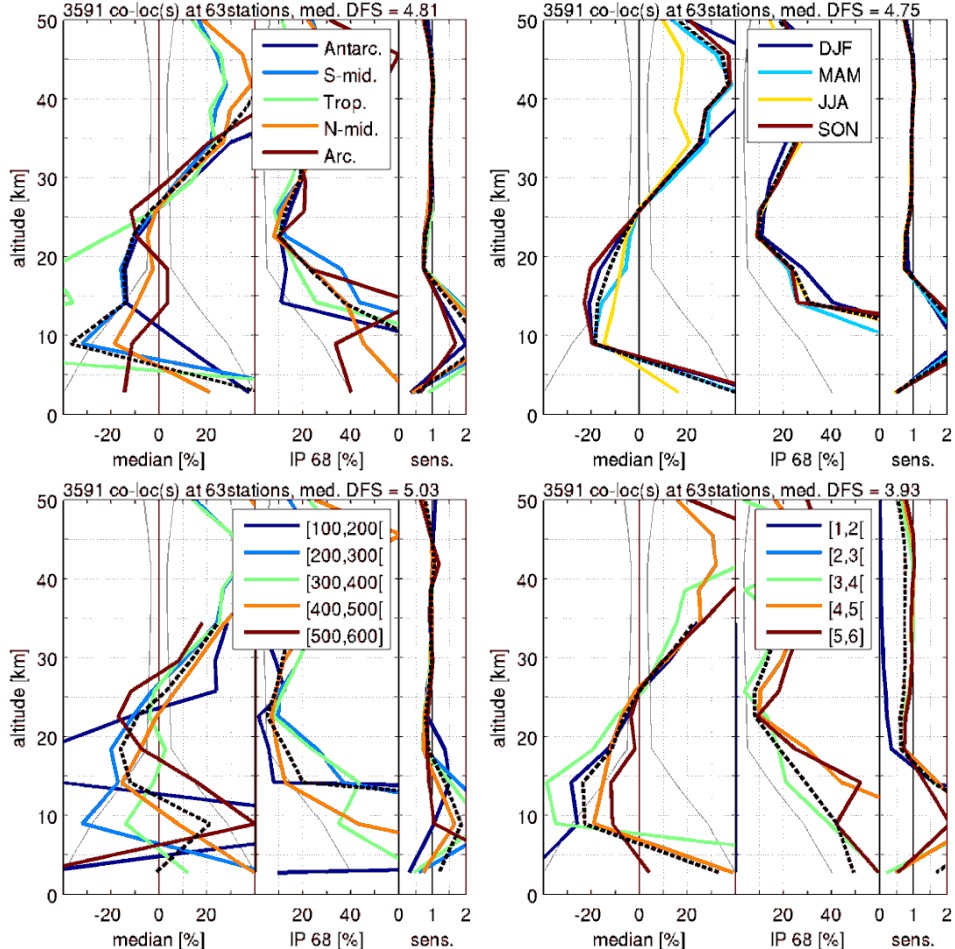




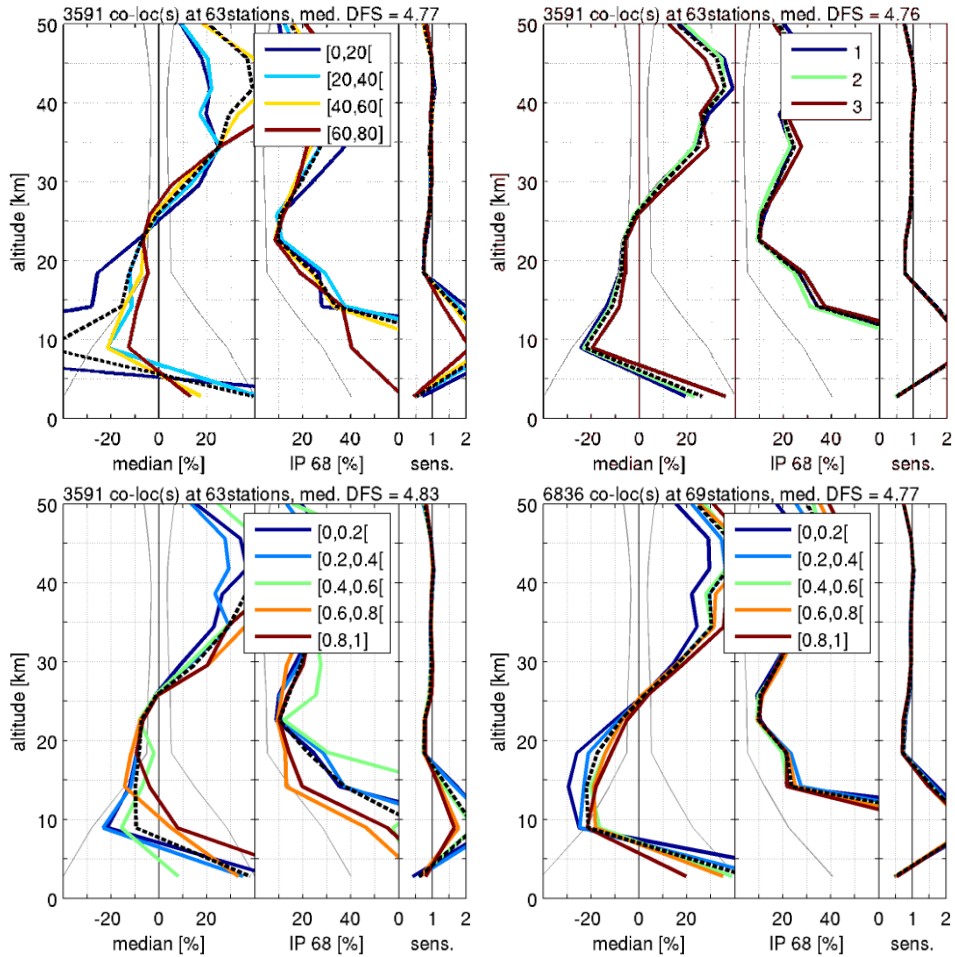

**Figure 6: Median relative differences, 68% interpercentile spreads, and vertical sensitivities for comparison of RAL v2.14 L2 GOME retrieved profiles with ground-based reference measurements (1996-2010). The same difference and information statistics are redistributed in each plot over several influence quantity ranges, with the influence quantities being (from left to right and top to bottom) latitude, quarter, total ozone column (DU), DFS, SZA, scan pixel index, surface albedo, and effective cloud fraction. The black dashed line shows the average of the coloured curves, while light grey lines indicate the satellite uncertainty provided in the product. The number of comparisons is higher for the latter as the ECF filter has been switched off.**





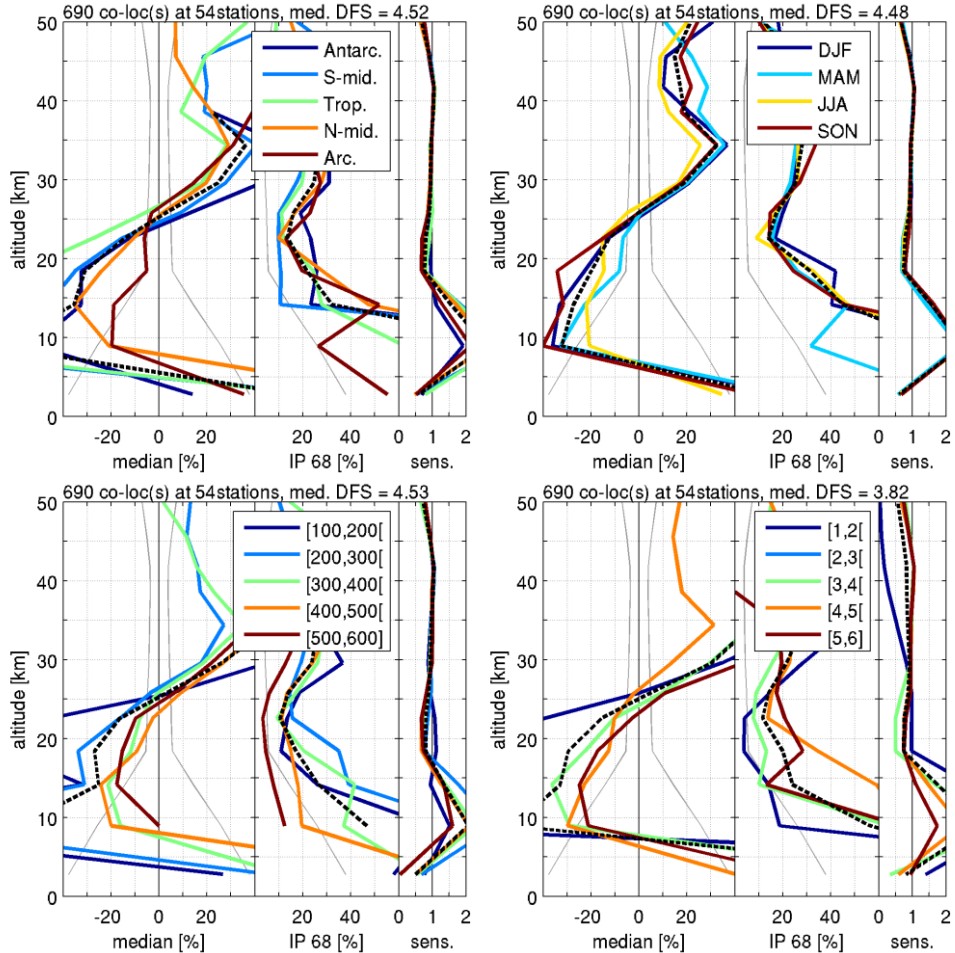



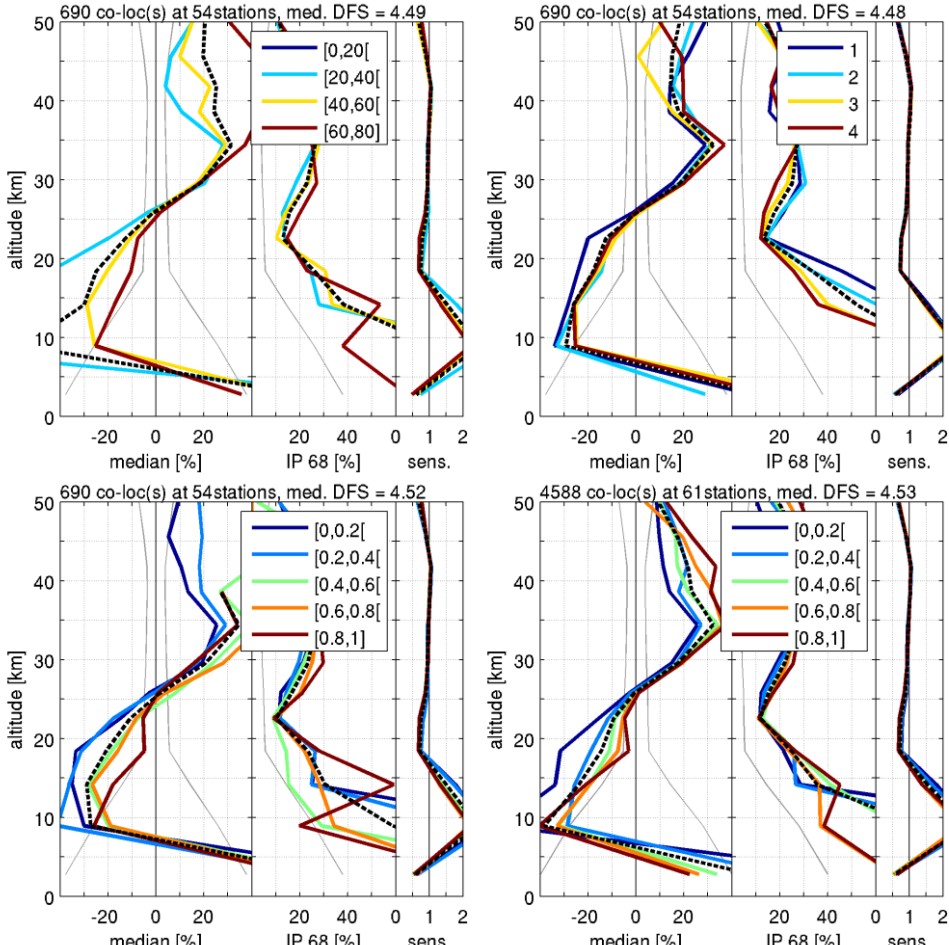

Figure 7: As for Figure 6, but for RAL v2.14 L2 SCIAMACHY data (2003-2010).





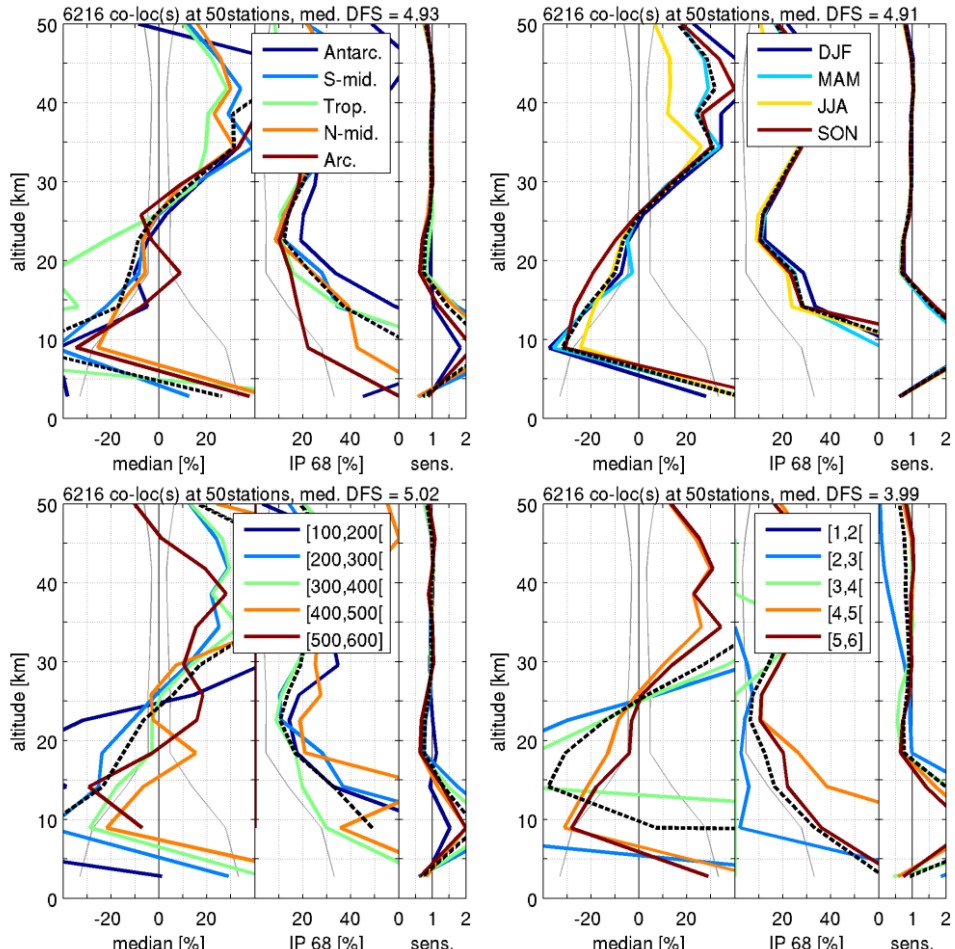





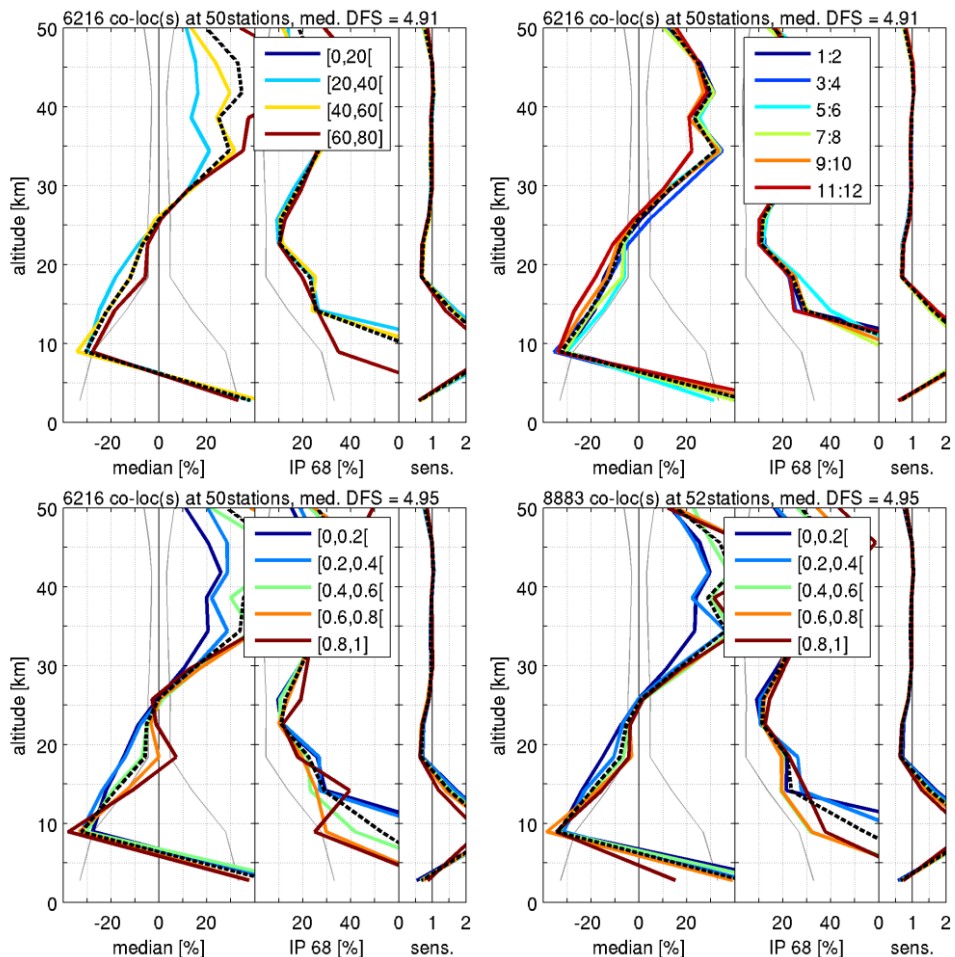

**Figure 8: As for Figure 6, but for RAL 2.14 L2 GOME-2A data (2008-2012).**





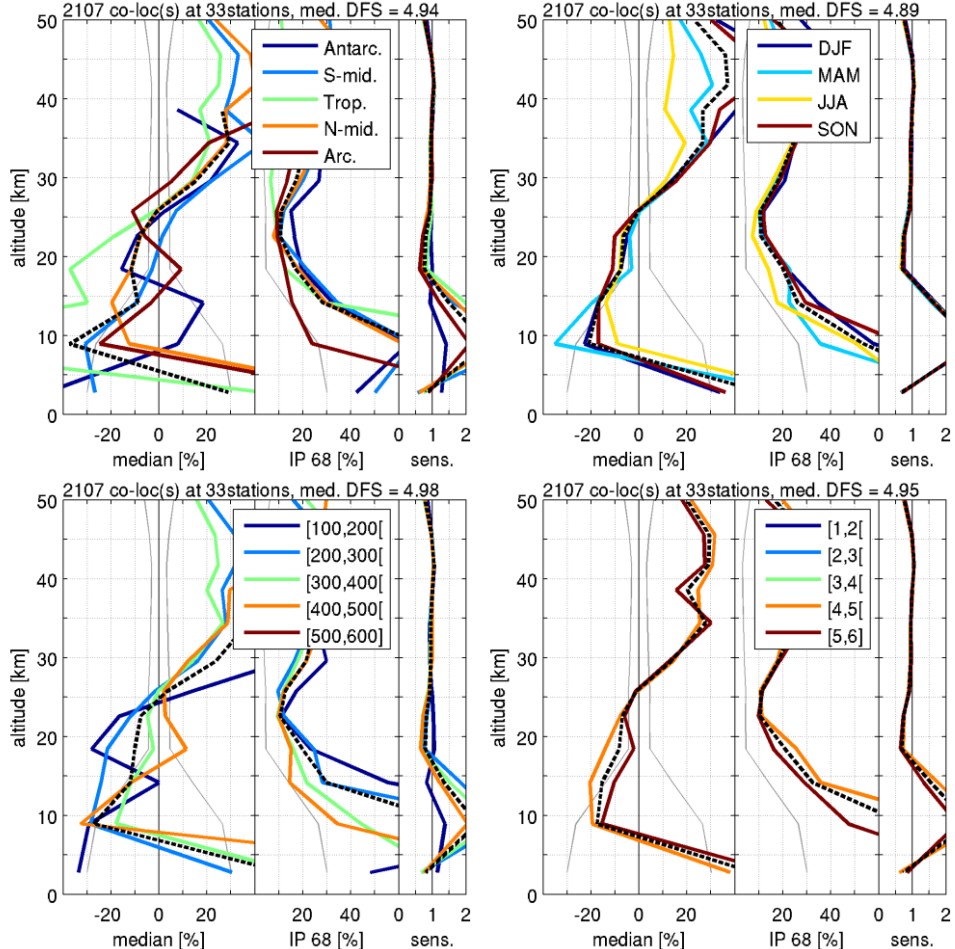





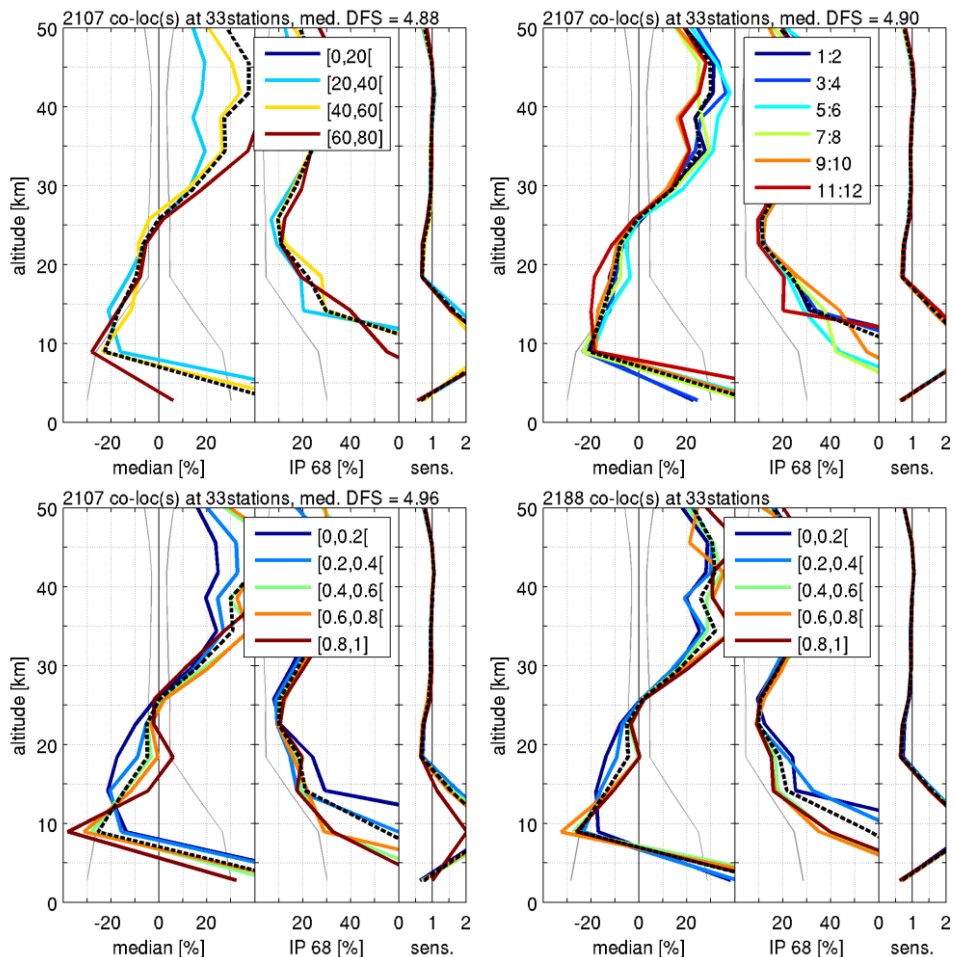

Figure 9: As for Figure 6, but for RAL v2.14 L2 GOME-2B data (2013-2015).





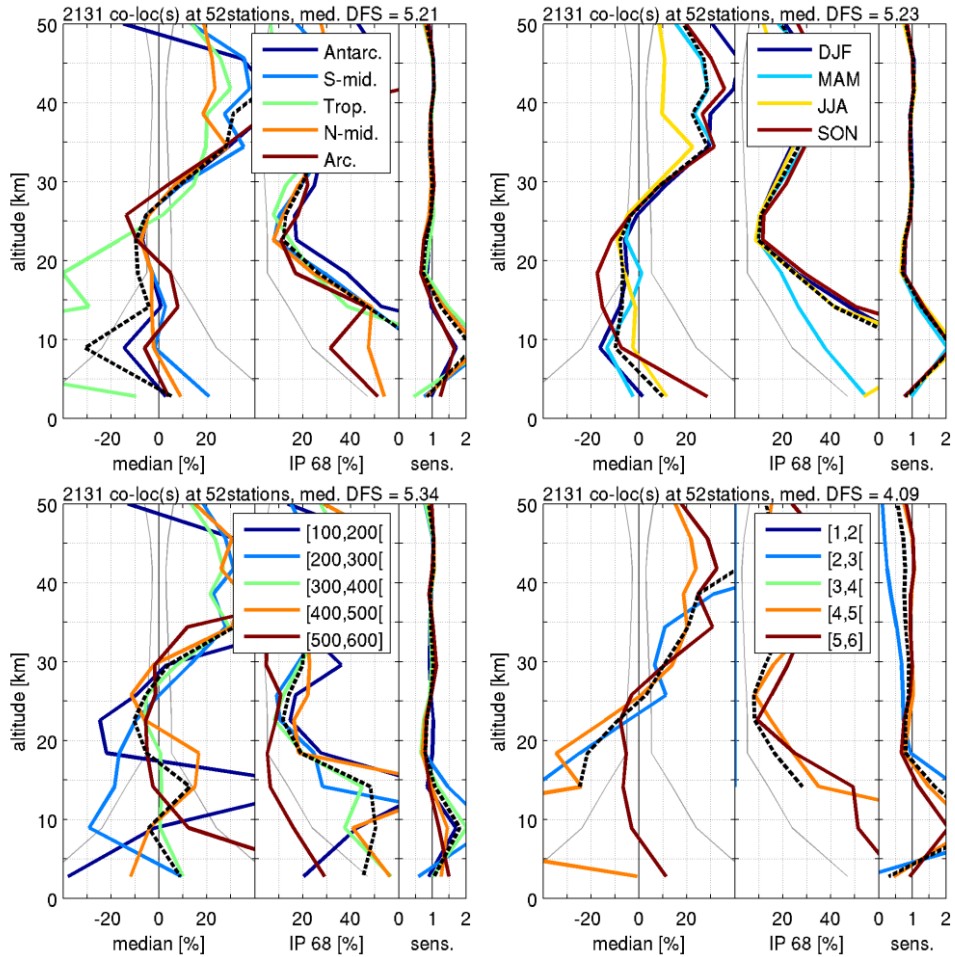





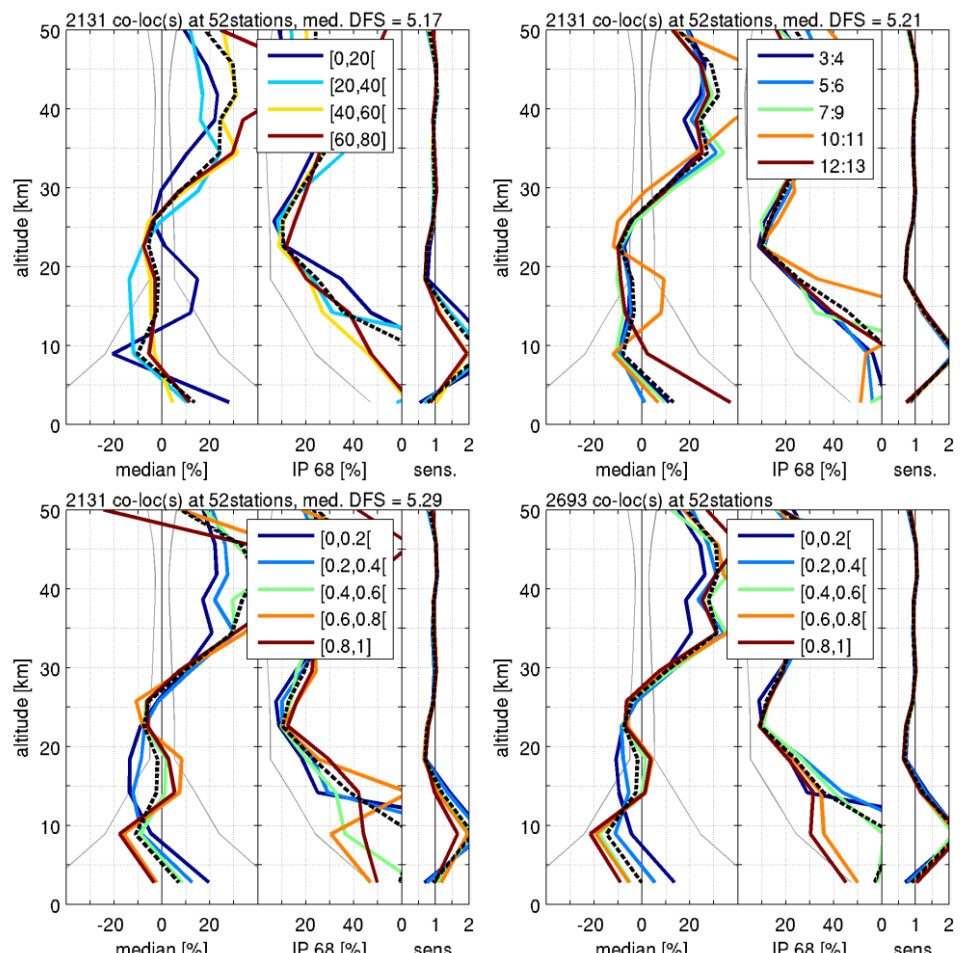

**Figure 10: As for Figure 6, but for RAL v2.14 L2 OMI data (2005-2015).**

## 5.3 L3 UV-VIS monthly gridded ozone product

5    Median relative differences and 68% interpercentile spreads for comparison of L3 GOME, SCIAMACHY, GOME-2A, and OMI data with ground-based reference measurements are presented in Figure 11. The same difference statistics are redistributed for each instrument over two influence quantity ranges, with the influence quantities being the latitude and quarter. Note the high numbers of co-locations in the title of each plot, as for each ground-based reference measurement an overlapping L3 data grid cell can be identified. As can be expected, the median relative differences roughly follow the bias

10    features of the respective L2 datasets for their comparison with ozonesonde and lidar data. These features, together with the corresponding spreads, however seem to be enlarged due to larger differences in spatiotemporal representativeness. The latter results from the lack averaging kernel smoothing that reduces vertical smoothing difference errors and the limited amount of reference data measurements per month (although at least six, see previous sections).



GOME level-3 data show a negative above-tropopause bias of 5-10 %, with exceptions in the tropical UTLS and Antarctic local spring (up to 50 %) due to ozone hole's vortex conditions. The corresponding spread is of the order of 10-30 %, with again outliers at the same two scenes. Especially during Antarctic spring (SON) the spread explodes to order of 100 %. Below the tropopause (100-200 hPa), GOME level-3 data show stronger negative and positive biases ranging between 10

and 30 %. Exceptions can be observed in the Arctic winter (DJF) and Antarctic spring (SON), with outliers ranging up to 60 % and -50 %, respectively. Corresponding spread values are of the order of 20-40 %, with the highest values again in Arctic winter.

The SCIAMACHY level-3 bias and spread values are very similar to those of the GOME level-3 comparison results. Only exceptions are the strong positive Arctic spring (MAM) bias in the troposphere (up to 40 %) and the availability of Antarctic

winter (JJA) data showing a strong negative bias in the UTLS and above (-30 to -40 %). Also the GOME-2 instrument on-board Metop-A shows a performance that is very similar to the GOME instrument in terms of level-3 bias and spread. The only significant difference is in the bias during the northern and southern DJF quarter: GOME-2A outliers are much more negative (up to -50 %) for the lowest partial columns. OMI's level-3 bias and spread again are very similar to those of the other three instruments, with the difference that the negative tropical tropospheric bias is more pronounced (-40 %) and a

positive tropospheric bias (30-50 %) is introduced in the southern hemisphere during local winter (JJA).

Overall one could state that between about 10 hPa and the tropopause (100-200 hPa), relative differences and spreads are of the order of -5 % and 10-30 %, respectively, for all four instruments, while the troposphere shows a 10-40 % bias (both positive and negative) and spread. Strong outliers however occur, typically in the troposphere of the Arctic winter (DJF), in the equatorial UTLS (order of 50 % positive for all seasons and instruments), and in the Antarctic local winter (JJA) and

spring (SON) due to strong ozone variability around the polar vortex.





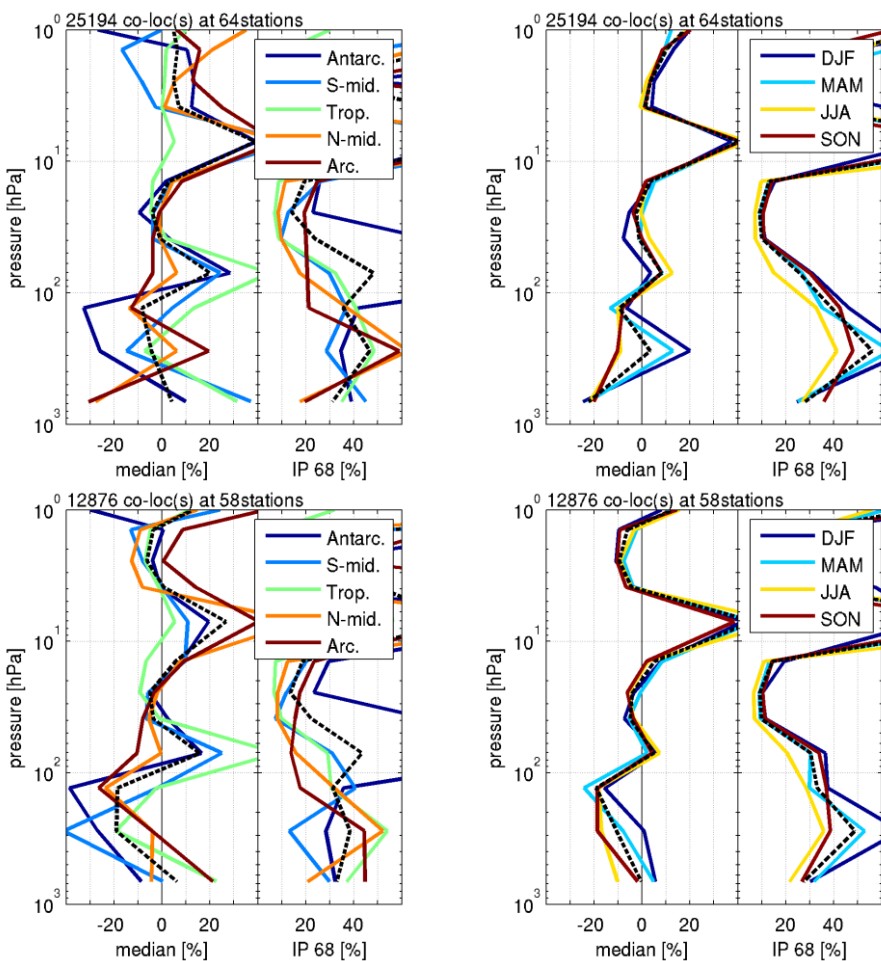





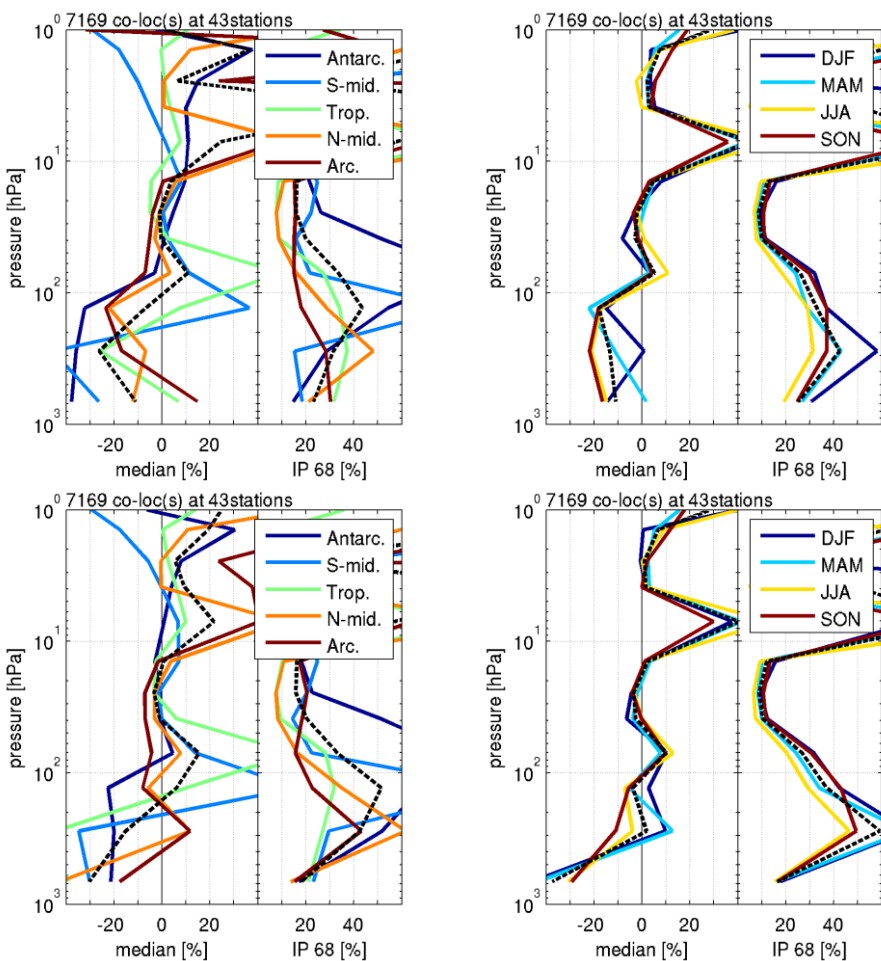

**Figure 11: Median relative differences and 68% interpercentile spreads for comparison of L3 GOME, SCIAMACHY, GOME-2A, and OMI data (top to bottom) with ground-based reference measurements. The same difference statistics are redistributed in each line over two influence quantity ranges, with the influence quantities being the latitude (left) and quarter (right). The black dashed line shows the average of the coloured curves.**

## 5.4 UV-VIS L2 and L3 drift studies

Relative decadal drift and 68% interpercentile spreads for comparisons of L2 and L3 GOME, SCIAMACHY, GOME-2A, and OMI data with ground-based reference measurements are collected in Figure 12. As discussed in the previous section for their bias and spread behaviour, the similarity between the L2 and L3 UV-VIS drift results for the same instrument appears very clearly. Again however, features in the L2 statistics are enlarged for the L3 data due to larger differences in spatiotemporal representativeness.

The GOME L2 and L3 stratospheric drift typically do not exceed 10 %/decade values, with the exception of an almost 20 %/decade positive drift near the southern pole lower stratosphere and an equally large L3 peak around 35 km. Only the latter however is clearly significant in terms of the corresponding 95 % drift confidence interval (CI, as horizontal error bars). This



can also be observed from the highly peaked (> 60 %) IP68 spread on the differences (right-hand panel in each plot of Figure 12). This peak indeed partially reflects the instrument's drift, as the spread is not determined from the drift residuals but with respect to the overall median difference. A large drift will as such contribute to a large spread. The negative drift values appearing above 45 km are considered less trustworthy because of the lidar reference data sparseness. The GOME

tropospheric drift equals about -5 % per decade on average, but at the lowest altitudes ranges from -20 %/dec. at the southern pole to 20 %/dec. near the equator. Yet again the L2 drifts remain within the CI and are therefore insignificant.

SCIAMACHY drift results strongly differ from the GOME observations: Although still mostly insignificant, the above-tropopause drift is of the order of -10 % per decade and shows the same L3 outlier at 35 km. Below the tropopause however, the drift ranges from about 20 %/dec. at the poles to 50-60 %/dec. towards the equator. This entails that in the mid-latitudes

(both north and south) and tropics this drift is significant. The GOME-2A drift results come close to the SCIAMACHY drift performance, although the sub-tropopause drift is even stronger (around 50 %/decade) and significant globally. Besides, a significant negative drift of the order of 30 %/dec. also appears in the UTLS, which is strongest around the equator, reaching -70 % per decade around 100 hPa.

Despite the occurrence of insignificant negative drifts in the northern hemisphere, the OMI L3 tropospheric drift is

significantly positive (around 40 %/decade on average) in the southern hemisphere and the tropics, resulting in a global average L3 tropospheric drift of the order of 15 % per decade (see Figure 12). The L2 tropospheric drift equals about 5 to 10 %/dec. only and is close to insignificant. It is remarkable that the OMI L3 drift is typically 10 % negative in the UTLS (with -40 % per decade values around the equator), while in the stratosphere above an average 10 %/dec. positive drift can be observed. Both L2 and L3 show a negative close to 20 % per decade value just below 40 km. These results and their

significance are in qualitative agreement with Huang et al. (2017) on the OMI PROFOZ retrieval product.

On the global scale, as shown in Figure 12, the decadal drift is order of 5 % negative and insignificant for GOME, and order of -15 % and 10 % insignificant (except for the Tropics) for OMI's L2 stratosphere and troposphere, respectively. A significant positive drift of the order of 40 % per decade is observed for SCIAMACHY and GOME-2A below the tropopause. GOME-2A moreover shows a significant 30 %/decade negative drift in the UTLS at all latitudes.





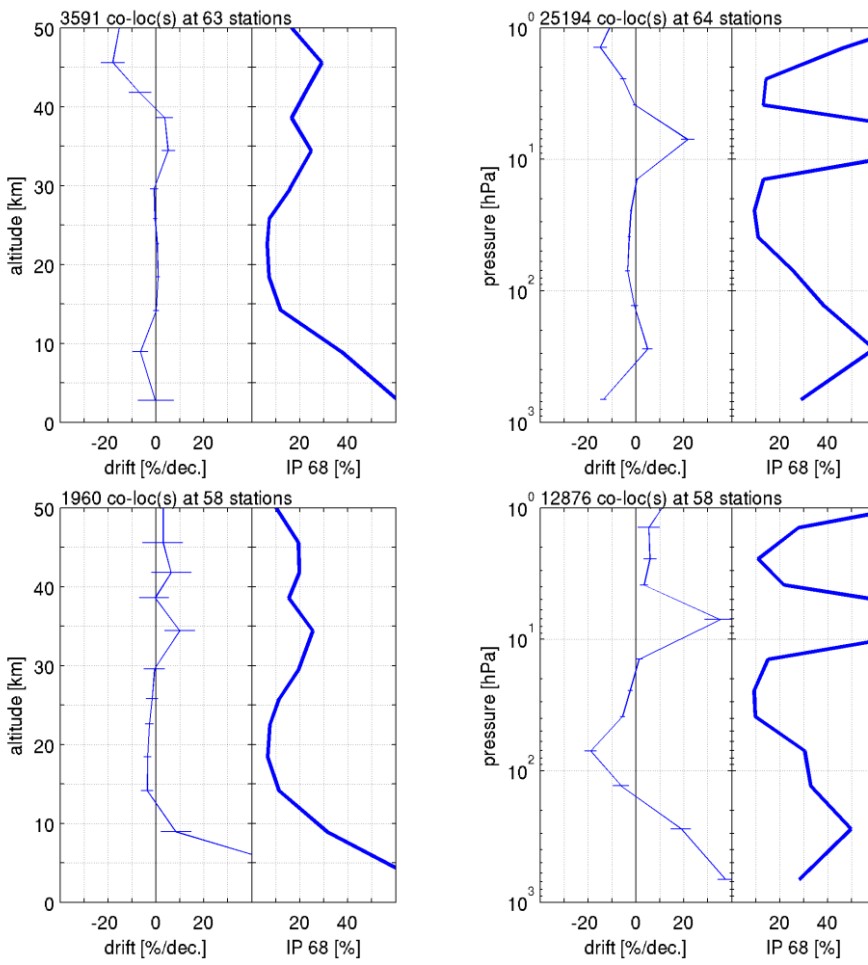




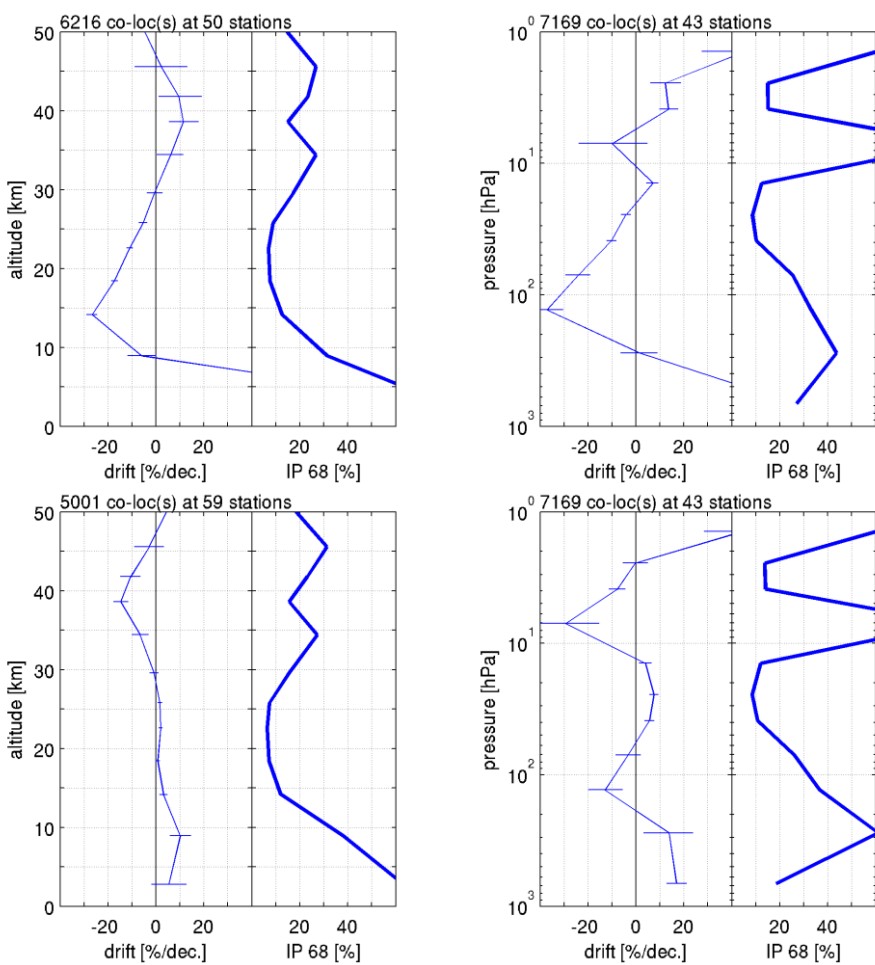

**Figure 12: Relative decadal drift and 68% interpercentile spreads for comparisons of L2 (left) and L3 (right) GOME, SCIAMACHY, GOME-2A, and OMI data (top to bottom) with ground-based reference measurements. Two sigma error bars, resulting from a bootstrapping with 1000 samples, are added to the drift profiles.**

## 5.5 L4 assimilated data

The L4 1996-2013 data, constructed by data assimilation at KNMI from merged RAL v2.14 GOME and GOME-2A observations, can be compared with ground-based reference profiles directly. The single two-by-three degree box that overlaps with the ground measurement within three hours is thereby taken as the co-located measurement. The number of co-locations and stations however is smaller than for the L3 data, as data from 64 stations (that have been used for satellite bias correction during assimilation) are omitted from the comparative analysis. Median relative differences and 68% interpercentile spreads for comparison of the L4 assimilated nadir ozone profile data with ground-based reference measurements are collected in Figure 13, redistributed over two influence quantity ranges (latitude and quarter). The corresponding relative decadal drift and overall 68% interpercentile spread profiles are added as well.



The most remarkable result that can be observed from the UV-VIS L4 comparison statistics is that, as a result of the model assimilation, the typical Z-shape of the L2 bias has disappeared. The L4 bias typically remains below 10 % (positive and negative) with the exception of a strong positive outlier around 5 hPa (as for the L3 data) and the surface boundary layer, and a 20 % positive to negative fluctuation around the UTLS that is strongest in the tropics (~ 50 % positive for all seasons, with

5 a similar but smaller bias feature in the southern hemisphere). This entails that the L2 and L3 comparison features in the Antarctic spring (SON) with ozone hole conditions and in most of the troposphere have been strongly reduced. The L4 spread remains close to the L2 and L3 values, though with an even stronger reduction (to 20 %) in the troposphere than the L3 comparisons. Moreover, the remaining L4 drift is of the order of a few percent only and insignificant, i.e. within the 95 % CI, for all altitudes up to about 40 km globally.

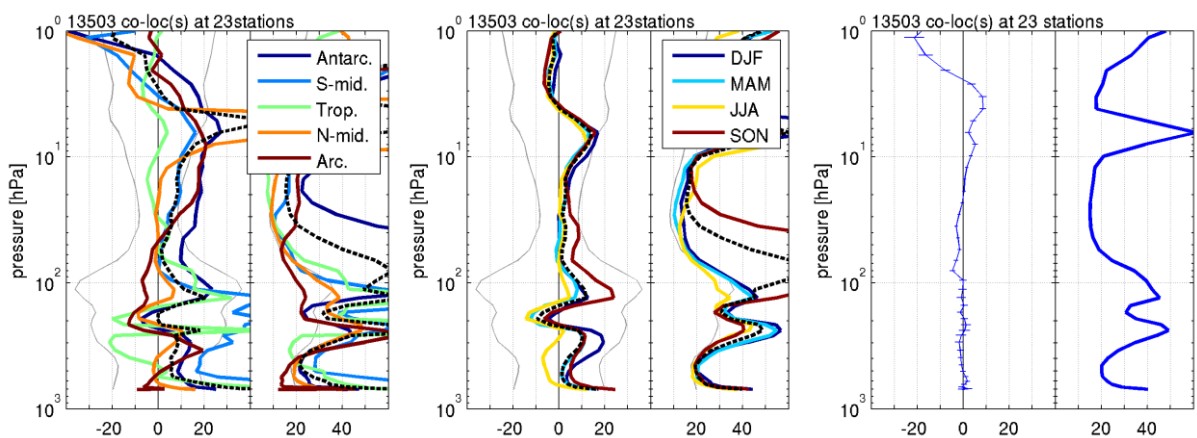

**Figure 13: Median relative differences and 68% interpercentile spreads for comparison of L4 assimilated nadir ozone profile data with ground-based reference measurements (top and middle). The same difference statistics are redistributed over two influence quantity ranges, with the influence quantities being the latitude (top) and quarter (middle). The black dashed line shows the average of the coloured curves, while light grey lines indicate the satellite uncertainty provided in the product.. The bottom plot shows the corresponding relative decadal drift and 68% interpercentile spread. Two sigma error bars, resulting from a bootstrapping with 1000 samples, are added to the drift profile.**

**5.6 L2 TIR nadir ozone profiles**

As for the L2 RAL v2.14 UV-VIS retrievals, Figure 14 and Figure 15 now contain the median relative differences, 68%

20 interpercentile spreads, and vertical sensitivities for the comparison of FORLI v20151001 retrieved IASI profiles with ground-based reference measurements (IASI-A for 2008-2015, IASI-B for 2013-2015). Difference and information statistics are again redistributed in each plot over several influence quantity ranges, with the influence quantities now being the latitude, quarter, total ozone column (DU), DFS, SZA, scan pixel index, and thermal contrast. For IASI-A in Figure 14, the corresponding relative decadal drift and overall 68% interpercentile spread are also added.

25 As already pointed out in the information content studies, the IASI-A and IASI-B results are very similar, showing no significant differences between their respective statistics. Overall the FORLI v20151001 IASI retrieval data products show a




less than 10 % and insignificant stratospheric bias, a 10 to 30 % positive bias in the UTLS, and an order of 10 % negative bias in the troposphere. The latter is in agreement with an initial IASI tropospheric ozone (also retrieved with FORLI v20151001) validation exercise using ozonesonde reference measurements performed by Boynard et al. (2016). Possible reasons for the UTLS bias are discussed in Dufour et al. (2012). Taking into account the FRM uncertainties discussed in

Section 3.2, the ex-ante IASI uncertainties provided in the product files (light grey lines in the plots) are typically of the order of the bias, except in the UTLS. The ex-post random uncertainty, as estimated by the spread, is roughly twice as large, except for the lower tropics. This means that overall the total satellite measurement and retrieval uncertainty is underestimated in the IASI FORLI v20151001 nadir ozone profile products. The comparison results show hardly any scan angle dependence or seasonality, except for some larger systematic differences around the Antarctic ozone hole that can be

partially attributed to co-location errors at the edge of the polar vortex. The remaining meridian dependences are typically limited to stronger UTLS bias fluctuations in the tropics.

Both the polar sub-tropopause and tropical UTLS outliers seem to go together with a thermal contrast dependence of the differences (clearer for IASI-A than for IASI-B) that also agrees with the sensitivity dependence. One would expect the thermal contrast to be mainly influential in the lowermost layers, but the information content studies on the IASI product

have indeed demonstrated that the corresponding averaging kernels show significant vertically interdependent oscillations. Therefore the polar sensitivity outliers around 30 km altitude can be related to the strongly negative thermal contrasts and typically go together with very low DFS values (below two, suggesting screening upon this threshold) and strong ozone over-estimations. The latter is again clearer for the longer IASI-A time series, wherein the highest total ozone column profiles have the lowest DFS values. Finally, differences can be observed between the IASI day-time (SZA < 83°) and night-

time (SZA > 91°) measurements, which are most clear for the largest solar zenith angles (140 to 180°). Due to the small numbers of co-locations for the latter however, it is difficult to attribute any significance to these differences.

Looking at latitude-resolved drift studies for the Ozone_cci IASI-A nadir ozone profiles (not shown), a significant decadal negative drift of the order of 25 % or higher can be observed in the Antarctic UTLS and the northern hemisphere troposphere. On the global scale (see Figure 14), the significance of these drifts remains in terms of the corresponding 95 %

drift confidence intervals (horizontal error bars) and is again reflected in the peaked UTLS IP68 spread on the differences (40 %) as the spread is not determined from the drift residuals but with respect to the overall median difference. A less pronounced positive drift is detected around 30 km altitude. Part of the overall negative tropospheric drift of the FORLI v20151001 IASI retrievals could however be due to a change in the processing of the IASI L2 processor (e.g. temperature profile) at EUMETSAT that changed to version 5.0.6 in September 2010. This idea is supported by Boynard et al. (2017),

who have observed that the IASI-A FORLI v20151001 tropospheric drift becomes statistically insignificant if calculated from the Sept. 2010 to 2016 period retrievals only.





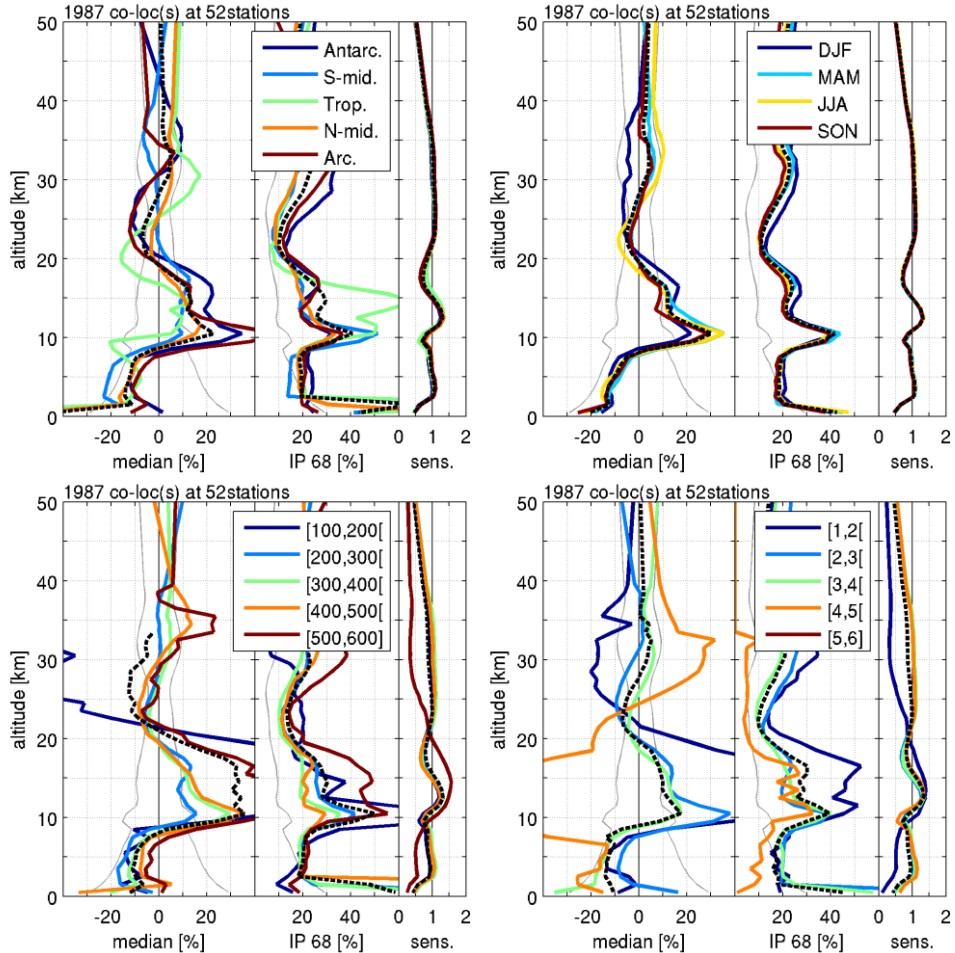





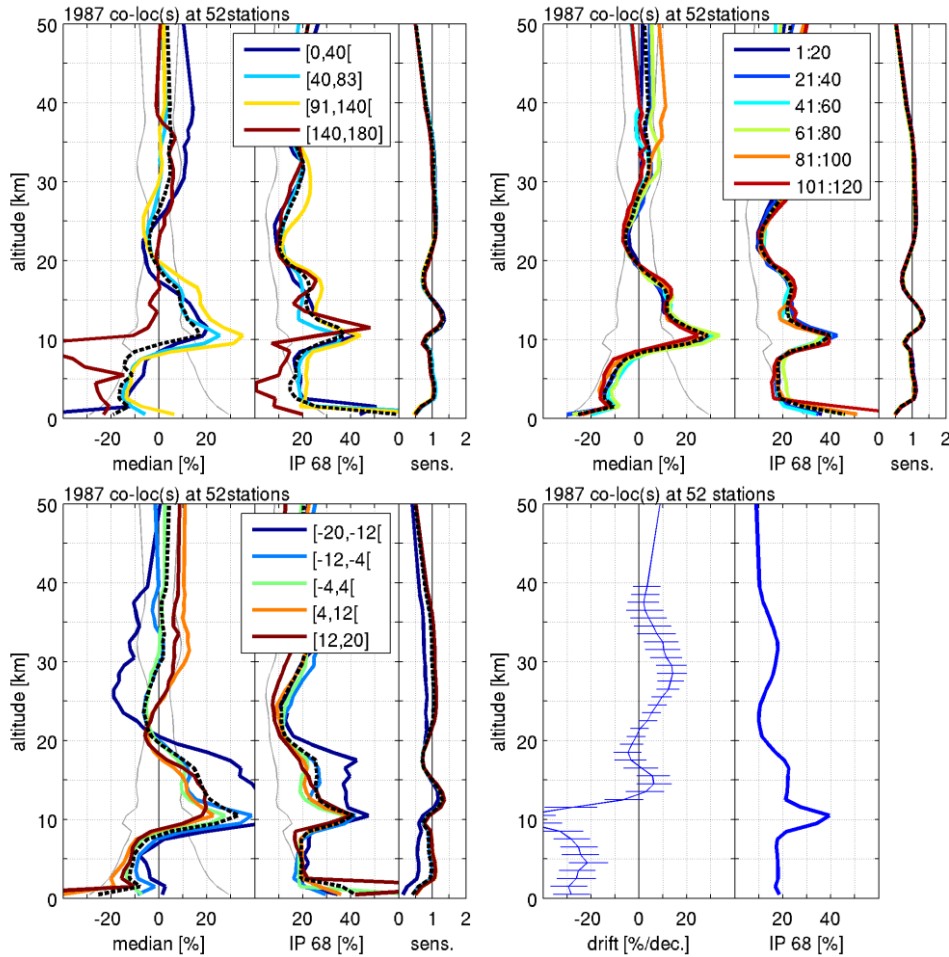

**Figure 14: Median relative differences, 68% interpercentile spreads, and vertical sensitivities for comparison of FORLI v20151001 L2 IASI-A retrieved profiles with ground-based reference measurements (2008-2015). The same difference and information statistics are redistributed in each plot over several influence quantity ranges, with the influence quantities being (from left to right and top to bottom) latitude, quarter, total ozone column (DU), DFS, SZA, scan pixel index, and thermal contrast. The black dashed line shows the average of the coloured curves, while light grey lines indicate the satellite uncertainty provided in the product.. The bottom right plot contains the corresponding relative decadal drift and 68% interpercentile spread. Two sigma error bars, resulting from a bootstrapping with 1000 samples, are added to the drift profile.**



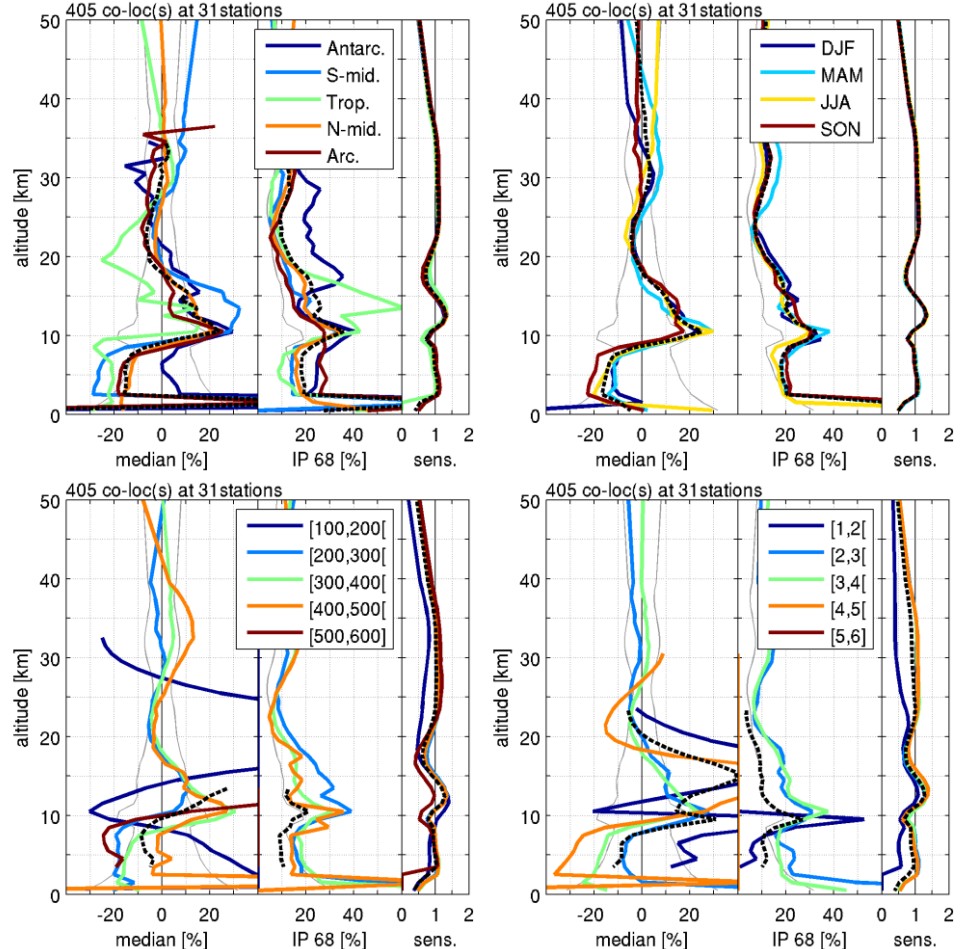





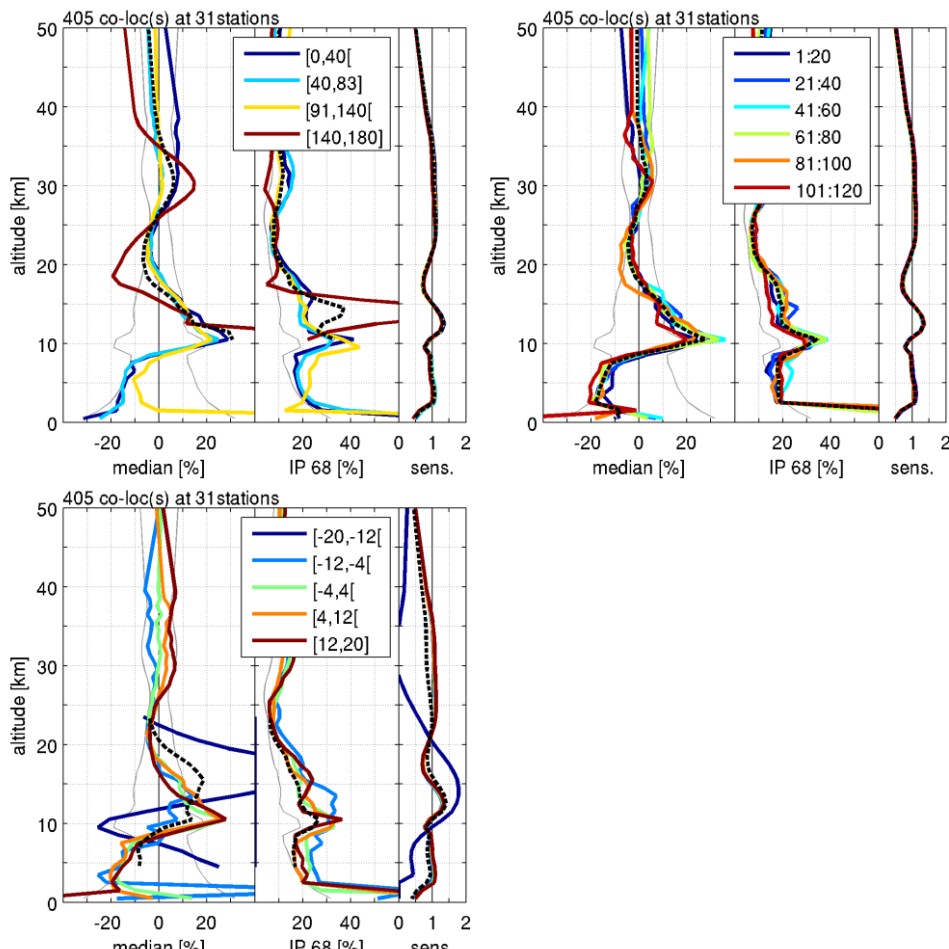

**Figure 15: As for Figure 14, but for FORLI v20151001L2 IASI-B data (2013-2015). Because of the limited temporal extent of this product, no drift study has been performed.**

## 5.7 L3 TIR monthly gridded tropospheric ozone product

Time series of median relative differences (in solid blue), spreads (in dashed blue), and linear drift (green) for direct comparisons of the IASI-A level-3 monthly gridded mean tropospheric ozone column data (integrated from 0 to 6 km) with integrated ozonesonde reference data (at stations with at least six valid measurements per month) are determined within five latitude bands and plotted in Figure 16. The yearly linear drift value and its 95 % confidence interval as an uncertainty estimate on the derived slope are both determined from a bootstrapping technique using 1000 subsamples and are added in the lower-left corner of each graph.

The IASI-A TIR monthly gridded tropospheric ozone column data for January 2008 to December 2012 show a strong seasonal variation in their comparison with the integrated ozonesonde data, ranging up to 100 %, especially around the southern pole. Despite this strong seasonality, and in agreement with the IASI-A L2 comparison statistics, median relative differences throughout the whole time series range between 25 % negative in the northern mid-latitudes and 30 % negative



in Antarctica, with a nearly zero overall bias around the equator. The corresponding spread decreases from about 25 % in the tropics to about 5-10 % towards the poles. The drift on the other hand increases from less than one percent per year negative in the tropics to up to -4 % per year around the southern pole. In contrast with the IASI-A L2 drift study results, none of these drifts however is significant, as the 95 % confidence intervals in combination with the comparison spreads indicate:

5  Where the confidence interval is fully negative, as is the case for the mid-latitudes, the distance of the confidence interval from zero drift is much smaller than the average spread on the differences. This difference between the IASI L2 and L3 significance of the drift is mainly due to their difference in spatiotemporal representativeness with respect to the ground-based reference data (averaging kernel smoothing, vertical integration, and monthly averaging).

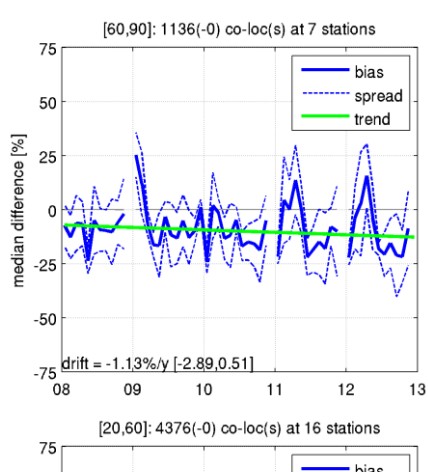

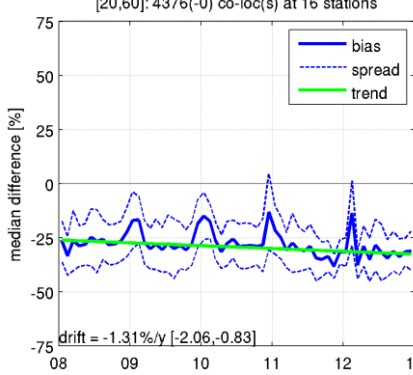





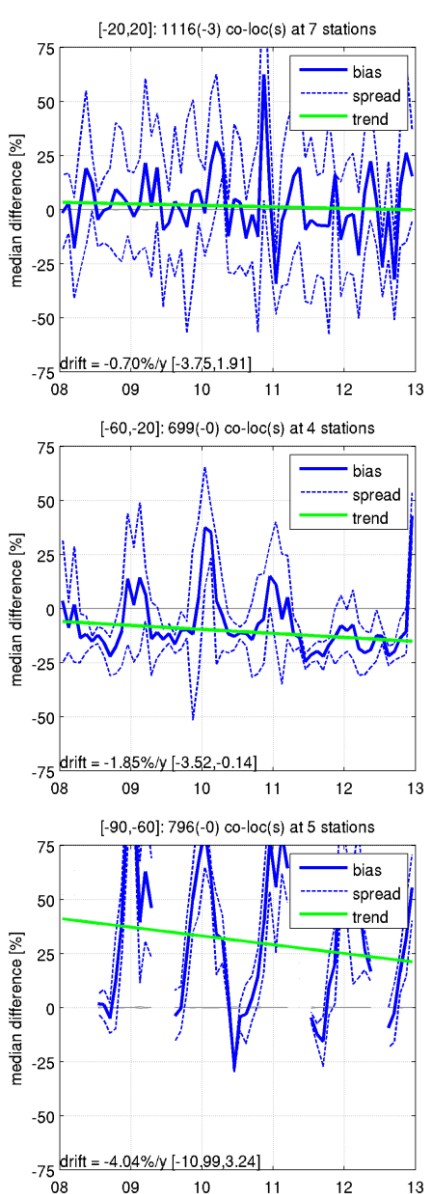

**Figure 16: Time series of the median bias (solid blue), spread (dashed blue), and linear drift (green line) for direct comparisons of**
**IASI L3 monthly gridded mean tropospheric ozone column data (0 to 6 km) with vertically integrated ozonesonde reference data**
**(at stations with at least six launches per month), divided into five latitude bands (sorted north to south). The number of filtered**
**values is added between brackets in the title of each plot, while the yearly linear drift value and its 95 % confidence interval are**
**added in the lower-left corner.**





# 6  Discussion

Table 5 summarises the major QA/validation quantities discussed throughout this work, their corresponding typical values as discussed in the previous sections, and provides associated GCOS user requirements for the entire Ozone_cci nadir ozone profile CRDP, meaning that UV-VIS and TIR measurement and retrieval based products are combined. These 13 ozone ECV datasets together cover the 1995 to 2015 time period globally, which is sufficiently long for (drift-corrected) ozone trend studies according to the GCOS user requirements (UR). Yet the ongoing and upcoming satellite observations of both the GOME-type (GOME-2 on Metop-A/B, Sentinel-5 Precursor TROPOMI, and the upcoming Copernicus Sentinel-5 series) and the IASI-type (IASI on Metop platforms and IASI-NG on Metop-SG platforms) will even extend the available time series. Expecting for these data a similar or even improved quality in terms of information content, total uncertainty, and especially horizontal resolution (cf. Sentinel-5p with a 7 by 7 km ground pixel), the Ozone_cci CRDP seems fit for long-term vertically-resolved ozone climate monitoring and modelling as e.g. done in the Tropospheric Ozone Assessment Report (TOAR), the WMO/UNEP Ozone Depletion Assessment, and the SPARC LOTUS initiative. All nadir ozone profile products under study indeed also fulfil the GCOS user requirements in terms of observation frequency and horizontal and vertical resolution. Only for the latter one has to keep in mind that all L2 nadir ozone profile observations show UTLS sensitivity outliers and are strongly correlated vertically due to averaging kernel fluctuations that extend far beyond the (typically tropospheric) kernel's 15 km FWHM.

The Ozone_cci CRDP nadir ozone profile products typically do not comply with the GCOS user requirements in terms of total uncertainty and decadal drift. The total uncertainty is thereby determined as the quadratic sum of the products' systematic and random uncertainties, which on their turn are estimated from the comparison (with ground-based reference measurement) bias and spread, respectively. Note that this as a conservative estimate, as the bias and spread also include uncertainties due to smoothing and sampling differences between the satellite data and the FRM. Whereas the RAL v2.14 UV-VIS retrieved products show a typical Z-curve bias with strong 20-40 % positive (stratosphere) and negative (UTLS) maxima, the FORLI v20151001 systematic uncertainty is rather consistently of the order of 10 % in the stratosphere and troposphere, but shows stronger fluctuations (20 to 40 %) in the (especially tropical) UTLS. Total uncertainties therefore range from about 10 % at minimum in the stratosphere to at least 20 % in the troposphere (for IASI), and even higher values in the UTLS and for the UV-VIS instruments. Comparison statistics for the L3 monthly gridded averages are obviously of the same order, but L2 features can be both enlarged or reduced due to clear differences in spatiotemporal representativeness (also with the FRM data). KNMI's L4 data contain a remaining 10 % bias, with the exception of a positive outlier around 5 hPa and near the Earth's surface, and an order of 20 % fluctuation around the UTLS that increases to about 50 % in the tropics.

Drift studies for all nadir ozone profile CRDP products (except for the Metop-B instruments) show that the 1 to 3 % per decade GCOS requirement is only met by the L4 UV-VIS data. The higher drift values are however found to be mostly insignificant for the L2 GOME and OMI instrument retrievals, and for the L3 TIR data. The SCIAMACHY and GOME-2A



products however have a strong positive drift (up to 40 %) in the troposphere, and GOME-2A moreover shows a 20 % per decade negative drift around the tropopause. The FORLI IASI-A instrument retrieval shows an order of 25 % significant negative drift in the Antarctic UTLS and Northern Hemisphere troposphere only. Together with the systematic uncertainty studies, these drift results call for an appropriate altitude-dependent bias and drift correction of the L2 Ozone_cci nadir

ozone profile products by data users for climate and atmospheric composition monitoring and modelling purposes.

Applying bias and drift corrections to the nadir ozone profile CRDP presented in this work straightforwardly might not yield optimal results however. Next to the L2 data screening recommended by the respective data providers as summarised in Table 3, the validation results presented in the previous sections point at additional data screening options. In the UV-VIS instrument datasets (except for GOME-2B), some satellite profiles with very low DFS, nearly-zero stratospheric sensitivity

and high bias occur, mainly around the South-Atlantic anomaly. By insertion of e.g. a DFS < 3 flag, these profiles could be fully screened or considered for tropospheric ozone monitoring only. The latter would be equivalent with an altitude-dependent screening, which could also be used as an alternative to the full-profile effective cloud screening advised by the RAL team. Comparison results have shown that one could apply a layer screening up to the UTLS instead, as the stratospheric ozone retrieval is hardly affected by the ECF (or surface albedo). Analogously, the bias outliers for the FORLI

v20151001 IASI retrievals in the polar troposphere and the tropical UTLS go together with a thermal contrast and sensitivity dependence of the differences. These profiles could therefore be excluded from any further use by insertion of a strongly negative thermal contrast or low DFS value screening, e.g. shifting the DFS screening threshold from one (as suggested by the ULB/LATMOS retrieval team) to two. As for the RAL data, vertically-resolved profile screening could additionally reject consistent altitude-dependent bias or drift outliers.

| QA quantity (GCOS UR) | UV-VIS L2 | UV-VIS L3 | UV-VIS L4 | TIR L2 | TIR L3 (TTC) |
|---|---|---|---|---|---|
| Time period (1996-2010) | 1995-2015 | 1996-2015 | 1996-2013 | 2008-2015 | 2008-2012 |
| L2 observation frequency (daily to weekly) | Global coverage within 3 days | / | / | Both day-time and night-time daily | / |
| Horizontal resolution (20-200 km) | 32 to 160 km along track, 52 to 320 km across | 1° by 1° (~115 km at equator) | 2° by 3° (~230 by 345 km at equator) | 12 km | 1° by 1° (~115 km at equator) |
| Vertical resolution (6 km to troposphere) | Fixed grid with up to 6 km layers but ~15 km kernel width and SZA dep. tropospheric fluctuations | Fixed layers of a few km thickness | Fixed layers of 1-2 km thickness | Fixed 1 km gird but 10-15 km kernel width and strong UTLS and tropospheric fluctuations | 0 to 6 km integrated column |
| DFS | 4 to 5.5 with 0.5 seasonality | / | / | 2-4 with strong meridian and seasonal dep. | / |



| Vertical sensitivity | UTLS peak ~3 with under-sensitivity right above and below | / | / | Outliers around UTLS from -1 to 2 | / |
|---|---|---|---|---|---|
| Height registration uncertainty | < 10 km | / | / | ~0 at 40 km to about 30 km near the surface | / |
| Systematic uncertainty estimated from comp. bias | Z-curve with maxima at 20-40 % pos. (stratosphere) and neg. (UTLS) | Overall -5 % in stratosphere, +/- 10-30 % in troposphere | < 10 % with exception pos. outlier around 5 hPa and surface, 20 % pos. to neg. fluctuation around UTLS (~50 % in tropics) | < 10 % stratospheric bias, 20-40 % pos. (UTLS) to ~10 % neg. (troposphere) | -25 % in NH, -30 % in Antarctica yet nearly zero around equator |
| Random uncertainty estimated from comp. spread | U-curve with 10 % minimum around 25 km | 10-30 % in stratosphere, 20-40 % in troposphere | 10-30 % in stratosphere, 20 % in troposphere | Order of bias, showing similar features | ~25 % in tropics to ~10 % towards the poles but up to 100 % seasonality |
| Total uncertainty (16 % below 20 km, 8 % above 20 km) | 10 % minimum at 25 km, increasing above and below | From ~10 % in stratosphere at minimum to 20-50 % in troposphere | 15-30 % in stratosphere at minimum, higher below | ~10 % stratosphere, 20 % in troposphere, higher in UTLS | ~25 % in tropics to ~30 % towards the poles with up to 100 % seasonality |
| Dependence on influence quantities | latitude and TOC have biggest impact especially in UTLS and troposphere, higher SZA corresponds to larger DFS and smaller bias, small surface albedo and ECF dep. propagates to higher altitudes | Strong bias outliers in the troposphere of Arctic winter, equatorial UTLS, and Antarctic local winter and spring | L2/3 features in Antarctic spring and troposphere are strongly reduced but tropical UTLS bias remains | TC especially in polar troposphere and tropical UTLS, agrees with sensitivity dep., no seasonality except for Antarctic ozone hole | Strong meridian dependence and seasonality |
| Stability (1-3 %/dec.) | No significant GOME and OMI drift, -20 %/dec. GOME-2A drift around TP, strong pos. SCIAMACHY and GOME-2A tropospheric drift | Significant ~20 %/dec. peaks around 40 km, -20 %/dec. GOME-2A drift around TP, strong pos. SCIAMACHY and GOME-2A tropospheric drift | Order of a few percent at maximum, insignificant up to 40 km | ~25 % neg. in Antarctic UTLS and troposphere, ~15 %/dec. pos. around 30 km | ~10 % neg. in tropics to ~40 % neg. around southern pole yet insignificant |

**Table 5: Major QA/validation quantities, their corresponding typical values, and indication of GCOS user requirement (UR) compliance for the Ozone_cci nadir ozone profile CRDP.**



## 7 Conclusions

This work, the second in a series of four Ozone_cci papers, reports for the first time on data content studies, information content studies, and comparisons with co-located ground-based reference observations for all thirteen nadir ozone profile data products that are part of the Climate Research Data Package (CRDP) on atmospheric ozone of the European Space

Agency's Climate Change Initiative. These products consist of five L2 UV-VIS instrument retrieval datasets, two L2 TIR retrieval datasets, four UV-VIS L3 monthly gridded data series, a merged UV-VIS L4 product, and a 0 to 6 km integrated tropospheric L3 product based on IASI-A data. To verify their fitness-for-purpose and especially their compliance with the requirements identified for the Global Climate Observing System (GCOS), these ozone datasets were subjected to a comprehensive quality assessment system developed in several heritage projects. The ground-based reference measurements

have thereby been taken from the well-established NDACC, SHADOZ, and WMO GAW ozonesonde and lidar networks. All nadir ozone profile products under study fulfil the GCOS user requirements in terms of observation frequency and horizontal and vertical resolution. Yet all L2 nadir ozone profile observations also show sensitivity outliers in the UTLS and are strongly correlated vertically due to substantial averaging kernel fluctuations that extend far beyond the (typically tropospheric) kernel's 15 km FWHM. The required observation period for climate modelling however is only fully covered

when several instrument time series are combined. Moreover, the nadir ozone profile CRDP typically does not comply with the GCOS user requirements in terms of total uncertainty and decadal drift (except for the UV-VIS L4 dataset). The drift values of the L2 GOME and OMI, the L3 IASI, and the L4 assimilated products are found to be overall insignificant however, and applying appropriate altitude-dependent bias and drift corrections make the data fit for climate and atmospheric composition monitoring and modelling purposes. The nadir ozone profile product validation in terms of several

influence quantities presented in this work correspondingly calls for the introduction of one or more L2 profile flags in addition to those recommended by the data providers, majorly based on a lower DFS threshold.

## Acknowledgements

The reported work was funded by ESA via the CCI – ECV Ozone project, with support from the Belgian Federal Science Policy Office (BELSPO) and ProDEx via project A3C. The ground-based ozonesonde and lidar data used in this publication

were obtained as part of WMO's Global Atmospheric Watch (GAW) programme, including the Network for the Detection of Atmospheric Composition Change (NDACC) and NASA's Southern Hemisphere Additional Ozonesonde programme (SHADOZ), and are publicly available via the NDACC Data Host Facility, the SHADOZ archive, and World Ozone and Ultraviolet Data Center (WOUDC) (see http://www.ndacc.org/, http://croc.gsfc.nasa.gov/shadoz/, and http://www.woudc.org/, respectively). We warmly thank several members of the NDACC ozonesonde and lidar working

groups for fruitful discussions. Lidar operation is funded through national collaborators and we are grateful to the following institutes and their co-workers who contributed to generating these data: CNRS and CNES (Dumont d'Urville station and Observatoire Haute Provence, PI is S. Godin-Beekmann), DWD (Höhenpeißenberg station, PI is H. Claude), RIVM and





NIWA (Lauder station, PI is D. P. J. Swart), NASA/JPL (Mauna Loa Observatory and Table Mountain Facility, PIs are I. S. McDermid and T. Leblanc), and NIES (Tsukuba station, PI is H. Nakane).

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
