# Peer review of "Quality assessment of the Ozone\_cci Climate Research Data Package (release 2017): 2. Ground-based validation of nadir ozone profile data products"

_Atmospheric Measurement Techniques, 2017_

## Referee Comment (RC1) · Anonymous Referee #1 · 17 Mar 2018

The article is in general very comprehensive and detailed. The level of detail is very useful, but so dense it is easy for the reader to get lost. Several tables and figures would benefit greatly by additional labeling to orient the reader. Particularly figures with multiple panels should be labeled with instrument names, reference quality, etc as appropriate so that at a glance the reader can identify what distinguishes one panel from another, and one figure from the next for those that are very similar in appearance.

The ordering of two sections seems illogical. This is based on the concept that the satellite data should be fully discussed before discussing the FRM. Yet a sentence in

the section on screening implies that the screening is not solely based on satellite data quality, but additionally on coincidence opportunities with FRM. If this is the case, the order presented makes sense, but how and why the coincidences with FRM factor into the screening is not motivated or explained.

The section describing the L3 data gridding process is not clear for the novice, and overkill for an expert. Choose your audience, and make adjustments.

Detailed comments follow.

P2, line 16-17: Needs references for SBUV/2, GOME and OMPS.

Section 2: An introduction to the orbital characteristics of the satellite vehicles will be useful for the reader to better understand the later discussions on gridding and co-location of ground data. The beginning of Section 2 might be a good place for such a discussion.

Table 1: Additional columns indicating physical characteristics (vertical units/resolution/range, horizontal grids) of the measurements would be useful. These are all discussed in the text, but Table 1 is an opportunity for easy reference.

P5 line 23: Change 'has to stay' to 'must stay'.

P6 line 14: The A priori for RAL and FORLI are both constructed from the same source as indicated. Are they also both global? It is not clear from this statement.

P7 line 7-9: Are these rejected data included before or after the 'screening' discussed later in the paper?

Section2.4 L3 monthly gridded data: This section is not needed for experts in gridding data, and not helpful to the novice, so it is not clear who the authors are writing to. Figure 1 and this section would benefit for a discussion of the orbital characteristics of the satellite vehicles (either here or at the beginning of Sect 2 as suggested.). Also relate A, B,… and 1,2,3… to the physical items they represent. Refer to the profiles

of the L2 data, and the grid points of L3. If A, B, etc. are the grid points, and 1, 2, 3 are the L2 profiles (and it is not clear that this is the case), is there an advantage to this approach of 4 grid points defining a rectangle, and subdividing the enclosed area, or is it the same as creating a rectangle around a grid point and assigning all profiles within that rectangle to the grid point? The latter seems so much simpler conceptually at least to a novice. What is the subtle missing difference?

P7, line 21: Is there a reference for the GOME/GOME2 convention?

Caption to Fig 1: Why is TM5 assimilation grid referenced here? This figure is used to illustrate the creation of L3 data, not the assimilated L4 data.

Section 2.5: There is a detailed, though difficult, description of how to create the L3 gridded data, but no discussion of how to move to the 2x3 degree L4 grid. This is confusing since Fig 1 refers to the transport model. This needs a little clean up.

P8, line 9: 44 ozone layers in what altitude range?

Section 3 leads with a description of the layout of the next several sections. This is very helpful given the complexity of the paper. But it is unclear why the choice is made to shift at this point to a description of the FRM data before completing the discussion on information content (screening) of the satellite data. Are these not separate concepts? Why not continue with the evaluation of satellite, and complete it before moving onto the description of the FRM? (See also related comment in section 4.1 specifically P12, line 14.)

P8, line 27: 'data harmonization' means different things to different people. Many think of it as bias correcting as a step preliminary to combining data. Perhaps use 'harmonization of data reporting units' to clarify.

P 9 line 17: It would be beneficial to add a line or two about the additional screening criteria used in this study and Hubert et al. 2016 for the ozonesonde data.

P 9 line 26: State measurement variables and resolution for the lidar as a parallel to

the ozonesonde description in the previous paragraphs.

Figure 2: When ozonesonde is removed as an FRM for the level 4 data, there is little left in the tropics to validate L4.

Section 3.3: As previously noted in section 2, knowledge of the orbital characteristics of the satellite vehicles would help in the understanding of the points in this section.

P10, line 13-14: Do you mean within one month (+/- one month) or within relevant month?

Table 4: The column name SPI needs more explanation. How to the numbers in this relate to Figure 1?

Section 4.1 Data Content: It is not clear how a measure of percent of data screened is a measure of data content. It is apparent that the desire is knowledge as to the distribution of the satellite data in latitude and time. It is noted in the description of Figure 3 that for IASI-A, there is little data removed by the screening process leaving a featureless contour implying an even distribution of data. But it is also stated that this is due to prescreening of data before release by the data providers. This technique does not show where the pre-screening removed data. Instead a more relevant measure of content and distribution would be the absolute number of measurements left after screening and its latitudinal and temporal distribution. P12, line 15: How can the latitudinal striping in the UVVis instruments be partially 'due to station overpass' if the screening is solely based on criteria in Table 3? Is screening based solely on data quality, or also on co-location? Additionally what data is in the CCI data release? Only the screened data? Only the screened co-located data?

Figure 3: This figure (and many after) need additional labeling. Label each panel with the satellite name so it is obvious at a glance.

Figure 3, first panel: What causes the gap in the GOME dataset after 2003 in the tropics?

Figure 3, caption: What is meant by 'The decreased BOME-2B data from 2015 onwards justifies additional screening' mean? Are you trying to say that it indicates additional screening?

P14, line 5: change to 'understanding of how the system...'.

Figure 4: Label each panel with the satellite name.

Figure 4, first panel: Why is the area in the tropics of missing data in the GOME panel larger than that in Figure 3?

P17, line 20: From here after there is inconsistent use of BG and of Backus-Gilbert. BG is used extensively in the Figure labels and captions, and occasionally in the text. Introduce the acronym here, then use BG only after.

Figure 5: This figure is difficult to interpret and needs more explanation and labeling. Label the columns with the instrument name. The offset in the second and third rows are labeled identically, but the graphs are different. The caption only states that 'different measures are used'. Are the measures direct and centroid? Differences in the measures for width are clearly indicated. Offset could also be simply added by label and in the caption.

P20, line 22: change 'fiver' to 'five'.

P21, line 18: Here 68% interpercentile spread is used for the first time, but the acronym IP68 is not introduced. Later in the text and graphs there is inconsistent use of the acronym and the full term. Introduce both here, and consistently use the acronym, or the full name in later text.

P21, line 26-27: Should 'vertical averaging smoothing' be 'vertical smoothing'?

Figures 6-10: These are very hard to distinguish when trying to compare the results. Label each figure with the instrument and years (Gome 1996-2010 for example). Also label each panel with the influence quantity. These are stated in the caption, but are

more easily interpreted if the panels are directly labeled.

Figure 11: Label the columns with Latitude and Quarter, and the rows with the instrument name for easy recognition.

Figure 12: Label the columsn with L2, L3 and the rows with the instrument name for easy reference.

P39, line 10: Add the word ozonesonde: '64 ozonesonde stations'.

Figure 13 caption refers to top, middle and bottom instead of left, middle and right. Label each panel.

P40, line 4-5: The Southern mid lats do not look smaller but similar to the tropics in the UTLS.

P40, line 19: Replace 'As for' with 'Similarly to' and remove the word 'now'.

Figure 14, 15: Label each panel with the influence quantity displayed, and 'drift' in the final panel of Fig. 14.

Figure 16. Why is the time series shown for IASI L3, but not for others? It might also be enlightening to show the profile of the L3 drift.

P 51, Acknowledgements: Some of the NDACC PIs listed are retired. It might be of use to additionally include the current persons in these positions as is done for TMF.

---

## Referee Comment (RC2) · Anonymous Referee #2 · 16 Apr 2018

General Comments:

This paper presents a comprehensive and lengthy assessment of the ozone_cci CRDP of 13 nadir ozone profile data products from both UV-VIS and TIR instruments as well as 1 data assimilation product. The evaluation includes data content studies, information content, and validation against ground-based ozonesonde and lidar observations in terms of median relative biases and the IP68 spread as a function of various influence quantities and relative decadal drift. It is a very useful study and its scope is very suitable for publication in AMT. This paper is generally well organized, the methodology

is generally very good and valid, and the results are well described. However, some of the sections are difficult to understand. For example, the section of L3 gridding could be made clearer and simpler and Figure 1 could be removed. The results of the vertical sensitivity are very difficult to interpret, and the derivation of vertical sensitivity could be improved. Also, the abstract does not include main conclusions. In addition, some texts need clarifications. Overall, I think that this paper can be published after addressing the comments mentioned here and specific comments below.

Specific Comments:

1. In abstract, no conclusions are given. So what are the main conclusions of this study? Some of the sentences in the conclusions/discussion sections can be paraphrased here.

2. In the introduction, full instrument names should be specified at their first occurrences.

3. In Sections 2.2 and 2.3, it is useful to mention the unit of the retrieved ozone profile for each algorithm: partial ozone column in DU, average ozone mixing ratio in ppbv, etc.

4. Figure 1 and the text on P7 are difficult to follow and confusing. I guess that grid cells refer to those 1 x 1 boxes, but Figure 1 caption says TM5 assimilation grid. Is TM5 grid 1x1 (looks like it is 2 x 3 based on sect. 2.5)? Also grid cells boundaries typically are not parallel or perpendicular to ground pixel edges as shown in the figure. The naming of grid cells based on pixel corners also makes it more confusing as depending on the pixel size, the entire pixel can lie in one grid cell. Also, what is the size of subpixels and how many subpixels for different instruments. I think that this can be described more clearly and also more concisely. The figure does not really help here and can be removed. Basically, each ground pixel is divided into subpixels (size, #), each subpixel contains the same value and uncertainty, then assign the subpixels to grid cells.

5. Please put table captions before the tables.

6. P10, L17-21, temperature profiles are not required for conversion between number density and average layer VMR. Assuming a layer is well mixed, then average VMR = 1.25 * (partial ozone column in DU) / (pressure difference of the layer in atm). Please see appendix B of Ziemke et al. ("Cloud slicing": A new technique to derive upper tropospheric ozone from satellite measurements, JGR, 106, (D9), P 9853–9867, 2001) for more detail. The partial ozone column is related to number density and altitude difference of the layer.

7. P11, it is not clear about the three numbers in SPI column separated by ":"

8. P12, L18, even if is difficult to know how much IASI data are screened as a function of latitude and time, the data providers should know on average how much data are screened out due to the use of cloud fraction greater than 13%.

9. P15, L15-16, It is not clear why "quite stable in time" reflects the signal degradation correction? Please clarify it.

10. P15, L18-19, It is useful to explain the lower DFS under SAA: shorter wavelengths with weak signals cannot be used due to SAA, thus significantly reducing DFS in the stratosphere

11. P14, equation (6), based on the text, A_F is provided from the FORLI algorithm, so should the defractionalisation operation derive A(m, n) from A_F rather than derive A_F from A(m,n)? I suggest changing this equation to A(m,n) = A_F * . . .

12. P14, L22, first paragraph of P17, Figure 5: It is not easy to under the meaning of vertical sensitivity. Based on the definition on P14, it is an indication of the fraction of the information that is from the data. But on Figure 5 and P17, the vertical sensitivity values peak in the UTLS with a median value of 3, and are often greater than 1 even below 6 km, which does not seem to be consistent with the definition of the fraction of information from the measurement. Also the vertical sensitivity should not peak

in the UTLS, as there is stronger vertical sensitivity in the stratosphere from UV-VIS measurements. Please check DFS at individual layers to make sure this is the case. It seems to me that this concept is not actually a good indicator of the vertical sensitivity or it might depend on how the vertical sensitivity is derived (e.g., from AKM or fractional AKM, what is the unit of state vector, e.g., DU or mixing ratio etc). Based on the definition, when you sum the sensitivity of retrieved ozone at a layer to the perturbations of ozone at all layers, the units of state vector or the weighting of the perturbations at each layer are important. Using mass conserved units like DU or the weighting of perturbations at each layer by a priori error (rather than the a priori or the retrieved profile) might make more sense. Between IASI and UV-VIS retrievals, it is good to convert the AK to the same units and then apply the same concept. Please clarify this on P4. You may also consider the use of DFS at each layer (diagonal elements of AK) normalized to the depth of layer (to account for non-uniform, variable vertical altitude grid) to show the vertical sensitivity, which is straightforward and independent of the retrieval scheme and might be more meaningful.

13. P17, L12-14, it is not clear why the strong sensitivity variability affects vertical smoothing and Eq. 3 introduces a bias. Please make it clearer.

14. P17, L17-18 and also in Fig. 5 caption, it is not clear which is direct and centroid offset between 2nd and 3rd rows. Please make it clear in the figure caption. Also please mentioned the dotted lines in the figure caption.

15. P17, L31, in "decreases first to about 20 km", it seems to me from the figure that the maximum median FWHM is $\sim$20 km, so should it be a smaller number here?

16. P18, first sentence, "slant column density" should not be parallel to "the sensitivity" because the larger slant column density, the smaller the sensitivity from surface to the lower stratosphere. The real reason is because, the larger slant path length or slant column density, the fewer photons penetrating into the troposphere, the smaller the sensitivity in the troposphere, and the larger the resolving length values.

17. P20, L4, suggest changing "quarter" to "season"

18. P20, L24-25, The sentence "This is with the exception of the Metop-B GOME-2 and IASI instruments however, that have not been used for drift studies" is difficult to understand. Suggest changing to "So Metop-B GOME-2 and IASI instruments are excluded for drift studies" and moved it after "for this drift assessment"

19. P20, L29, in addition to latitude and season, the influence quantity of time should be added.

20. P21, the sentence above Eq 9 is difficult to read. Suggest changing to "Yet both approaches introduce similar spatial and temporal representativeness errors into the difference statistics because taking (monthly) averages as a bias estimator âŇľ△xâŇł yields comparable outcomes: "

21. P22, L15, is the ex-ante uncertainty from the retrievals for random noise errors or for both random noise errors and smoothing errors? As the averaging kernels are applied to reference data to remove smoothing errors, ex-ante uncertainty of random noise errors should be shown here. Please clarify this.

22. P22, L26, suggest adding "because the retrievals only include random noise errors and smoothing errors in the ex-ante uncertainty" after the "nadir ozone profile products"

23. P22, L26, OMI is not an exception in that the total satellite measurement uncertainty is underestimated because the ex-ante uncertainty should be compared to comparison spread or the quadratic sum of comparison bias and spread rather than the comparison bias only.

24. P23, L1, it is not generally true that there are smaller biases for larger total ozone columns based on the figures as the biases often increases when the total ozone increases from 300-400 to 400-500 or 500-600 DU (very clearly for GOME and OMI retrievals).

25. P23, L3-4, the relationship between SZA/DFS and the biases are altitudedependent. From the figures, the biases are typically smaller at larger SZAs/DFS in the troposphere, but are larger at larger SZAs/DFS. Larger SZAs typically lead to larger total DFS due to the increase of DFS in the stratosphere and often lead to smaller DFS in the troposphere due to reduced photon penetration. Smaller biases in the troposphere at larger SZAs/DFS could be due to the reduced retrieval sensitivity in the troposphere (i.e., retrievals are closer to the a priori). So the causal relationship is not as straightforward as larger DFS means better retrieval sensitivity and therefore smaller biases.

26. P33, L11, based on figures, the spread is not always enlarged in the L3 comparison. Instead, the spread is typically significantly reduced below ∼6 km. This should be mentioned and explained.

27. P34, L1, says "GOME L3 data show a negative above-tropopause bias of 5-10%". But based on the first panel of Fig. 11, I see mostly positive biases above 100 hPa, especially with large positive biases of 20% around 70 hPa and positive biases of 40% around 8 hPa. Please clarify this.

28. P36, L11, again the spread values for the L3 comparison can be smaller below ∼6 km, which should be mentioned.

29. P40, L7-8, it is useful to explain to the readers why there is stronger tropospheric reduction to 20% and why the drift is small (e.g., due to bias correction).

30. P45, last line, the 30% negative should be 30% positive in Antarctica as shown in last panel of Fig. 16. Also please change Table 5 correspondingly.

31. In table 5, suggest changing "Vertical resolution (6 km to troposphere)" to "Vertical grid/ resolution", changing to "115 km2", "230 by 345 km2", "12 km2", "115 km2"

32. In Figs. 6-10, 14-15, change second bracket from "[" to "]" in Fig. captions

Technical Comments:

1. P2 last line, and P8 L25, change "Keppens et al., 2015" to "Keppens et al. (2015)"

2. P4, L6, change to "time series"

3. P4, L13, P5, L15, P6, L13, change "priori" to "a priori"

4. P9, L14, change "beyond" to "above"

5. P11, change "prior" or "a-priori" to "a priori"

6. P11, suggest changing "averaging kernel smoothing of the FRM" to "smoothing the FRM with averaging kernel"

7. P12, L6, suggest changing "relative amount" to "percentage"

8. P12, L18, suggest changing "delivery. E.g." to "delivery, i.e., "

9. P13, L10, change "prior" or "a-priori" to "a priori"

10. P14, L5-6, change to "help understand"

11. P14, L16 & L26, change "prior" or "a-priori" to "a priori"

12. P21, L13, suggest changing "Thanks to" to "Due to" to make it formal.

13. P21, L21, change "vertical averaging smoothing of ground-based reference data" to "vertical smoothing of ground-based reference data with averaging kernels"

14. P21, L29-30, change "smoothing difference error" to "retrieval smoothing error" and "Tropics" to "tropics"

15. P23, L6, change to "instruments except for GOME-2B"

16. P33, L12, change to "lack of"

17. P48, L10, change to "7 km by 7 km" or "7 by 7 km2"

18. P49, L11, change to "equivalent to"

19. P49, L12, suggest changing "as an alternative" to "along with"

---

## Author Comment (AC1) · 14 May 2018

The response to the Referees shall be structured in a clear and easy-to-follow sequence: (1) comments from Referees, (2) author's response, (3) author's changes in manuscript.

Anonymous Referee #1

General comments:

The article is in general very comprehensive and detailed. The level of detail is very useful, but so dense it is easy for the reader to get lost. Several tables and figures would benefit greatly by additional labelling to orient the reader. Particularly figures with multiple panels should be labelled with instrument names, reference quality, etc. as appropriate so that at a glance the reader can identify what distinguishes one panel from another and one figure from the next for those that are very similar in appearance.

The ordering of two sections seems illogical. This is based on the concept that the satellite data should be fully discussed before discussing the FRM. Yet a sentence in the section on screening implies that the screening is not solely based on satellite data quality, but additionally on coincidence opportunities with FRM. If this is the case, the order presented makes sense, but how and why the coincidences with FRM factor into the screening is not motivated or explained.

The section describing the L3 data gridding process is not clear for the novice, and overkill for an expert. Choose your audience, and make adjustments.

Detailed comments:

(1) P2, line 16-17: Needs references for SBUV/2, GOME and OMPS.
(2) The authors agree that references are required here. References to (Heath et al., 1975), (Burrows et al., 1999), and (Flynn et al., 2006) have been added in the text and in the reference list.
(3) The second sentence of the introduction now reads as follows: "Atmospheric ozone concentration profiles have been retrieved from solar backscatter ultraviolet radiation measurements by nadir viewing satellite spectrometers since the 1960s, starting with the USSR Kosmos missions in 1964-1965 (Iozenas et al., 1969) and NASA's Orbiting Geophysical Observatory in 1967-1969 (Anderson et al., 1969) and Backscatter Ultraviolet (BUV) on Nimbus 4 in 1970-1975 (Heath et al., 1973), and continuing with the Solar SBUV(/2) series after 1978 (Heath et al., 1975), the Global Ozone Monitoring Experiment (GOME) family of sensors since 1995 (Burrows et al., 1999), and the Ozone Mapping Profiler Suite (OMPS-nadir) series started in 2011 (Flynn et al., 2006)."

(1) Section 2: An introduction to the orbital characteristics of the satellite vehicles will be useful for the reader to better understand the later discussions on gridding and colocation of ground data. The beginning of Section 2 might be a good place for such a discussion. Section 3.3: As previously noted in section 2, knowledge of the orbital characteristics of the satellite vehicles would help in the understanding of the points in this section.
(2) The authors agree that some knowledge on the orbital characteristics of the satellites might be of help to the user. This information has been added in Section 2.1. However, regarding the co-location criteria that are used, knowledge of the LST (as indicated in Table 5) is sufficient. Section 3.3 has therefore been slightly extended with reference to the orbital characteristics mentioned in Section 2.1.
(3) Section 2.1 has been extended with the following sentence: "All instruments listed in Table 1 are on satellite vehicles with a sun-synchronous low-earth-orbit, resulting in fixed local solar overpass times (also see Section 3.3)." and Section 3.3 has been slightly changed with reference to the first addition: "These time windows are chosen to generally have at least one satellite co-location with each FRM, given the satellite's fixed local solar time (LST, also see Section 2.1) and the fact that ozonesondes are typically launched around local noon, while lidar measurements are taken during the night."

(1) Table 1: Additional columns indicating physical characteristics (vertical units/resolution/range, horizontal grids) of the measurements would be useful. These are all discussed in the text, but Table 1 is an opportunity for easy reference.

(2) The authors agree that such overview would be helpful in understanding all CRDP products, and have added two columns to Table 1 and extended its caption.

(3) Two columns have been added to Table 1, and its caption has been extended as follows: "The products' vertical range (with number of levels or layers between brackets) and original units are added in the last two columns."

(1) P5 line 23: Change 'has to stay' to 'must stay'.

(2) Agree

(3) On page 5 line 23 "has to stay" has been replaced by "must stay"

(1) P6 line 14: The A priori for RAL and FORLI are both constructed from the same source as indicated. Are they also both global? It is not clear from this statement.

(2) The authors agree that this statement is not fully clear and have modified the text to make the similarities and distinctions between RAL and FORLI prior data clear.

(3) P6 line 13-15 has been updated as follows: "The a priori information used in the FORLI algorithm consists of a single global ozone prior profile. The prior variance-covariance matrix is built from the McPeters-Labow-Logan climatology (McPeters et al., 2007), as for RAL."

(1) P7 line 7-9: Are these rejected data included before or after the 'screening' discussed later in the paper?

(2) This question is not fully clear to the authors. All data screening is discussed in Section 2, summarised in Table 3, and studied in Section 4.1. The relative screening numbers in Figure 3 refer to all screening as discussed in Section 2 relative to the total number of retrieved profiles.

(3) No further action has been taken.

(1) Section2.4 L3 monthly gridded data: This section is not needed for experts in gridding data, and not helpful to the novice, so it is not clear who the authors are writing to. Figure 1 and this section would benefit for a discussion of the orbital characteristics of the satellite vehicles (either here or at the beginning of Sect 2 as suggested.). Also relate A, B,. . . and 1,2,3. . . to the physical items they represent. Refer to the profiles of the L2 data, and the grid points of L3. If A, B, etc. are the grid points, and 1, 2, 3 are the L2 profiles (and it is not clear that this is the case), is there an advantage to this approach of 4 grid points defining a rectangle, and subdividing the enclosed area, or is it the same as creating a rectangle around a grid point and assigning all profiles within that rectangle to the grid point? The latter seems so much simpler conceptually at least to a novice. What is the subtle missing difference?

P7, line 21: Is there a reference for the GOME/GOME2 convention?

Caption to Fig 1: Why is TM5 assimilation grid referenced here? This figure is used to illustrate the creation of L3 data, not the assimilated L4 data.

Section 2.5: There is a detailed, though difficult, description of how to create the L3 gridded data, but no discussion of how to move to the 2x3 degree L4 grid. This is confusing since Fig 1 refers to the transport model. This needs a little clean up.

(2) In the context of the KNMI L3 product, a pixel refers to a satellite measurement, while a lat-lon grid cell refers to the regular 1x1 degree latitude-longitude grid for which the mean and standard deviation are calculated. Each pixel is divided into 25 subpixels, which are assigned to the grid cell containing the subpixel. The mean and standard deviation for the grid cell are calculated according to the equations given in the text. The authors agree with both reviewers that the text on which subpixels are assigned to which grid cell is unclear and the text of section 2.4 and the caption of figure 1, which is preferably maintained, have been updated accordingly.

(3) The paragraph before Eq. (1) has been replaced by the following: "Monthly-averaged L3 profile products are produced from the filtered RAL v2.14 GOME, GOME-2A, SCIAMACHY, and OMI data by the Royal Meteorological Institute of the Netherlands (KNMI). Version 0004 of the KNMI L3 products has been used in this work (see Table 1). The KNMI level-3 data consist of monthly ozone

profile averages, also on a one-by-one degree latitude-longitude grid, containing 19 layers between 20 fixed pressure levels at each grid-point. The algorithm that calculates the monthly-averaged ozone fields assumes that the L2 satellite ground pixel vertices (labelled ABCD) are ordered as indicated in Figure 1. Each pixel's across-track direction is defined by the lines AD and BC, while the along-track direction is defined by the lines AB and DC. The satellite pixel is divided into 25 subpixels, five in the along-track direction and five in the cross-track direction, and each subpixel is assigned to the L3 grid cell (the boundaries are indicated with the dashed lines in Figure 1) containing the subpixel. The subpixel values $x_i$ are weighted by the square inverse of their uncertainties ($\sigma_i^{-2}$), so the weighted mean grid cell value $x_c$ and the corresponding standard deviation $\sigma_c$ are given by"
The caption of Figure 1 now reads as follows: "Figure 1: A L2 satellite pixel ABCD is divided into subpixels (diamonds 1 to 7). Each subpixel is assigned to a L3 grid cell (indicated with the dashed boundaries) and the average and standard deviation are calculated (see text). In this example, subpixels 1-3 would be assigned to the lower-right grid cell and subpixels 4-7 would be assigned to the lower-left grid cell. The satellite pixel ABCD may have any orientation with respect to the L3 grid."

(1) P8, line 9: 44 ozone layers in what altitude range?
(2) The authors agree that this was not clear.
(3) "surface to 1 hPa" has been added as a clarification between brackets.

(1) P8, line 27: 'data harmonization' means different things to different people. Many think of it as bias correcting as a step preliminary to combining data. Perhaps use 'harmonization of data reporting units' to clarify.
(2) Thanks for pointing out this ambiguity. The authors have changed the text to clarify.
(3) "data harmonisation" has been replaced by "harmonisation of data representation in terms of vertical sampling and units"

(1) P 9 line 17: It would be beneficial to add a line or two about the additional screening criteria used in this study and Hubert et al. 2016 for the ozonesonde data.
(2) The authors agree. A sentence has been added after the reference to Hubert et al.
(3) Added sentence: "Entire FRM profiles are discarded when more than half of the levels are tagged bad or when less than 30 levels are tagged good."

(1) P 9 line 26: State measurement variables and resolution for the lidar as a parallel to the ozonesonde description in the previous paragraphs.
(2) The authors believe that the information requested by the reviewer was already available at the end of the paragraph under consideration (thus not above).
(3) No changes have been made.

(1) Figure 2: When ozonesonde is removed as an FRM for the level 4 data, there is little left in the tropics to validate L4.
(2) The authors are somewhat confused by this statement. Nowhere it is stated that ozonesondes are not used for L4 validation. On the contrary, it is stated in the text that "For the six-hourly assimilated L4 data, the unique temporally closest ground-based reference measurement is always less than 3 hours away." Meaning that there is a co-location for each FRM.
(3) No action has been taken.

(1) P10, line 13-14: Do you mean within one month (+/- one month) or within relevant month?
(2) Thanks for pointing out this unclarity. The text has been updated to make elucidate this statement.
(3) "All FRM within this grid cell within one month are included in the analyses for the L3 comparisons." has been replaced by "All FRM within this grid cell and within the relevant month are included in the analyses for the L3 comparisons."

(1) Table 4: The column name SPI needs more explanation. How to the numbers in this relate to Figure 1?

(2) The authors admit that the meaning of the SPI values had erroneously not been mentioned in the text. Therefore the text has been extended with reference to Table 4. This however does not immediately relate to Figure 1, as should now be clear from the updated text.

(3) After the first reference to Table 4 in Section 3.3, the following sentence has been added: "The possible satellite pixel index (SPI) values within each cross-track scan and the resulting number of pixels per scan are provided for each instrument in Table 4 (taking into account pixel co-adding, see Section 2)." The notation of the possible SPI in Table 4 has been changed from X:X:X (start, step, end) to X,X,X,…,X (start, start+1, start+2,…,end).

(1) Section 3 leads with a description of the layout of the next several sections. This is very helpful given the complexity of the paper. But it is unclear why the choice is made to shift at this point to a description of the FRM data before completing the discussion on information content (screening) of the satellite data. Are these not separate concepts? Why not continue with the evaluation of satellite, and complete it before moving onto the description of the FRM? (See also related comment in section 4.1 specifically P12, line 14.)

(2) The authors agree that this approach might be somewhat misleading as the pre-processing of the data might not have been fully clear from the text: The data and information content studies are performed on ground station overpass data, i.e. satellite pixels must be within a 300 km radius from a FRM station. Section 3.1 has been rewritten to make this clear and motivate the subsequent ordering of sections.

(3) The end of Section 3.1 has been replaced by the following: "The satellite data collection and post-processing (mainly L2 profile screening) is described by the previous section. The L2 datasets have however been reduced to 300 km ground station overpass datasets for the quality assessment in this work, in order to reduce the total amount of data processing (i.e. satellite pixels must be within a 300 km radius from a FRM station). The FRM data selection, co-located datasets study, and data harmonisation are therefore included as the successive subsections within this section. The satellite data content studies and information content studies are discussed in the next Section 4. These include statistics on the L2 station overpass data screening and spatiotemporal coverage, and averaging kernel-based information content measures, respectively. The comparative analysis with both spatially and temporally co-located FRM data follows later in Section 5."

(1) Section 4.1 Data Content: It is not clear how a measure of percent of data screened is a measure of data content. It is apparent that the desire is knowledge as to the distribution of the satellite data in latitude and time. It is noted in the description of Figure 3 that for IASI-A, there is little data removed by the screening process leaving a featureless contour implying an even distribution of data. But it is also stated that this is due to pre-screening of data before release by the data providers. This technique does not show where the pre-screening removed data. Instead a more relevant measure of content and distribution would be the absolute number of measurements left after screening and its latitudinal and temporal distribution.

(2) In line with the previous comment and corresponding answer, the authors believe that the presentation of percentages is now better motivated: As station overpass data are studied, absolute numbers would be misleading and even more stress the spatial selection of the data. Figure 3 mainly wants to show were L2 data can be found and what the impact is of the screening suggested by the data providers. This has been made more clear in the first paragraph of Section 4.1.

(3) The beginning of Section 4.1 has been updated as follows: "The nadir ozone profile CRDP L2 data content study focuses on the spatiotemporal distribution and the effect of screening of the retrieved satellite profiles in the first place, next to the regular file structure, file content, and value checks for the quantities of highest relevance (also see Table 3). Figure 3 displays the latitude-time distribution per 10° latitude band and per month of the percentage of screened profiles for all nadir profile L2 station overpass (300 km) datasets (except for IASI on Metop-B)."

(1) P12, line 15: How can the latitudinal striping in the UV-Vis instruments be partially 'due to station overpass' if the screening is solely based on criteria in Table 3? Is screening based solely on data quality, or also on co-location? Additionally what data is in the CCI data release? Only the screened data? Only the screened co-located data?

(2) The authors agree that this was unclear from the original text, but believe that this is now clarified by the previous two answers on the use of 300 km station overpass data. It should be clear that the full L2 datasets are available in the CCI data release, without screening and without any co-location.
(3) No additional changes have been made to the text.

(1) Figure 3, first panel: What causes the gap in the GOME dataset after 2003 in the tropics?
(2) As ground stations are located near the South-Atlantic Anomaly (SAA) and a quite severe GOME data screening has to be applied, no (near) SAA data are left. This has now been made clear in the text.
(3) "The lack of GOME data in the southern mid-latitudes from 2003 onwards is due to severe screening of L2 overpass data for ground stations that are all located near the South-Atlantic anomaly (SAA)." has been added.

(1) Figure 3, caption: What is meant by 'The decreased GOME-2B data from 2015 onwards justifies additional screening' mean? Are you trying to say that it indicates additional screening?
(2) The authors agree that this statement is misleading. The caption has therefore been brought in line with the main text.
(3) "justifies additional screening" has been replaced by "points at a retrieval issue"

(1) P14, line 5: change to 'understanding of how the system'
(2) The reviewer's proposal for improving the readability has been followed, yet somewhat differently, in agreement with the suggestion by the second reviewer.
(3) "understanding how the system" has been replaced by "understand how the system"

(1) Figure 4, first panel: Why is the area in the tropics of missing data in the GOME panel larger than that in Figure 3?
(2) If all data are screened (100 % values in Fig. 3) than the DFS and other information content values are empty.
(3) No changes made.

(1) P17, line 20: From here after there is inconsistent use of BG and of Backus-Gilbert. BG is used extensively in the Figure labels and captions, and occasionally in the text. Introduce the acronym here, then use BG only after.
(2) Thanks for pointing this out. The acronym as been added.
(3) P17, line 20 "Backus-Gilbert spread" is replaced by "Backus-Gilbert (BG) spread"

(1) Figure 5: The offset in the second and third rows are labelled identically, but the graphs are different. The caption only states that 'different measures are used'. Are the measures direct and centroid? Differences in the measures for width are clearly indicated. Offset could also be simply added by label and in the caption.
(2) The authors agree that offset and spread indications in Figure 5 can be improved. The caption of Figure 5 has been updated accordingly.
(3) The caption of Figure 5 has been updated as follows: "Global GOME-2A (left) and IASI-A (right) information content in terms of vertical sensitivity, retrieval offset (in km), and averaging kernel width (in km) and their dependence on DFS, SZA, or thermal contrast (TC). Black dashed lines represent median values, while out-of-range profiles are plotted in magenta. Different measures are used for the offset and kernel width in the second and third rows, which include the centroid offset and Backus-Gilbert spread, and the direct offset and FWHM, respectively. Plot titles provide the absolute and relative amounts of profiles after screening, and the number of ground-based overpass stations."

(1) P20, line 22: change 'fiver' to 'five'.
(2) Thank you for spotting this typo; the text has been corrected.
(3) 'fiver' has been changed to 'five'

(1) P21, line 18: Here 68% interpercentile spread is used for the first time, but the acronym IP68 is not introduced. Later in the text and graphs there is inconsistent use of the acronym and the full term. Introduce both here, and consistently use the acronym or the full name in later text.
(2) Actually the 68% interpercentile spread and its acronym are first introduced on page 20, line 8, as Q84-Q16.
(3) The authors have added "68 % interpercentile" explicitly to page 20, line 8 to avoid the impression of inconsistent use of terms and acronyms.

(1) P21, line 26-27: Should 'vertical averaging smoothing' be 'vertical smoothing'?
(2) The authors intended either "vertical smoothing" or "vertical averaging kernel smoothing". The latter has been chosen here. Yet in agreement with a comment by the second reviewer, the phrasing has been changed.
(3) "vertical averaging smoothing" is replaced by "vertical smoothing of ground-based reference data with averaging kernels"

(1) P39, line 10: Add the word ozonesonde: '64 ozonesonde stations'.
(2) The authors have followed the suggestion by the reviewer.
(3) "64 stations" has been changed into "64 ozonesonde stations"

(1) Figure 13 caption refers to top, middle and bottom instead of left, middle and right. Label each panel.
(2) The caption of Figure 13 has been written with the final mark-up of the paper in mind, i.e. with the three plots combined into a column (not a row).
(3) No action has been taken.

(1) P40, line 4-5: The Southern mid lats do not look smaller but similar to the tropics in the UTLS.
(2) It was the intention of the authors to state that the bias indeed looks similar, but has only positive values. This observation has been made more explicit in the update of the text.
(3) "but smaller" has been replaced by "but only positive"

(1) P40, line 19: Replace 'As for' with 'Similarly to' and remove the word 'now'.
(2) The authors have adopted the suggestions by the reviewer to increase readability.
(3) The sentence referred to now reads as follows "Similarly to the L2 RAL v2.14 UV-VIS retrievals, Figure 14 and Figure 15 contain…"

(1) Figure 3: This figure (and many after) need additional labelling. Label each panel with the satellite name so it is obvious at a glance.
Figure 4: Label each panel with the satellite name.
Figure 5: This figure is difficult to interpret and needs more explanation and labelling. Label the columns with the instrument name.
Figures 6-10: These are very hard to distinguish when trying to compare the results. Label each figure with the instrument and years (GOME 1996-2010 for example). Also label each panel with the influence quantity. These are stated in the caption, but are more easily interpreted if the panels are directly labelled.
Figure 11: Label the columns with Latitude and Quarter, and the rows with the instrument name for easy recognition.
Figure 12: Label the columns with L2, L3 and the rows with the instrument name for easy reference.
Figure 14, 15: Label each panel with the influence quantity displayed, and 'drift' in the final panel of Fig. 14.
(2) The authors agree that readability and interpretation of graphs can be improved upon insertion of satellite instrument and influence quantity labels on the relevant plots.
(3) All plots have been updated with the requested labels.

(1) Figure 16: Why is the time series shown for IASI L3, but not for others? It might also be enlightening to show the profile of the L3 drift.

(2) For the FORLI IASI product only tropospheric column L3 data are available, so only columnar values can be shown. Such values however allow for a more easy time series representation. Vertical drift profiles are not possible, and have been replaced by a trend line.

(3) No further action has been taken.

(1) P 51, Acknowledgements: Some of the NDACC PIs listed are retired. It might be of use to additionally include the current persons in these positions as is done for TMF.

(2) The authors acknowledge that some PI references require updating. The names of R. Querel and R. C. Schnell have been added.

(3) The lidar PI acknowledgement now reads as follows: "CNRS and CNES (Dumont d'Urville station and Observatoire Haute Provence, PI is S. Godin-Beekmann), DWD (Höhenpeißenberg station, PI is H. Claude), RIVM and NIWA (Lauder station, PIs is are D. P. J. Swart and R. Querel), NASA/JPL (Mauna Loa Observatory and Table Mountain Facility, PIs are I. S. McDermidR. C. Schnell and T. Leblanc), and NIES (Tsukuba station, PI is H. Nakane)"

---

## Author Comment (AC2) · 14 May 2018

The response to the Referees shall be structured in a clear and easy-to-follow sequence: (1) comments from Referees, (2) author's response, (3) author's changes in manuscript.

Anonymous Referee #2

General Comments:

This paper presents a comprehensive and lengthy assessment of the Ozone_cci CRDP of 13 nadir ozone profile data products from both UV-VIS and TIR instruments as well as 1 data assimilation product. The evaluation includes data content studies, information content, and validation against ground-based ozonesonde and lidar observations in terms of median relative biases and the IP68 spread as a function of various influence quantities and relative decadal drift. It is a very useful study and its scope is very suitable for publication in AMT. This paper is generally well organized, the methodology is generally very good and valid, and the results are well described. However, some of the sections are difficult to understand. For example, the section of L3 gridding could be made clearer and simpler and Figure 1 could be removed. The results of the vertical sensitivity are very difficult to interpret, and the derivation of vertical sensitivity could be improved. Also, the abstract does not include main conclusions. In addition, some texts need clarifications. Overall, I think that this paper can be published after addressing the comments mentioned here and specific comments below.

Specific Comments:

(1) 1. In abstract, no conclusions are given. So what are the main conclusions of this study? Some of the sentences in the conclusions/discussion sections can be paraphrased here.
(2) The authors agree that some of the major conclusions should be provided in the abstract as well, and have extended the abstract accordingly.
(3) After "(WMO GAW)." the abstract has been extended as follows: "The nadir ozone profile CRDP quality assessment reveals that all nadir ozone profile products under study fulfil the GCOS user requirements in terms of observation frequency and horizontal and vertical resolution. Yet all L2 observations also show sensitivity outliers in the UTLS and are strongly correlated vertically due to substantial averaging kernel fluctuations that extend far beyond the kernel's 15 km FWHM. The CRDP typically does not comply with the GCOS user requirements in terms of total uncertainty and decadal drift, except for the UV-VIS L4 dataset. The drift values of the L2 GOME and OMI, the L3 IASI, and the L4 assimilated products are found to be overall insignificant however ,and applying appropriate altitude-dependent bias and drift corrections make the data fit for climate and atmospheric composition monitoring and modelling purposes. Dependence of the Ozone_cci data quality on major influence quantities – resulting in data screening suggestions to users – and perspectives for the Copernicus Sentinel missions are additionally discussed."

2. In the introduction, full instrument names should be specified at their first occurrences.

(1) 3. In Sections 2.2 and 2.3, it is useful to mention the unit of the retrieved ozone profile for each algorithm: partial ozone column in DU, average ozone mixing ratio in ppbv, etc.
(2) In reply to a similar comment by reviewer 1, the authors have added two columns to Table 1 and extended its caption.
(3) Two columns have been added to Table 1, and its caption has been extended as follows: "The products' vertical range (with number of levels or layers between brackets) and original units are added in the last two columns."

(1) 4. Figure 1 and the text on P7 are difficult to follow and confusing. I guess that grid cells refer to those 1 x 1 boxes, but Figure 1 caption says TM5 assimilation grid. Is TM5 grid 1x1 (looks like it is 2 x 3 based on sect. 2.5)? Also grid cells boundaries typically are not parallel or perpendicular to ground pixel edges as shown in the figure. The naming of grid cells based on pixel corners also makes it more confusing as depending on the pixel size, the entire pixel can lie in one grid cell. Also, what is the size

of subpixels and how many subpixels for different instruments? I think that this can be described more clearly and also more concisely. The figure does not really help here and can be removed. Basically, each ground pixel is divided into subpixels (size, #), each subpixel contains the same value and uncertainty, then assign the subpixels to grid cells.

(2) In the context of the KNMI L3 product, a pixel refers to a satellite measurement, while a lat-lon grid cell refers to the regular 1x1 degree latitude-longitude grid for which the mean and standard deviation are calculated. Each pixel is divided into 25 subpixels, which are assigned to the grid cell containing the subpixel. The mean and standard deviation for the grid cell are calculated according to the equations given in the text. The authors agree with both reviewers that the text on which subpixels are assigned to which grid cell is unclear and the text of section 2.4 and the caption of figure 1, which is preferably maintained, have been updated accordingly.

(3) The paragraph before Eq. (1) has been replaced by the following: "Monthly-averaged L3 profile products are produced from the filtered RAL v2.14 GOME, GOME-2A, SCIAMACHY, and OMI data by the Royal Meteorological Institute of the Netherlands (KNMI). Version 0004 of the KNMI L3 products has been used in this work (see Table 1). The KNMI level-3 data consist of monthly ozone profile averages, also on a one-by-one degree latitude-longitude grid, containing 19 layers between 20 fixed pressure levels at each grid-point. The algorithm that calculates the monthly-averaged ozone fields assumes that the L2 satellite ground pixel vertices (labelled ABCD) are ordered as indicated in Figure 1. Each pixel's across-track direction is defined by the lines AD and BC, while the along-track direction is defined by the lines AB and DC. The satellite pixel is divided into 25 subpixels, five in the along-track direction and five in the cross-track direction, and each subpixel is assigned to the L3 grid cell (the boundaries are indicated with the dashed lines in Figure 1) containing the subpixel. The subpixel values $x_i$ are weighted by the square inverse of their uncertainties ($\sigma_i^{(-2)}$), so the weighted mean grid cell value $x_c$ and the corresponding standard deviation $\sigma_c$ are given by"

The caption of Figure 1 now reads as follows: "Figure 1: A L2 satellite pixel ABCD is divided into subpixels (diamonds 1 to 7). Each subpixel is assigned to a L3 grid cell (indicated with the dashed boundaries) and the average and standard deviation are calculated (see text). In this example, subpixels 1-3 would be assigned to the lower-right grid cell and subpixels 4-7 would be assigned to the lower-left grid cell. The satellite pixel ABCD may have any orientation with respect to the L3 grid."

(1) 5. Please put table captions before the tables.
(2) This is done automatically in the final publication mark-up.
(3) Action is taken upon final submission of the manuscript.

(1) 6. P10, L17-21, temperature profiles are not required for conversion between number density and average layer VMR. Assuming a layer is well mixed, then average VMR = 1.25 * (partial ozone column in DU) / (pressure difference of the layer in atm). Please see appendix B of Ziemke et al. ("Cloud slicing": A new technique to derive upper tropospheric ozone from satellite measurements, JGR, 106, (D9), P 9853–9867, 2001) for more detail. The partial ozone column is related to number density and altitude difference of the layer.
(2) In these lines the level-related VMR value is intended, not the layer-average VMR. In that case the temperature profile is required. The authors are familiar with the literature referred to, but thank the reviewer for pointing this out again.
(3) No changes have been made.

(1) 7. P11, it is not clear about the three numbers in SPI column separated by ":"
(2) The authors admit that the meaning of the SPI values had erroneously not been mentioned in the text. Therefore the text has been extended with reference to Table 4, also in agreement with a comment by reviewer 1. The notation in Table 4 has moreover been changed.
(3) After the first reference to Table 4 in Section 3.3, the following sentence has been added: "The possible satellite pixel index (SPI) values within each cross-track scan and the resulting number of pixels per scan are provided for each instrument in Table 4 (taking into account pixel co-adding, see Section 2)." The notation of the possible SPI in Table 4 has been changed from X:X:X (start, step, end) to X,X,X,…,X (start, start+1, start+2,…,end).

(1) 8. P12, L18, even if is difficult to know how much IASI data are screened as a function of latitude and time, the data providers should know on average how much data are screened out due to the use of cloud fraction greater than 13%.
(2) This section has been somewhat modified according to suggestions by the first reviewer. The authors agree that IASI's pre-screening does not allow a full assessment of the data (screening) distribution. Yet cloud screening has globally a similar effect on both the UV-VIS and TIR data. This information has been added to the text.
(3) The sentence on the IASI screening has been extended as follows: "The IASI screening on the other hand appears very low, yet this is due to the pre-screening by the product providers before data delivery, i.e., mainly the IASI cloud screening (if the fraction is higher than 13 %) cannot be observed from the plots, but is roughly of the same order as the UV-VIS data screening."

(1) 9. P15, L15-16, It is not clear why "quite stable in time" reflects the signal degradation correction? Please clarify it.
(2) The authors considered it quite obvious that a satellite signal degradation correction results in maintaining the signal-to-noise ratio close to its initial level and hence in a quite stable DFS upon nadir ozone profile retrieval. The DFS reduces in correlation with the signal if no signal degradation correction is applied. Yet for clarity, the authors have added this explanation in the text.
(3) "This correction maintains the instrument's signal-to-noise ratio close to its initial level and hence reduces the effect of the instrument degradation on the retrieval's DFS." has been added.

(1) 10. P15, L18-19, It is useful to explain the lower DFS under SAA: shorter wavelengths with weak signals cannot be used due to SAA, thus significantly reducing DFS in the stratosphere.
(2) The authors agree that some clarification would be helpful.
(3) "due to stratospheric intrusion of high-energetic particles (the tropospheric DFS is mostly maintained)" has been added.

(1) 11. P14, equation (6), based on the text, A_F is provided from the FORLI algorithm, so should the defractionalisation operation derive A(m, n) from A_F rather than derive A_F from A(m,n)? I suggest changing this equation to A(m,n) = A_F * . . .
(2) The reviewer might have misunderstood: the absolute averaging kernel A is given in the FORLI data, yet this AK is calculated by the data providers from the fractional AK that results from the retrieval. This calculation is done using the prior profile, hence in this work we also use the prior profile to invert this operation and again obtain the fractional kernel matrix. The authors believe that this is clearly stated in the text already: "These fractional matrices are made unit-dependent by use of the prior profile before saving into the data files, allowing for more straightforward application (e.g. for vertical smoothing operations) by data users. For the information content studies presented here, this defractionalisation operation therefore has to be inverted"
(3) No further changes have been made.

(1) 12. P14, L22, and first paragraph of P17, Figure 5: It is not easy to understand the meaning of vertical sensitivity. Based on the definition on P14, it is an indication of the fraction of the information that is from the data. But on Figure 5 and P17, the vertical sensitivity values peak in the UTLS with a median value of 3, and are often greater than 1 even below 6 km, which does not seem to be consistent with the definition of the fraction of information from the measurement. Also the vertical sensitivity should not peak in the UTLS, as there is stronger vertical sensitivity in the stratosphere from UV-VIS measurements. Please check DFS at individual layers to make sure this is the case. It seems to me that this concept is not actually a good indicator of the vertical sensitivity or it might depend on how the vertical sensitivity is derived (e.g., from AKM or fractional AKM, what is the unit of state vector, e.g., DU or mixing ratio etc.). Based on the definition, when you sum the sensitivity of retrieved ozone at a layer to the perturbations of ozone at all layers, the units of state vector or the weighting of the perturbations at each layer are important. Using mass conserved units like DU or the weighting of perturbations at each layer by a priori error (rather than the a priori or the retrieved profile) might make more sense. Between IASI and UV-VIS retrievals, it is good to convert

the AK to the same units and then apply the same concept. Please clarify this on P4. You may also consider the use of DFS at each layer (diagonal elements of AK) normalized to the depth of layer (to account for non-uniform, variable vertical altitude grid) to show the vertical sensitivity, which is straightforward and independent of the retrieval scheme and might be more meaningful.
(2) The authors acknowledge that a fraction-like measure below zero or above one makes little sense. This sensitivity interpretation by Rodgers however is only a rough approximation (as stated in the text). In practice, the sensitivity at a specific retrieval level can nevertheless be negative or exceed unity (over-sensitivity) due to kernel fluctuations and correlations between adjacent retrieval levels, as reflected in the kernel width. This has now been made clear in the text. The over-sensitivity in the UTLS hence is no surprise, as especially the UV-VIS nadir ozone profile retrieval shows difficulties around the tropopause and below. The normalized layer-DFS (that is known to the authors) certainly is a useful quantity that has been considered in this work, but has been found to add little to the combined DFS and vertical sensitivity discussion. It has therefore not been additionally discussed. Both (overall) DFS and vertical sensitivity calculated from fractional averaging kernel matrices are fully unit-independent measures (the authors refer to Keppens at al., 2015, for a more extensive discussion), as requested by the reviewer.
(3) In section 4.2.1, right after the vertical sensitivity interpretation by Rodgers (2000), it has been added that "Note however that the sensitivity at a specific retrieval level can nevertheless be negative or exceed unity (over-sensitivity) due to kernel fluctuations and correlations between adjacent retrieval levels, as reflected in the kernel width (see below)."

(1) 13. P17, L12-14, it is not clear why the strong sensitivity variability affects vertical smoothing and Eq. 3 introduces a bias. Please make it clearer.
(2) The authors agree that this might not be clear immediately. It has now been pointed out that the sensitivity variability points at outliers in the averaging kernel matrices, thus introducing biases upon averaging kernel smoothing.
(3) It has been added that the sensitivity variability is "pointing at outliers in the averaging kernel matrices"

(1) 14. P17, L17-18 and also in Fig. 5 caption, it is not clear which is direct and centroid offset between 2nd and 3rd rows. Please make it clear in the figure caption. Also please mention the dotted lines in the figure caption.
(2) The authors agree that offset and spread indications can be improved. The caption of Figure 5 has been updated accordingly.
(3) The caption of Figure 5 has been updated as follows: "Global GOME-2A (left) and IASI-A (right) information content in terms of vertical sensitivity, retrieval offset (in km), and averaging kernel width (in km) and their dependence on DFS, SZA, or thermal contrast (TC). Black dashed lines represent median values, while out-of-range profiles are plotted in magenta. Different measures are used for the offset and kernel width in the second and third rows, which include the centroid offset and Backus-Gilbert spread, and the direct offset and FWHM, respectively. Plot titles provide the absolute and relative amounts of profiles after screening, and the number of ground-based overpass stations."

(1) 15. P17, L31, in "decreases first to about 20 km", it seems to me from the figure that the maximum median FWHM is 20 km, so should it be a smaller number here?
(2) At this point the BG spread was intended. The authors agree that this was not fully clear and have made this explicit.
(3) The beginning of the sentence has been changed into "At higher altitudes, the median BG kernel width"

(1) 16. P18, first sentence, "slant column density" should not be parallel to "the sensitivity" because the larger slant column density, the smaller the sensitivity from surface to the lower stratosphere. The real reason is because, the larger slant path length or slant column density, the fewer photons penetrating into the troposphere, the smaller the sensitivity in the troposphere, and the larger the resolving length values.

(2) The authors agree that the phrasing "and thus of the sensitivity" might be confusing to the non-expert reader. The reference to the vertical sensitivity has therefore been omitted.
(3) "and thus of the sensitivity" has been removed.

(1) 17. P20, L4, suggest changing "quarter" to "season"
(2) The authors agree that the 'seasonal' dependence is studied, yet for clarity and simplicity, full months are considered for grouping. This specification in terms of quarters is maintained throughout the text. In order to explicitly refer to the study of the "seasonal dependence" of the difference statistics however, this formulation has now been added to the text.
(3) The text has been updated as follows: "The influence quantities considered in this work are latitude (for meridian dependence), quarter (for seasonal dependence)…"

(1) 18. P20, L24-25, The sentence "This is with the exception of the Metop-B GOME-2 and IASI instruments however, that have not been used for drift studies" is difficult to understand. Suggest changing to "So Metop-B GOME-2 and IASI instruments are excluded for drift studies" and move it after "for this drift assessment"
(2) The authors agree that the reviewer's phrasing is more clear, and have adopted it accordingly.
(3) The second part of the paragraph has been changed into "To avoid spurious effects due to a seasonal cycle in the differences, only time series of fiver years or longer are used for this drift assessment. Therefore Metop-B GOME-2 and IASI instruments are excluded from the drift studies (indicated with an asterisk in Table 4). Moreover, only fully available years of the satellite datasets have been considered for comparative analysis in order not to introduce seasonal effects at the beginning and the end of each time series."

(1) 19. P20, L29, in addition to latitude and season, the influence quantity of time should be added.
(2) The authors agree; time has been added.
(3) The sentence has been changed to "These have therefore been limited to the latitude, and quarter, and time (drift)."

(1) 20. P21, the sentence above Eq. 9 is difficult to read. Suggest changing to "Yet both approaches introduce similar spatial and temporal representativeness errors into the difference statistics because taking (monthly) averages as a bias estimator XXX yields comparable outcomes:"
(2) The authors agree with the reviewer that the phrasing of this sentence can be simplified and have therefore adopted the reviewer's suggestion.
(3) The sentence before Eq. (9) has been changed into "Yet both approaches introduce similar spatial and temporal representativeness errors into the difference statistics because taking (monthly) averages as a bias estimator X yields comparable outcomes:"

(1) 21. P22, L15, is the ex-ante uncertainty from the retrievals for random noise errors or for both random noise errors and smoothing errors? As the averaging kernels are applied to reference data to remove smoothing errors, ex-ante uncertainty of random noise errors should be shown here. Please clarify this.
(2) The authors acknowledge this point with respect to error budget assessments. The ex-ante errors referred to here only include random errors. This has now been made clear in the text, with additional reference to Miles et al. (2015).
(3) The beginning of the paragraph has been changed into "The individual L2 UV-VIS comparison graphs also contain information on the validity of ex-ante uncertainties provided for the satellite nadir ozone profile retrievals (thin grey lines). The relative random error reported in the RAL v2.14 data files amounts to about 5 % at the altitude of the ozone maximum, up to about 10 % at higher altitudes, and up to 40 % in the lower troposphere. In theory the IP68 spread should be close to the combined uncertainty of the satellite data, the ground-based data, and metrology errors due to remaining differences in vertical and horizontal smoothing of atmospheric variability (including co-location mismatch errors). The latter is difficult to assess, but one can expect that the bias and spread estimates resulting from the comparisons including AK smoothing are close to the combined uncertainty of

satellite and ground-based data, or at least the ex-ante satellite uncertainty in practice (Miles et al., 2015)."

(1) 22. P22, L26, suggest adding "because the retrievals only include random noise errors and smoothing errors in the ex-ante uncertainty" after the "nadir ozone profile products"
(2) The authors have only partially adopted this suggestion in order to clarify the statement, because the ex-ante errors referred to only contain random uncertainties.
(3) "because the ex-ante uncertainty under consideration only includes random noise errors" has been added to the text.

(1) 23. P22, L26, OMI is not an exception in that the total satellite measurement uncertainty is underestimated because the ex-ante uncertainty should be compared to comparison spread or the quadratic sum of comparison bias and spread rather than the comparison bias only.
(2) The authors agree that this statement confuses the uncertainty contributions. It has therefore been changed.
(3) "The only exception is given by the OMI tropospheric ozone data that have a bias below their ex-ante uncertainty." has been replaced by "Only for the OMI tropospheric ozone data with a bias within 10 % does the combined uncertainty come close to the ex-ante uncertainty."

(1) 24. P23, L1, it is not generally true that there are smaller biases for larger total ozone columns based on the figures as the biases often increases when the total ozone increases from 300-400 to 400-500 or 500-600 DU (very clearly for GOME and OMI retrievals).
(2) The authors agree that the dependence is not linear, and have modified the text to avoid giving this impression.
(3) "Smaller biases are typically obtained in the northern hemisphere and for larger total ozone columns. The latter is indeed expected to result in an improved satellite measurement and retrieval sensitivity, and thus more stable averaging kernel behaviour with smaller vertical dependences." has been changed into "Smaller biases are typically obtained in the northern hemisphere and for intermediate to larger total ozone columns. Larger ozone columns are indeed expected to result in an improved satellite measurement and retrieval sensitivity, and thus more stable averaging kernel behaviour with smaller vertical dependences."

(1) 25. P23, L3-4, the relationship between SZA/DFS and the biases are altitude-dependent. From the figures, the biases are typically smaller at larger SZAs/DFS in the troposphere, but are larger at larger SZAs/DFS. Larger SZAs typically lead to larger total DFS due to the increase of DFS in the stratosphere and often lead to smaller DFS in the troposphere due to reduced photon penetration. Smaller biases in the troposphere at larger SZAs/DFS could be due to the reduced retrieval sensitivity in the troposphere (i.e., retrievals are closer to the a priori). So the causal relationship is not as straightforward as larger DFS means better retrieval sensitivity and therefore smaller biases.
(2) The authors thank the reviewer for clearly pointing this out. In order to introduce this vertical dependence of the SZA/DFS relationship, the text has been extended.
(3) The original statement has been replaced by the following: "On the other hand, the DFS and SZA behaviour is somewhat smaller and, as one can again expect for UV-VIS observations, rather similar, with the higher solar zenith angles typically corresponding to the larger DFS values (mainly from the stratosphere), the largest stratospheric biases, and the smallest tropospheric biases. The latter could be due however to a somewhat reduced tropospheric sensitivity, bringing the retrieved profile closer to the prior profile."

(1) 26. P33, L11, based on figures, the spread is not always enlarged in the L3 comparison. Instead, the spread is typically significantly reduced below 6 km. This should be mentioned and explained.
(2) The authors agree that this reduction of the L3 spread for the lowest level was not discussed in the original text. A statement has been added that points at this effect and provides a brief explanation.
(3) The following statement has been added: "Note however that the lack of kernel smoothing instead reduces the L3 spread for the lowest level, which has a strongly reduced sensitivity in comparison with the levels above."

(1) 27. P34, L1, says "GOME L3 data show a negative above-tropopause bias of 5-10%". But based on the first panel of Fig. 11, I see mostly positive biases above 100 hPa, especially with large positive biases of 20% around 70 hPa and positive biases of 40% around 8 hPa. Please clarify this.
(2) The authors acknowledge this mistake and thank the reviewer for his/her thorough reading. This statement has been changed and extended for clarity.
(3) The first sentence of the paragraph now reads as follows: "GOME level-3 data show an above-tropopause bias of 5-10 % positive to negative, with strong outliers around 70 hPa and 8 hPa, especially in the tropical UTLS and Antarctic local spring (up to 50 %) due to ozone hole's vortex conditions."

(1) 28. P36, L11, again the spread values for the L3 comparison can be smaller below 6 km, which should be mentioned.
(2) The authors agree that this deviation should be mentioned for completeness. This has been done by reference to the previous section.
(3) "(except for the lowest-level spread, see previous section)" has been added.

(1) 29. P40, L7-8, it is useful to explain to the readers why there is stronger tropospheric reduction to 20% and why the drift is small (e.g., due to bias correction).
(2) The authors agree that some additional explanation is helpful to the reader. The text has been extended with the requested information.
(3) The last sentences of Section 5.5 have been changed into "The L4 spread remains close to the L2 and L3 values, though with an even stronger reduction (to 20 %) in the troposphere than the L3 comparisons as no monthly averages are considered. Moreover, due to the ozonesonde-based bias correction the remaining L4 drift is of the order of a few percent only and insignificant, i.e. within the 95 % CI, for all altitudes up to about 40 km globally."

(1) 30. P45, last line, the 30% negative should be 30% positive in Antarctica as shown in last panel of Fig. 16. Also please change Table 5 correspondingly.
(2) The authors thank the reviewer for pointing out this error and have corrected the text and table accordingly.
(3) The text has been changed into "Despite this strong seasonality, and in agreement with the IASI-A L2 comparison statistics, median relative differences throughout the whole time series range between 25 % negative in the northern mid-latitudes and 30 % positive in Antarctica, with a nearly zero overall bias around the equator." In Table 5, row 9 column 6, "-30" has been changed into "30".

(1) 31. In table 5, suggest changing "Vertical resolution (6 km to troposphere)" to "Vertical grid/ resolution", changing to "115 km2", "230 by 345 km2", "12 km2", "115 km2"
(2) The "Vertical resolution (6 km to troposphere)" QA quantity and user-requirement are given by GCOS and therefore preferably left unchanged. The horizontal resolution user requirements on the other hand are provided for a single dimension by GCOS (see first column) and therefore expressed in km.
(3) No changes have been made.

(1) 32. In Figs. 6-10, 14-15, change second bracket from "[" to "]" in Fig. captions
(2) The open bracket "[" is used commonly to indicate that the last value is not included, i.e. it indicates that values included in the set go up the last value (<), but don't equal the last value (<=). The authors wish to preserve this indication as such, as it better represents the content of the sets.
(3) No action has been taken.

(1) Technical Comments:
1. P2 last line, and P8 L25, change "Keppens et al., 2015" to "Keppens et al. (2015)"
2. P4, L6, change to "time series"
3. P4, L13, P5, L15, P6, L13, change "priori" to "a priori"
4. P9, L14, change "beyond" to "above"

5. P11, change "prior" or "a-priori" to "a priori"

6. P11, suggest changing "averaging kernel smoothing of the FRM" to "smoothing the FRM with averaging kernel"

7. P12, L6, suggest changing "relative amount" to "percentage"

8. P12, L18, suggest changing "delivery. E.g." to "delivery, i.e., "

9. P13, L10, change "prior" or "a-priori" to "a priori"

10. P14, L5-6, change to "help understand"

11. P14, L16 & L26, change "prior" or "a-priori" to "a priori"

12. P21, L13, suggest changing "Thanks to" to "Due to" to make it formal.

13. P21, L21, change "vertical averaging smoothing of ground-based reference data" to "vertical smoothing of ground-based reference data with averaging kernels"

14. P21, L29-30, change "smoothing difference error" to "retrieval smoothing error" and "Tropics" to "tropics"

15. P23, L6, change to "instruments except for GOME-2B"

16. P33, L12, change to "lack of"

17. P48, L10, change to "7 km by 7 km" or "7 by 7 km2"

18. P49, L11, change to "equivalent to"

19. P49, L12, suggest changing "as an alternative" to "along with"

(2) The authors thank the reviewer for the extensive technical check and suggestions. All suggestions have been incorporated, except for those suggesting to change "prior" into "a priori" (3, 5, 9, 11). The use of the synonym "prior" sometimes improves readability and has therefore been kept. Suggestion 14 has only been partially followed: "Tropics" has been changed to "tropics" at all instances, but "smoothing difference error" has not been changed. This phrasing indicates that the error is due to the difference in smoothing between the satellite and reference profiles, and not to the satellite profile smoothing only.

(3) All suggestions have been incorporated, except for those suggesting to change "prior" into "a priori" (3, 5, 9, 11), and to rephrase "smoothing difference error".

---

## Author Response (AR2)

The response shall be structured in a clear and easy-to-follow sequence:
(1) comments, (2) author's response, (3) author's changes in manuscript.

Associate Editor Decision: Publish subject to minor revisions (review by editor) (01 Jun 2018) by Irina Petropavlovskikh
Comments to the Author:
Dear Authors,
You did a wonderful job at responding to the reviewers' comments. It improved the paper content and made information in the paper better organized. I believe this paper is almost ready to be published. I would like you to consider adding information requested by reviewers to the paper for the following comments as it would help the reader to understand information you are providing in the manuscript.

1) (1) Comment from the Reviewer 2 was "In Figs. 6-10, 14-15, change second bracket from "[" to "]" in Fig. captions." You answer was "The open bracket "[" is used commonly to indicate that the last value is not included, i.e. it indicates that values included in the set go up the last value (<), but don't equal the last value (<=). The authors wish to preserve this indication as such, as it better represents the content of the sets." Can you add your explanation of the open brackets to the manuscript to explain to the common reader (not familiar with the common practices in the field) the meaning of it?
(2) The authors agree that this explanation can be added to the text to improve readability and for the figures to be self-explanatory.
(3) In the caption of Figures 6 and 14, it has been added that "In the corresponding legend entries, open brackets are used to indicate that the last value is not included (i.e. values in the set go up to, but don't equal the last value)." The captions of the other figures make reference to the captions of Figures 6 and 14.

2) (1) Comment from Reviewer 1 was "Figure 4, first panel: Why is the area in the tropics of missing data in the GOME panel larger than that in Figure 3?" Your response was "If all data are screened (100 % values in Fig. 3) than the DFS and other information content values are empty." Can you please add the above information to the paper? It will make the discussion of data screening easier to understand.
(2) The authors agree that this explanation can be added to the text in order to improve the interpretation of Figs. 3 and 4.

[revised manuscript text omitted]